# 21st century marine climate projections for the NW European Shelf Seas based on a Perturbed Parameter Ensemble.

Jonathan Tinker[1], Matthew D. Palmer[1,2], Benjamin J. Harrison[1], Enda O'Dea[1,3], David M.H. Sexton[1], Kuniko Yamazaki[1], John W. Rostron[1]

[1] Met Office, Exeter, EX1 3PB, UK.
[2] School of Earth Sciences, University of Bristol, UK.
[3] Met Éireann, Dublin, D09 Y921, Ireland.

*Correspondence to*: Jonathan Tinker (jonathan.tinker@metoffice.gov.uk)

**Abstract.** The North West European Shelf Seas (NWS) are environmentally and economically important, and an understanding of how their climate may change helps with their management. However, as the NWS are poorly represented in Global Climate Models, a common approach is to dynamically downscale with an appropriate shelf seas model. We develop a set of physical marine climate projections for the NWS. We dynamically downscale 12 members of the HadGEM3-GC3.05 Perturbed Parameter Ensemble (approximately 70 km horizontal resolution over Europe), developed for UKCP18, using the shelf-seas model NEMO CO9 (7 km horizontal resolution). These are run under the high greenhouse gas emissions RCP8.5 scenario as continuous simulations over the period 1990-2098. We evaluate the simulations against observations in terms of tides, sea surface temperature, surface and near-bed temperature and salinity, and sea surface-height. These simulations represent the state-of-the-art for NWS marine projections. We project a Sea-Surface Temperature (SST) rise of 3.11 °C ($\pm 2\sigma = 0.98$ °C), and a Sea-Surface Salinity (SSS) freshening of $-1.01$ psu ($\pm 2\sigma = 0.93$ psu) for 2079-2098 relative to 2000-2019, averaged over the NWS (approximately bounded by the 200m isobar and excluding the Norwegian Trench, Skagerrak, and Kattegat), a substantial seasonal stratification increase (23 days over the NWS), and a general weakening of the NWS residual circulation. While the patterns of NWS changes are similar to our previous projections, there is a greater warming and freshening, that could reflect the change from the A1B emissions scenario to the RCP8.5 concentrations pathway or the higher climate sensitivity exhibited by HadGEM3-GC3.05. Off the shelf, south of Iceland, there is limited warming, consistent with a reduction in the Atlantic Meridional Overturning Circulation and associated northward heat transport. These projections have been publicly released, along with a consistent 200-year present day control simulation, to provide an evidence-base for climate change assessments and to facilitate climate impact studies. For example, we illustrate how the two products can be used to estimate of climate trends, unforced variability, and the Time of Emergence (ToE) of the climate signals. We calculate the average NWS SST ToE to be 2034 (with an 8-year range) and 2046 (with a 33-year range) for SSS. We also discuss how these projections can be used to describe NWS conditions under 2°C and 4°C global mean warming (compared to 1850-1900), as a policy relevant exemplar use-case.

## 1    Introduction

The North West European Shelf Seas (NWS) are economically and environmentally important (Pugh, 2008; Tinker et al., 2018). They are quasi-isolated from, and behave differently to, the North Atlantic, and have a different response to climate change (Holt et al., 2010; Wakelin et al., 2009). They are generally poorly represented in Global Climate Models (GCMs) and global ocean models due to limited vertical and horizontal resolution and the absence of key processes, such as tides (Tinker et al., 2022). Therefore, assessing the NWS response to climate change directly from GCMs may be inappropriate for many applications (Hermans et al., 2020b; Tinker et al., 2022; Mathis et al., 2013). A common approach to mitigate these shortcomings is to dynamic downscale GCM climate projections with a higher resolution shelf seas model, which includes the additional relevant processes (e.g. Holt et al., 2010; Tinker et al., 2016, 2020).

The marine component (Palmer et al., 2018) of the recent national UK Climate projections, UKCP18 (Murphy et al., 2018), had a sea level focus (see their table 5.1). UKCP18 provided mean sea level projections (for the 21st century and exploratory extended projections to 2300, Palmer et al., 2020), estimates of the change in the surge and wave climate (Howard et al., 2019), and a quantification of the present day sea level variability (Tinker et al., 2020). The climate projections of ocean water properties for the NWS from UKCP09 (Lowe et al., 2009) and the Minerva Projections (Tinker et al., 2015, 2016) were not updated in UKCP18. The UK's third Climate Change Risk Assessment (CCRA3) used UKCP18 as a primary source of evidence. However as the NWS climate projections had not been updated assessment of those aspects was based on the older UKCP09/Minerva NWS projections. As well as being based on outdated science, use of the earlier NWS projections may introduce inconsistencies with the wider CCRA3 findings based on UKCP18. With the evidence call for the upcoming 4[th] CCRA approaching, we have developed a set of NWS climate projections, based on, and consistent with the UKCP18 marine projections.

Beyond the UK, there are numerous legislative mechanisms that support the protection and management of the NWS Marine Environment. At the European level examples include: the EU Common Fisheries Policy (CFP, 2013); The EU Habitats Directive (1992); The Oslo-Paris (OSPAR) Convention for the Protection of the Marine Environment of the North-East Atlantic (1993); and The EU Marine Strategic Framework Directive (2008). There are also important international treaties and conventions such as the United Nations Framework Convention on Climate Change (UNFCCC, 1992) and the Convention on Biological Diversity (CBD, 1992). These are all implemented at the UK level (Frost et al., 2016) and all have varying and overlapping goals and targets linked to the protection and monitoring of the marine environment. While few of these policies explicitly mention climate change, they capture its effect through natural variability and broader environmental change (Frost et al., 2016). For example, the MSFD aims to achieve Good Environmental Status (GES), which varies by region across the EU waters. After an initial series of surveys to define GES, the MSFD has driven a series of monitoring programmes to chart the progress towards GES and the implementation of a set of management strategies (programs of measures) to help its achievement. Marine climate projections help inform all three steps in this process – without an idea of how the marine climate may change, it is difficult to make informed management decisions, and monitoring must consider the background climate change to help interpret any observations. Similarly, the OSPAR evidence groups (Physical Evidence Group (PEG), Clean and Safe Seas Evidence Group (CSSEG), Healthy & Biologically Diverse Seas Evidence Group (HBDSEG)) all drive monitoring programmes that report via the OSPAR quality status report. OSPAR has adopted an ambitious strategy for the North-East Atlantic region (North-East Atlantic Environment Strategy 2030), which has a strong emphasis on climate change and ocean acidification. ICES (International Council for the Exploration of the Sea), the body that provides impartial evidence the underpins the CFP, derives this evidence with an exhaustive monitoring strategy, which are reported on in the ICES State of the Ocean Climate reports (e.g. González-Pola et al., 2018). All these legislative mechanisms, and the monitoring and management that they underpin, provide a framework though which to observe how the climate is changing seas around of the NWS. There is also an increased focus on developing conservation and planning legislation that takes into account future projections of climate change (Queirós et al., 2021; MCCIP, 2003). Effective marine management and adaptation planning

requires information on the past and present environment through monitoring and information on future changes from climate projections.

In this paper (and dataset) we downscale the UKCP18 12-member Perturbed Parameter Ensemble (PPE), based HadGEM3-GC3.05, run under the RCP8.5 high emissions climate change scenario. The PPE simulations are downscaled with the shelf
seas version of NEMO 4.04 (Coastal Ocean version 9, CO9), as transient simulations for the period 1990-2098. We use the methodology and evaluation developed by Tinker et al. (2020) to produce a set of projections consistent with their estimate of NWS unforced variability. These updated NWS climate projections (hereinafter NWSPPE) are released, with the present-day control simulation (hereinafter PDCtrl) of Tinker et al. (2020), in time for use in the UK's 4[th] CCRA.

## 1.1 Choice of RCP8.5

Our set of projections are based on the RCP8.5 concentration trajectory. RCP8.5 is a relatively high impact "business as usual" scenario from the CMIP5 suite of models, rather than the more recent CMIP6, which are based on the "Shared Socioeconomic Pathway" (SSPs). RCP8.5 has very similar total radiative forcings to the SSP5-8.5 scenario (Tebaldi et al., 2021), although RCP8.5 has a slightly weaker global temperature response, attributed to a lower $CO_2$ concentration (Fyfe et al., 2021). The choice of forcing scenario is motivated primarily by the desire for a set of NWS marine climate projections consistent with the
latest set of UK Climate Projections (UKCP18), which were run under RCP8.5 and before the SSP scenarios became available. This consistency allows researchers to look across both the land and marine domains in multi-variate space in a way that has previously not been possible to facilitate, e.g., consideration of compound hazards and the combined effects of multiple climate-impacts drivers.

Recent studies have criticised the use of RCP8.5 in climate projections, particularly in terms of its apparent low likelihood,
given emissions reductions pledges associated with the Paris Agreement (Hausfather and Peters, 2020). However, there are several scientific reasons why this remains a useful scenario in the context of policy-relevant climate information. Firstly, it has a high signal-to-noise ratio and can therefore better separate the forced climate response from internal variability. Secondly, RCP8.5 can be readily translated into warming levels, which was the primary basis of the last UK Climate Change Risk Assessment and appear prominently in the 6[th] IPCC Assessment Report (AR6; IPCC, 2021a). Thirdly, risk-based decision
making requires a comprehensive picture of the future risk landscape, including higher warming levels associated with any combination of emissions back-tracking, positive carbon-cycle feedbacks, and high climate sensitivity(IPCC, 2021b). Finally, high emissions scenarios such as RCP8.5 provide a useful baseline scenario from which the benefits of mitigation action and avoided costs can be assessed.

## 2    Models and Methods

We use CO9 to downscale the UKCP18 HadGEM3-GC3.05 PPE (Yamazaki et al., 2021). This ensemble is downscaled as transient simulations from January 1980-November 2099, but the first 10 years are considered spin up, and so are discarded. This leads to a 12 member PPE downscaled for the NWS (NWSPPE), from January 1990 to December 2098, for RCP8.5. NWSPPE is consistent with PDCtrl, which can provide an estimate of a different source of uncertainty, and so the two datasets

are released together.

Here we give a brief overview of: HadGEM3-GC3.05 (our driving GCM); and NEMO 4.0.4 CO9 (our regional ocean model), its treatment of tides, and where the model code is available. A detailed description of HadGEM3-GC3.05 and its evaluation for the NWS is given in Tinker et al. (2020). We then describe how HadGEM3-GC3.05 has been run as the Perturbed Parameter Ensemble before describing some of our methodologies.

### 2.1    HadGEM3-GC3.05

HadGEM3-GC3.05 (Williams et al., 2018; Yamazaki et al., 2021) is the Met Office GCM used in the 2018 UK Climate projections (UKCP18) and very similar to the model submitted to CMIP6 (Coupled Model Intercomparison Project Phase 6, Eyring et al., 2016). HadGEM3-GC3.05 is based on the Met Office Unified Model atmosphere (Walters et al., 2019) (with approximately 70km resolution of Europe) and the ocean model NEMO, run on the ORCA025 grid ($\sim 1/4°$ horizontal resolution on a tri-polar grid). HadGEM3-GC3.05 uses the CICE sea ice model (Hunke et al., 2015) with the GSI8.0 sea ice

configuration (Ridley et al., 2018) on the same ORCA025 grid. HadGEM3-GC3.05 does not simulate tides or other important shelf seas processes. HadGEM3-GC3.05 is run with a 360-day calendar, with 12 months of 30 days. This has little impact on the climate projections, but is not directly compatible with observed tides, which are constrained by astronomical frequencies. This is discussed later.

We use HadGEM3-GC3.05 model output from the atmosphere and the ocean model to drive our downscaled shelf seas simulations. HadGEM3-GC3.05 atmospheric surface fluxes of heat, fresh water and momentum are used, as are ocean temperature, salinity, currents, and sea surface height from the HadGEM3-GC3.05 ocean. Tinker et al. (2020) lists all the HadGEM3-GC3.05 variables used to drive our simulations, and their frequency, in their Table S9.

#### 2.1.1    The HadGEM3-GC3.05 Perturbed Parameter Ensemble

HadGEM3-GC3.05 was run as a Perturbed Parameter Ensemble (PPE) for the UKCP18 climate projections. To explore the uncertainty associated with the choice of parameters and parameterisations within the GCM, a PPE perturbs the parameters (within a reasonable range) and chooses different parameterisations for each ensemble member. As more ensemble members are added, the sensitivity to these parameters increases the ensemble spread. This spread was designed to span the climate uncertainty associated with these uncertain parameters. The GCM simulations used, and the PPE framework are described in

Sexton et al. (2021) and Yamazaki et al (2021).

Yamazaki et al (2021) ran a 25 member HadGEM3-GC3.05-PPE for the period 1900–2100, using CMIP5 historical and RCP8.5 emissions. The variants had different combinations of perturbations to 47 parameters in the model's atmosphere, land, and aerosol schemes, and they were selected by running a set of cheap, coarser-resolution atmosphere-only simulations from a large sample of nearly 3000 variants. Poor performing variants were filtered out by assessing retrospective 5-day weather

forecasts and simulations of 2004–2009 with prescribed SST and sea ice. Further idealised atmosphere-only experiments were then used to select 25 variants which were as diverse as possible in terms of their atmospheric climate feedbacks and their regional responses to warmer SSTs (Sexton et al., 2021). These variants were then run with HadGEM-GC3.05 for historical and future simulations from 1900-2100 under RCP8.5. 10 members were dropped as being too cold by 1970, or when evaluated compared against 13 CMIP5 models over the historical period (1900-2005). The resulting 15 ensemble members gave plausible

yet diverse atmosphere and ocean model behaviours (Yamazaki et al., 2021), and were used within the UKCP18 climate projections. Of this 15-member ensemble, 3 ensemble members exhibited unrealistic Atlantic Meridional Overturning

Circulation (AMOC) behaviour and so were downscaled for the NWS, but were excluded from our analysis. The remaining 12 downscaled PPE is referred to as NWSPPE.

A heat and salinity flux adjustments as applied to the HadGEM3-GC3.05 PPE to prevent the surface climate from drifting too far from the realistic state despite the parameter perturbations giving rise to a range of net top-of-atmosphere radiative fluxes. Details of how the flux adjustments were applied are described in Yamazaki et al. (2021).

A recent study compared UKCP18 PPE simulations for RCP8.5 to a selection of available CMIP6 models (Murphy et al., 2023). The evaluation of the climatology, large-scale modes of variability and the historical change in global mean surface temperature, showed that the overall level of performance from the UKCP18 global PPE simulations was competitive with the CMIP6 ensemble. Both sets of simulations exhibit a range of biases across their individual members, with evidence of a systematic component to the biases for some variables (Murphy et al., 2023). When looking at the multidecadal average changes, the CMIP6 ensemble develops a broad range of warming signals in response to SSP5-8.5 (e.g. 2061-2080 relative to 1981-2000), commensurate with a broad range of climate sensitivity values among models (e.g. IPCC, 2021c). This wide CMIP6 range show substantial overlap with those of UKCP18 global PPE for both surface temperature and precipitation, confirming the continued importance of accounting for the relevant uncertainties in impacts studies. Both UKCP18 global PPE and CMIP6 estimates span positive and negative changes in winter and summer North Atlantic Oscillation (NAO), although the UKCP18 show a slight narrowing of the range of summer NAO compared to CMIP6. The study highlights importance of assessing sources of difference sources of uncertainty (including structural modelling uncertainty) which is often overlooked in regional marine climate projections (see section 6 for further discussion).

## 2.2    NEMO AMM7

We use the shelf seas version of NEMO (O'Dea et al., 2017) to dynamically downscale HadGEM3-GC3.05. It is run on the AMM7 NWS domain, which extends from 40° 4′ N 19° W to 65° N 13° E with a 7km horizontal resolution with hybrid terrain following vertical levels (s-levels, Siddorn and Furner (2013) for the PDCtrl and Bruciaferri et al. (2018) and Wise et al. (2022) for the transient projections).

Two slightly different version of the model are run for the Present-Day Control Simulation and the climate projections. Coastal Ocean version 6 (CO6), based on NEMO version 3.6, was used in the PDCtrl – see Tinker et al. (2020) for full details. Coastal Ocean version 9 (CO9, NEMO 4.04) is used for the NWSPPE climate projections. Tests showed little difference between the two models, so we consider the two versions to be consistent. We use COx when we refer to both model versions.

COx is driven by several different classes of boundary conditions. The atmospheric and ocean boundary conditions are provided by HadGEM3-GC3.05, and are interpolated onto the COx grid. The Baltic exchange and the rivers are specified by observation based climatologies. Full details of how these are implemented and their frequencies are given in Tinker et al. (2020) and their Table S9 in particular.

### 2.2.1    NEMO AMM7 tides

The standard version of NEMO introduces errors into the tides when using the 360-day climate. It resets the tides at the beginning of each month, and then runs sequentially. This leads to a jump at the end of the months with lengths other than 30 days. This doesn't seem to cause the model to crash but must affect the internal model solution. It also, on average, changes the $M_2$ period and so rendered standard tidal analysis useless.

We have adapted the tidal module within NEMO to work with the 360-day calendar, with three different methods: "reset", the standard method, where the tides are reset to every month; "drift" (which is used here), where we allow the tides to run continuously from a given date, but the annual tides (such as the equinoctial tides) drift by about 5 days every year, compared to the Gregorian calendar; and "compress", where we alter one of the astronomical frequencies within the tide module - this changes the $M_2$ (and most other) tidal frequency by ~2 seconds, but keeps the equinoctial tides with their correct timings. We describe these three methods and evaluate the impact they have on the tidal behaviour in appendix B. In these simulations, we

use the drift method, where the model time, in number of seconds since 1$^{st}$ Jan 1980 (in the 360-day calendar), is counted forward with the Gregorian calendar to give the tidal conditions.

Both model versions use the same tidal constituents and tidal boundary conditions, however in the later version of NEMO (4.04) the default tidal Love Number was changed from 0.7 to 0, effectively turning off the internal tidal generating potential. This was not realised until the projections had been completed. In a small domain like the NWS, most tidal energy is propagated in from the boundary, and very little energy is internally generated, however this slightly affects the tidal range. A sensitivity test was run (running the unperturbed ensemble member with the tidal generating potential on and off) showing this only had a small impact on the tidal range (Figure S1), changing the amplitude of the $M_2$ tide by up to 4cm (which is up to about 3.5m). This was not considered significant for the climate projections, although in a different context (such as operational surge forecast systems), this could be more of a problem.

### 2.2.2    NEMO COx GitHub configurations.

Both NEMO configurations used in this study are available in GitHub.

NEMO version 3.6 CO6 (AMM7)

https://github.com/hadjt/NEMO_3.6_CO6_shelf_climate

Development of this configuration has frozen. The latest version available on GitHub was used in PDCtrl.

NEMO version 4.0.4 CO9 (AMM7 and AMM15)

https://github.com/hadjt/NEMO_4.0.4_CO9_shelf_climate/

Development of this configuration has continued since the NWSPPE was run.

The version used for NWSPPE is available as:

https://github.com/hadjt/NEMO_4.0.4_CO9_shelf_climate/tree/bddd0f68980632229c5afef9772c9fd0d0d6e930

A fix has since been added to set the default tidal Love Number to 0.7:

https://github.com/hadjt/NEMO_4.0.4_CO9_shelf_climate/tree/4ef0ec5e0c20a9aa88b42f395cc9b1bfd689a221

### 2.3    Present-Day Control Simulation

HadGEM3-GC3.05 was also run as a present-day control simulation for UKCP18, and underpinned Tinker et al. (2020). It was run for 270 years while simulating a stable climate equivalent to the year 2000. This was forced with green-house gases and aerosols from the year 2000 (as fixed values, or annually repeating seasonal cycles). This allowed the model to simulate the natural, unforced climate variability that would have occurred during the year 2000 and the absence of climate change. This simulation should capture the range of climate variability associated with different phases of climate modes such as El Niño Southern Oscillation, North Atlantic Oscillation and Atlantic Multidecadal Variability. The first 70 years of the simulation still show trends associated with the deep ocean spinning up, so the last 200 years were used for analysis by Tinker et al. (2020) and form the second part of this dataset. We refer to the downscaled control run as PDCtrl.

### 2.4    Copernicus Marine NWS Reanalysis (RAN)

A Met Office led consortia provides a marine reanalysis for the NWS to the Copernicus Marine Service (product number: NWSHELF_MULTIYEAR_PHY_004_009). This ran from 1993 to present, using the same regional shelf seas model and domain as used here (NEMO CO6). By running a shelf seas model, and assimilating observations (SST, temperature and salinity profiles, and satellite altimeters), it can be considered a best guess estimate of the current state of the NWS.

We make limited use of this NWS Reanalysis (hereinafter RAN), but comparing its regional mean timeseries (Tinker et al., 2019) to those of our NWSPPE. This gives some early-century context to the NWSPPE temporal evolution in terms of interannual variability and trends.

## 2.5    Circulation, currents, and transport

Residual currents are the small remaining currents after the large oscillatory tidal currents are averaged out. They are an important feature of the NWS, and are an important determinant of the spatial distribution of heat, salt and nutrients. We use two methods to analyse the NWS residual circulation: spatial maps of barotropic (depth-mean) residual currents; and timeseries of transport through pre-defined cross-sections. Following Tinker et al. (2022), we use 30-day mean barotropic currents (all simulated months have 30 days with a 360-day calendars) as a proxy for the residual circulation because they effectively average out the $S_2$ and $M_2$ tide – 720 hours is 57.97 cycles of the $M_2$ tide (Tinker et al., 2022). We initially consider the spatial configuration of the NWS residual circulation using the magnitudes of the mean currents.

We also assess the how much the currents vary about their mean using the residual current uncertainty ellipse method of Tinker et al., (2022). Normally distributed U and V currents together have a bivariate normal distribution (Figure S2a), which follows a gaussian surface rather than a gaussian curve. Contours (of equal probability) on that surface encloses a given proportion of the data – the ellipse associated with 2.45 standard deviations encloses 95% of the data (Figure S2bc). By fitting bivariate normal distribution and these ellipses to the residual currents we can use the ellipse properties to analysis the circulation in novel ways (Figure S2c-f).

We can use the ellipse properties to compare the mean of the residual currents to their variability. When the mean is greater than their variability the residual currents are relatively steady (not stopping, or reversing), and when the variability is greater than the mean, the residual current may have a preferential direction and strength, but at any given time, they could be going in any direction. We can also use this approach to compare two sets of residual current distributions, from different times, or from different ensemble members, and see how similar they are by assessing the overlap of their bivariate normal distribution (OVL, Figure S2hi). The OVL method finds the volume (or area) under two Gaussian surfaces (or curves) to quantify how similar they are. The volume under a single Gaussian function integrates to 1, so the integral under two Gaussian functions must integrates to ≤ 1 (see Figure S2i), but also must be greater than 0, as Gaussian functions extend to infinity. OVL ≈ 1 when the two distributions are very similar (in terms of their means, variances and co-variance) and OVL ≈ 0 when they are very different (see Figure S2i). We extend the OVL method of Tinker et al., (2022) to quantify how similar the residual current distributions are across the ensemble (ens_OVL). By finding the volume under the bivariate normal distribution for all 12 ensemble members, we get a pointwise measure of the similarity of the residual circulation across the NWSPPE and can then show how this changes over the NWS, and over the 21st Century.

Tinker et al., (2022) give a full description of the methodology, and provide a python toolbox on github. We have included their figure explaining their methodology (Figure S2).

Timeseries of transport through pre-defined cross-sections are calculated every timestep at run time. 76 cross-sections across the domain quantify the net, positive and negative transport of volume, heat and salt. We also output spatial maps of monthly mean transport (the instantaneous product of each current component with the depth of water), and are able to calculate the net volume cross-section transport – this allows us to add additional cross-sections *post hoc*.

We also use these volume transport cross-sections to help evaluate the model, using the same cross-sections and observations as used in UKCP09 (Lowe et al., 2009; Holt et al., 2010).

## 2.6    Regions and Regional Mean statistics

Timeseries are useful to show temporal evolution and how an ensemble behaves. We use a predefined region mask to calculate timeseries of the regional means (and other statistics). We use the region mask of Wakelin et al. (2012) as it divides the NWS into regions that make geographic and oceanographic sense. We also include a "shelf" region, by combining several NWS regions (excluding the Atlantic, Norwegian Trench, Skagerrak/Kattegat and Armorican shelf regions). When considering the sparse EN4 datasets, we combine these regions into larger validation regions (Tinker et al., 2019) to improve the data coverage, and sample size. While there is relatively high data coverage within the North Sea, other NWS regions are still relatively data poor, especially given the need to average out the differing phases of variability, to compare the model and observations

"climate" rather than "weather". For completeness, we include the EN4 analysis on the Wakelin et al. (2012) mask in the additional materials (Figure S3).

### 2.7 Normalised Bias

During our evaluation, we calculate the normalised bias between the model and observations, to show where the observations sit within the distribution of the ensemble. To do this, we subtract the observations from the ensemble mean, and divide by the ensemble standard deviation:

$$Norm\ Bias = \frac{Ens\ Mean - Obs}{Ens\ Std}$$ ( 1 )

This gives the number of (ensemble) standard deviations the observations are above or below the ensemble mean. We typically consider a value with an absolute normalised bias less than 2 to be within the NWSPPE.

### 2.8 Potential Energy Anomaly

We use the Potential Energy Anomaly (PEA, Simpson and Bowers, 1981) to quantify of the strength of the stratification. PEA is equivalent to the amount of energy required to fully mix a stratified water column.

PEA is quantified internally by NEMO as:

$$PEA = -\frac{g}{h} \int_{z=-h}^{0} z\left(\rho(T,S) - \rho(\bar{T},\bar{S})\right) dz$$ ( 2 )

where $h$ is the depth of water (limited to the upper 400 m), $g$ is gravity, $z$ the vertical coordinate (positive upwards), and an over bar represents depth average.

The thermal and haline component of PEA (PEAT and PEAS) can also be calculated by ( 2 ) by replacing the $S$ with $\bar{S}$ for PEA and $T$ with $\bar{T}$ for PEAS.

We follow the Holt et al. (2010) definition of stratification, where the water column is stratified when PEA > 10 J m$^{-3}$ and (mixed layer depth) MLD < 50m following.

### 2.9 Stratification initialisation and duration

Daily mean fields of PEA and MLD are used to assess the timing and duration of stratification.

For each year, and each grid box, we identify stratified days (PEA > 10 J m$^{-3}$, MLD < 50m), and days where stratification initialises (a stratified day following a mixed day) and breaks down (a mixed day following a stratified day). We note the grid boxes that are stratified throughout the year, and those that are mixed throughout the year. We then cycle through these periods of stratification and count their duration. We identify the longest single period of stratification and note the day of year that it initialises, breaks down, and its duration. This discards short periods of stratification that may proceed the main period of stratification, or may occur after the main break down. We also note the number of periods of stratification in the year, and the total days of stratification. This method assumes that seasonally stratifying regions are mixed in the winter; it doesn't account for stratified periods that continue from one year to the next. We then calculate 20-year means of the stratification duration, and day of the year of the initialisation and break down. These are then considered across the ensemble. This method is consistent with that of Holt et al. (2010), and Jardine et al., (2023)

The analysis was also repeated with a stratification threshold of the DFT (surface minus near bed temperature differences) being greater than 0.5 °C, and PEAT > 10 J m$^{-3}$, both of which gave a similar result.

### 2.10 Estimation of Time of Emergence (ToE)

We estimate the Time of Emergence (ToE) of the climate signal for SST and SSS, using the method of Lyu et al., (2014). We compare a modelled estimate of the present-day unforced variability (the noise, $N$), to the smoothed climate trend of an

ensemble (the signal $S$) and ask when the ratio of the signal to noise is greater than a given threshold (2 standard deviations),
310 doesn't drop below threshold for the rest of the record, and is at least 20 years long. We calculate the ToE for each ensemble member, and then report the median value, and the $16^{th}$-$84^{th}$ percentile range.

We estimate the Noise as the present-day unforced variability by taking the standard deviation of the linearly detrended PDCtrl annual means. As the PDCtrl has very little trend, detrending the PDCtrl has little effect on the ToE. We calculate the signal of each ensemble member by smoothing with a $4^{th}$ order polynomial and convert to anomalies by removing the 2000-2019
315 baseline-period. We then find where the $S$:$N$ ratio exceeds the threshold, remains above the threshold for the rest of the simulation. We make one further criterion, that the ToE doesn't occur in the final 20 years of the projection (and so the emergence time is at least 20-year long), as they may not represent the true emergence of climate signals.

## 3 Observational Datasets

In this section we describe the observational datasets used in our NWEPPE evaluation (Table 1). Tinker et al. (2020) evaluated PDCtrl, and we follow a similar approach here.

### 3.1 OSTIA

We use the OSTIA (Operational Sea-Surface Temperature and Sea-Ice Analysis) SST analysis product (Roberts-Jones et al., 2012) to evaluate the NWSPPE SST. OSTIA is a largely satellite-based SST dataset on a horizontal grid of 0.25° which we bi-linearly interpolated onto the AMM7 model grid. The OSTIA assimilates several bias corrected satellite products to reduce the bias of the overall product.

We calculate an SST climatology (mean and standard deviation) for 2000-2019, from monthly, seasonal, and annual mean OSTIA data. This is compared to the NWSPPE ensemble mean for the equivalent period (Figure 2). We also assess where the OSTIA 20-year mean sits within the NWSPPE (when processed equivalently), using the normalised bias in ( 1 ), asking how many NWSPPE ensemble standard deviations OSTIA is form the NWSPPE ensemble mean.

### 3.2 EN4

We evaluate the surface and near bed temperature and salinity of the NWSPPE with the EN4 quality-controlled temperature and salinity profile dataset (Good et al., 2013). We use the individual observed vertical profiles of version EN.4.2.2.g10.

As the EN4 data are relatively sparse on the NWS, it is a complex dataset to use in a climate context, and so we follow a similar methodology to Tinker et al. (2020). There are few EN4 locations where observations are available for every year for a given month and grid box, and so it is seldom possible to separate the observed climatology from the observed interannual variability. Therefore, rather than comparing the observed climatology from the distribution of NWSPPE climatologies (the distribution of the 20 year means from the NWSPPE), we compare the EN4 observation to the NWSPPE distribution of both interannual variability and ensemble variability (12 ensemble members and 20 years). This is a like-for-like analysis. This approach gives a reasonable spatial coverage of the NWS, albeit with very low number of repeat samples.

#### 3.2.1 EN4 evaluation methodology

This comparison is designed to evaluate the broad characteristics of any model biases. A more detailed comparison is beyond the scope of the present study.

Creating the NWSPPE distribution fields.

1. We create a NWSPPE climatology.
   a. For each of the 12 months we average all the monthly means from all the ensembles between 2000 and 2019.
   b. As the EN4 dataset is sparse, for each of the 12 months, we also calculate the ensemble standard deviation across all 240 points (20 years and 12 ensemble-members).

Pre-processing the EN4 data

2. We discard the profiles with QC flags ('POSITION_QC', 'PROFILE_POTM_QC', 'PROFILE_PSAL_QC') = 4.
3. For each month, and for each year, we:
   a. We assign each profile to the nearest COx grid box, and linearly interpolate it onto the vertical s-levels.
   b. If there are more than one profile for a given month, we average them.
   c. We calculate the normalised bias:

$$NormBias = \frac{\left(PPE\_mean_{i,j,z,m} - Obs_{i,j,z,y,m}\right)}{PPE\_std_{i,j,z,m}}$$

Where $i, j, z$ are the grid index in three dimensions and $y$ and $m$ are the year and month.

   d. For each month, for each grid box we add all the NormBias from all the years, add the square of the NormBias, and count the number of years with an observation (Sum_NormBias, SSq_NormBias, cnt_NormBias).

e. For each month, we calculate the mean and the standard deviation of the NormBias for each grid box (Mean_NormBias, Std_NormBias).

f. We combine Sum_NormBias, SSq_NormBias, cnt_NormBias for each month into seasons (DJF, MAM, JJA, and SON), and annual means.

This gives us a gridded assessment, for each month, season, and year, of where the EN4 observations sit within the NWSPPE (SST: Figure S4; SSS: Figure S5; NBT: Figure S6). For each grid box with an EN4 observation, we can then give the number of NWSPPE standard deviations it is from the ensemble mean. If the grid box (for a given season or month), has observations from different years, they are averaged. These sparsely sampled maps are hard to use quantitatively, and so regional summaries are also produced (Figure 3, and Figure S3).

### 3.3 ICES climate timeseries

While the EN4 data gives a reasonable coverage of the NWS, very few points have enough observations to allow the climate to be quantified in terms of its mean, and variability. We supplement the EN4 dataset with several long timeseries observations, that expand our comparison of the NWSPPE simulations and the observations.

ICES (International Council for the Exploration of the Sea) support as series of sustained observation timeseries to allow long term monitoring of the ocean climate of the North Atlantic. These are formed by regular measurements of ocean temperature and salinity over decades (González-Pola et al., 2022). Furthermore, many ICES records are for the subsurface, which is not captured by satellites.

We evaluate the NWSPPE temperature and salinity against 11 ICES ocean climate timeseries (González-Pola et al., 2022; Hughes et al., 2018) listed in Table 2 (Data retrieved from https://ocean.ices.dk/core/iroc on the December 2023). Of these 11, 8 are from the NWS, two to the north of the NWS, towards the Faeroe Islands (Faroe Shetland NAW, Faroe Shetland MNAW), and one from the Norwegian Trench (Utsira B). The timeseries are a mix of surface, near bed, depth averaged, the average of a portion of the water column, or of a particular water bodies, and are listed in Table 2. We therefore extracted the annual mean temperature and salinity water column data from the nearest model grid box, and applied the equivalent depth processing. We compare data from 2000-2019, and consider whether the observations are within the range of the NWSPPE, as well as comparing the mean and the interannual variability (Figure 4, Table S1). We calculate the normalised bias following ( 1 ), to show how much warmer/saltier the model is from the observations, in ensemble standard deviations. To assess the variability of the model and observations, we calculate the standard deviation of the ICES timeseries and for each of the 12 ensemble members, and then calculate the normalised bias following ( 1 ).

### 3.4 Satellite SSH data

We evaluate the NWSPPE SSH following the methodology of Tinker et al. (2020), where we refer the reader for evaluation of PDCtrl SSH. We compare the mean pattern of the modelled sea level to satellite altimetry Mean Dynamic Topography (MDT), which indicates the average strength of the geostrophic currents (Hermans et al., 2020a). The pattern of the interannual SSH variability is compared to a satellite altimetry sea level anomaly product.

We use the AVISO CLS18 MDT product (Figure 5c), estimated from the period 1993–2012 (Mulet et al., 2021), and compare to the NWSPPE ensemble mean SSH for the present-day ensemble statistics (2000-2019). As AVISO has a lower horizontal resolution (0.25°), and is not able to resolve features with spatial wavelengths less than ~ 180 km (Legeais, 2018), we smooth model by convolving with a uniform filter of 13-by-13 grid boxes (~ 90 km, Figure 5a).

We compare the simulated interannual SSH variability to that of the Copernicus Climate Change Service (C3S) Sea Level Anomaly (SLA) product (Legeais et al., 2018). We compute annual means (1993–2018) of the (daily, 0.25° horizontal resolution) C3S SLA product, linearly detrend (temporally), and calculate the temporal standard deviation. This is compared to the interannual variability from the present-day ensemble statistics (2000-2019) - the square root of $\sigma_{int}^2$ from ( A1 ) in section 15.4.3.

### 3.5 Tide Gauge data

The NWSPPE was compared to 20 tide gauges around the UK and NWS (Figure 6, Table 3), which were selected for the length of overlap with the NWSPPE. These were downloaded from the Permanent Service for Mean Sea Level (PSMSL, Holgate et al. (2013); Data retrieved 24th of November 2022), and processed into annual means. We use PSMSL monthly revised local reference (RLR) data to account for changing baselines and we reject data with quality issues (calculations for mean tide level, suspect data, etc.). Annual mean timeseries are created from monthly mean anomalies (where the climatological season cycle has been removed) where there is data from at least 11 months.

The SSH from the nearest (sea) grid box was extracted from the monthly mean data, and processed into annual means, for each ensemble member and then processed into ensemble means. The tide gauges were not corrected for Glacial Isostatic Adjustment (GIA), and so Smögen (which exhibits substantial GIA) was excluded. For each tide gauge an offset between the ensemble mean was removed. This was the difference between the NWSPPE Ensemble mean temporal mean and tide gauge temporal mean, for the common overlap period.

### 3.6 Volume Transport estimates

We assess the NWSPPE volume transport through several cross-sections to observational estimates from the literature (Fernand et al., 2006; Brown et al., 1999; Svendsen et al., 1991; Turrell et al., 1992; Danielssen et al., 1997; Prandle et al., 1996; Holt et al., 2001), following Lowe et al. (2009). We define these cross-sections in Table 4. Most observational estimates are for the net transport, apart from cross-section 1, 4, 5 and 12 (Figure 7m), which are for the positive transport component. Most modelled transport cross-sections timeseries have been processed *post-hoc*, and so are limited to net transport, sections 1, 4, 5 and 6, being the exceptions, where the net, and positive and negative transport components are available. Section 12 (Shelf Current FS current) is based on the observed positive (north-westward) transport component, and compared to the net modelled transport – therefore the modelled estimates include a negative component, which would reduce the magnitude of the modelled transport.

### 4 Evaluation

We evaluate the NWSPPE between 2000 and 2019 for tides, temperature, salinity, sea surface height, and transport (Table 1). We show the modelled co-tidal charts and compare these to other modelling systems. We compare the modelled SST with the OSTIA analysis (Roberts-Jones et al., 2012), and temperature and salinity with the EN4 profiles dataset (Good et al., 2013) and ICES climate timeseries (González-Pola et al., 2022). We compare the modelled sea surface height to tide gauge data retrieved from the Permanent Service for Mean Sea Level (PSMSL, Holgate et al. (2013) and two satellite products (Rio et al., 2014; Legeais et al., 2018). We compare modelled transport to estimate calculated from observations from the literature.

Evaluating climate projections is always complicated by the fact that they are not designed to simulate the observed phase of weather and climate variability. For example, while the approximate number and frequency of warm years should agree with observations, their ordering will not. This means the climate must be evaluated, rather than the "weather". To do this, we compare the statistics of long records of observations with the models – we would expect a level of agreement between the 20-year mean (and standard deviation) of the model and the observations. This works well for SST, where there are satellite records since the 1990s, but other long timeseries of observations are relatively sparse. Another complication is that reality is a single realisation of what conditions could be expected, given the current state of the climate. If there were 100 worlds with today's greenhouse-gas and aerosol conditions, we would expect a range of temperatures, due to unforced variability – different states of the North Atlantic Oscillation, different states of the Atlantic Multidecadal Variability, different weather conditions. With our PPE, we have a range of realisations of the present-day climate, any of which could be the reality. We therefore ask if the observed climatology is consistent with the distribution of modelled climatologies (the NWSPPE). We do this by calculating the observed 20-year mean, and the modelled 20-year mean for each ensemble member. We then calculate the normalised bias using ( 1 ) to ask how many (PPE ensemble) standard deviations the observed 20-year mean is from the PPE

ensemble mean. If the (absolute) normalised bias is less than 2, the observations are within 2 standard deviations of the ensemble mean, and we consider the PPE to be consistent with the observations.

The approach works well for OSTIA SST and the ICES timeseries, where there are regular observations every year between 2000-2019. It is complicated by the sparsity of the EN4 data, where there may only be a single observation for a given grid box for a 20-year period. This means the observed "weather" is not separated from the observed climate – this must be taken into consideration (see section 3.2.1).

### 4.1 Tidal evaluation

We have evaluated the tides by producing co-tidal charts for leading constituents. The tidal harmonic analysis is done online by CO9, for consecutive 20-year periods. We show the ensemble mean for the files created for the 2000-2019 period (Figure 1). The ensemble mean phase (angle) is converted to its northward and eastward component before averaging. The $M_2$ component visually agrees well with O'Dea et al. (2012, 2017).

### 4.2 Sea Surface Temperature

We find the NWSPPE ensemble mean SST has a high spatial correlation (>0.95) with the OSTIA analysis across the domain, for all months, seasons, and in the annual mean. Figure 2 gives the absolute SST bias (NWSPPE ensemble mean minus OSTIA SST) for the annual mean, and the 4 seasons. When averaged over the domain, the absolute mean bias is less than 0.4 °C for all seasons (and months and in the annual mean). We also assess where OSTIA sits within the distribution of the NWSPPE – where the OSTIA SSTs are more than 2 standard deviations from the ensemble mean, we hatch out the biases in Figure 2.

In the annual mean we find that the OSTIA SSTs of most of the NWS are within the ensemble spread of the NWSPPE. 95% of the NWS have OSTIA SST biases less than 0.68°C. Around the edge of the NWS (as delineated by the green contour in Figure 2), the shelf slope current is slightly warmer than the ensemble and the Norwegian Trench is slightly cooler. In the wider oceanic parts the domain, the Norway Basin (63°N 5°W) is too cold, and a region along the western boundary extends to 15°W to too warm.

When looking across the seasons we see that OSTIA SSTs tend to be within the NWSPPE, albeit with less agreement than in the annual mean. While the winter and autumn OSTIA SSTs are largely within the NWSPPE, the summer OSTIA SSTs to the west of the UK are warmer than OSTIA, while the spring OSTIA SSTs in the northern North Sea, and around Scotland are cooler than the NWSPPE. Despite the regions of warm summer bias, and cool spring bias, overall we consider the NWSPPE SSTs to be broadly consistent with the OSTIA analysis.

### 4.3 Sub-surface Temperature and Salinity

The EN4 data processing procedure (outlined in section 3.2.1) produces sparsely sampled maps of the NWS, at monthly, seasonal, or annual granularity. The seasonal maps are given in the supplement (SST: Figure S4; SSS: Figure S5; NBT: Figure S6), however as they are still relatively sparse, they are hard to interpret quantitatively. We have therefore averaged these data across the evaluation regions (in green in e.g. Figure S3), and presented the regional means, and spread (2 time the spatial standard deviation) for the larger validation region mask in Figure 3. For completeness, we also include the regional summary for the Wakelin et al. (2012) region mask in Figure S3.

This shows that in most regions, and in most time of the year, the EN4 SST observations are within the PPE ensemble, although less so in the poorly sampled Norwegian Trench. The NBTs also tend to be within the PPE, and tends to agree with the SST in most regions and months. There is more disagreement with the EN4 SSS, with much greater spatial variability within the regions (reflecting the greater spatial variability in Figure S6). However, when averaging over the regions, the EN4 SSS is typically within the PPE ensemble.

#### 4.4 *in situ* temperature and salinity timeseries

We now evaluate the NWSPPE against the ICES climate timeseries. All ICES timeseries have absolute temperature biases less than 1.25 °C (NWS timeseries <0.85 °C) and absolute salinity biases less than 0.75 psu (Figure 4, Table S1). Most of the NWS ICES Temperature timeseries have a good overlap between the model and the observations mean, with NWS mooring being

within the NWSPPE. The Fair Isle current is modelled too cold, and outside the NWSPPE, although the absolute temperature bias is -0.83 °C. All off-shelf ICES temperature timeseries are too cold, and are outside the NWSPPE, reflecting the region of cold bias in the Norway Basin shown in the OSTIA analysis. Conversely, the all the off-shelf ICES salinity timeseries are within the NWSPPE, with very low absolute biases (<0.15psu), suggesting a good agreement. The NWS ICES Salinity timeseries are typically too salty, which may, in part, relate to the coastal locations of many of the sites, where there are

typically large horizontal salinity gradients. The Fair Isle Current is 0.42 psu too salty and is outside the NWSPPE (with a large relative bias) and little overlap of the interannual variability between the observations and model.

We can also use the ICES timeseries to assess the interannual variability. On the shelf, most locations have similar interannual variability between the observations and the model, with the observations sitting within the spread of the ensemble (having an absolute Relative Interannual Variability Bias less than 2). The Plymouth WCO E1 temperature, and Helgoland Roads salinity

are the exceptions, with the observations being much more variable than the model. Off the shelf, the observed salinity interannual variability is well represented in the NWSPPE, while the Faroe Shetland NAW temperature interannual variability is modelled as too variable compared to the observations.

Overall, the NWSPPE is in relatively good agreement with the ICES timeseries, giving further confidence in the NWSPPE simulations.

#### 4.5 Sea Surface Height

COx has a non-linear free surface, and so simulates the dynamic response of the sea level to the local dynamics of the model. This illuminates important aspects of the model behaviour. It is also a component of sea level but should not be used directly as a set of sea level projections (see section 15.5).

First, we compare the NWSPPE and the AVISO CLS2018 satellite MDT (Figure 5). There is a good agreement in the overall

pattern, with a large-scale NW/SE gradient and a similar range of values. On the NWS, the highest levels are in the German Bight, with lower values in the northern North Sea, north of the Dooley Current.

We then compare the simulated interannual SSH variability is to that of the C3S SLA product (Legeais et al., 2018). There is good spatial agreement between the PPE and C3S SLA product. In the open ocean there is greater sea level variability in deeper regions (e.g. the Rockall trough and the Icelandic basin). On the shelf the greatest variability in both the PPE and the

C3S SLA product is in the German Bight, and the lowest variability is in the Celtic Sea, and to the west of Ireland and adjacent the shelf break, this is more pronounced in the C3S SLA product.

There is good agreement in the trend and interannual variability of the tide gauges and the PPE (Figure 6). The interannual variability of the PPE Ensemble mean is averaged out, but the tide gauge variability tends to agree with the ensemble spread (ensemble mean ± 2 standard deviations). In most locations (apart from Dieppe and Bergen), there is a good agreement in the

trends. The tide gauge at Dieppe was reinstalled in 2009, and the levels appear to have changed (https://psmsl.org/data/obtaining/stations/474.php), with the later values in closer agreement with nearby tide gauges (La Havre and Boulogne). The tide gauge at Bergen may be affected by GIA.

#### 4.6 Volume transport through cross-sections

We now evaluate the NWSPPE residual circulation. While it is difficult directly measure residual currents (as they are typically

much smaller than the oscillatory tidal currents) there are several observation-based estimates of volume transport through cross sections across the NWS (Figure 7m, Table 4) that may be used for model evaluation (Fernand et al., 2006; Brown et al., 1999; Svendsen et al., 1991; Turrell et al., 1992; Danielssen et al., 1997; Prandle et al., 1996; Holt et al., 2001). We use the

same 12 cross-sections as used by Lowe et al. (2009) and Tinker et al. (2015). Here we report the modelled transport as an ensemble annual mean, and the seasonal range of the ensemble mean (minimum – maximum)

Overall there is a relatively good agreement between the model and the observations, with all transports being of the correct direction, and with the observations generally being within the modelled seasonal cycle. Furthermore, the transport through most cross-sections are similar, or show an improvement when compared to the UKCP09 (Lowe et al., 2009) and Minerva (Tinker et al., 2015) projections.

The North Sea inflow through the Fair Isle [6] is modelled to be too strong (0.32 Sv (0.20 - 0.46 Sv)) compared to observations
(0.2Sv), while the outflow through the Norwegian Trench [5; 1.56 Sv (1.31 - 1.81 Sv)] is modelled as being too weak (1.8Sv), although the observations are overlap with modelled seasonal cycle. The North Sea inflow between Shetland and the Norwegian Trench [4: 0.61 Sv (0.51 - 0.71 Sv)] and through the Dover Strait [10: 0.07 Sv (0.03 - 0.12 Sv)] are in good agreement with the observations (0.60 Sv and 0.1 Sv respectively).

There is a large inflow and outflow from the North Sea in the Skagerrak, which are nearly equal, with a very small residual
net outflow. This modelled inflow [1; 1.21 Sv (0.94 - 1.41 Sv)] is very close to the observations (1.0 Sv (0.5-1.5 Sv)). The Dooley Current [3; 0.25 Sv (0.23 - 0.30 Sv)], and the eastward flow to the north of the Dogger Bank [2; 0.08 Sv (0.03 - 0.11 Sv) ] are in good agreement with the observations (0.25 Sv (0.12 - 0.38 Sv) and 0.05 Sv respectively). The St George Outflow [9: 0.11 Sv (0.02 - 0.20 Sv)] is in good agreement with the observations (0.18 Sv). While the Irish Shelf [8: 0.12 Sv (-0.04 - 0.20 Sv)] and the Hebrides Shelf [7: 0.43 Sv (0.33 - 0.58 Sv)] are modelled as being too weak (0.25 Sv) and too strong
respectively (0.25 Sv), although where cross sections are not across channels, the transport is sensitive to the exact width of the cross-section.  The Shelf Current FS Current [12: 2.68 Sv (1.08 - 3.89 Sv)] is modelled as being too weak (5.50 Sv (4.00 - 7.00 Sv)), however, this could be due to the comparison of the modelled net transport and the observed north-westward transport. Overall, the transport comparisons suggest the NWSPPE simulates the configuration of the NWS residual transport well, and generally captures the magnitude of the transport pathways.


## 5    Results and Applications

We now explore some of the results of the NWSPPE climate projections – we refer the reader to Tinker et al. (2020) for a description of the results of the present day control simulation. We first describe the changes to the residual circulation, before considering the spatial patterns of the climate projections, and then turning to the temporal evolution with regional mean timeseries.

### 5.1    Changes in circulation

We assess the NWS residual circulation to give a background to the NWS changes. We consider how the barotropic residual currents change across the 21$^{st}$ century (2000-2019 and 2079-2098), and transport through predefined cross-sections (Figure S7-Figure S14, Table S2).

#### 5.1.1    Mean residual circulation

There is a general weakening of most of the NWS circulation over the 21st century (Figure 8i), but the general configuration of the residual circulation is relatively consistent across the ensemble and the 21$^{st}$ century (Figure 8d, h). A small region to the south and west of Ireland is an exception to both points, which we consider later. There is no evidence of a change in the large-scale configuration of the northern North Sea circulation as seen by Tinker et al. (2016), and analysed by Holt et al. (2018). However, to the west of Ireland (52°N 13°W), we do see slight change in configuration, related to a localised change in the slope current.

There is a general increase in the strength of the shelf slope current south of 54°N (Figure 8i, Figure S7), and then a decrease between 54°N and 60°N 6°W (Figure 8i, Figure S8). To the north of the North Sea (north west of 60°N 6°W) there is a substantial strengthening of the slope current (Figure S9). The portion of the shelf slope current that follows the isobaths turning into the Norwegian Trench slightly increases from ~0.8Sv to 1.2Sv over the 21$^{st}$ century (Figure S10b), however as the strength the slope current increases, this represents a relative decrease from 35% to 10% (Figure S10b cf. Figure S10a). Similarly, the NT outflow is similar (Figure S11c), but represents a much smaller proportion of the total downstream net transport (40% to 20%, Figure S11c c.f. Figure. S10b). This is consistent with the finding so Holt et al. (2018), who showed that an increase in haline stratification in this region increases the Rossby Radius, and so makes it harder for the current to follow the bathymetry. When looking across the ensemble, all members show a strengthening of the most northern sections of these currents (north of 61°N, 3°W), but there is a general decrease of the currents further south, in main part of the Norwegian Trench (Figure S15 g, h).

There is an overall weakening of the northern North Sea Inflow (NNSI), the Norwegian Trench outflow (Figure S15 h, i), and the wider North Sea circulation. The current though the Pentland Firth (58°45'N 3°6W, Figure S12a) and the Fair Isle current (59°30'N 1°45'W 0.3-0.2Sv, Figure S12b) reduce by about 15% and 40% respectively. The inflow to the west of Shetland increases slightly (14%, Figure S12c), but this may reflect an increasing flow into the Norwegian Trench. The Dooley Current also shows a 20% reduction in the ensemble mean (from 0.256 Sv to 0.202 Sv, Figure S14a, Figure S15 j, k, l). Further south, the North Sea inflow though the English Channel decreases by 40% (from 0.07 to 0.04Sv Figure S13a), and this is reflected in the lower downstream currents though the Southern Bight (Figure S13b) towards the Skagerrak. The Skagerrak recirculation weakens from 1.2 Sv to 1.0 Sv (Figure S14bc), although this reduces to 0.7 Sv for one ensemble member.

There is a tendency for an increase in the residual current magnitude to the southwest of the UK and Ireland, which leads to a slight change in circulation configuration: 80% of this region shows an increase in the residual current magnitudes, whereas 60% of the rest of the NWS shows a weakening. In the present day, there is a clockwise coastal current around Ireland, and a northward slope current that follows the shelf break from the Goban Spur (49°30'N 12°W), around the Porcupine Seabight (50.5°N 12°W) and Porcupine Bank (52°N 14°W), before following the shelf break north around Ireland. Between the coastal current and the shelf slope current, there is a weak, disorganised southward contra-current. In the future, the coastal current weakens, and the slope current strengthens, which helps strengthening this part of the slope current. The southward contra-

current strengthens substantially by the end of the 21$^{st}$ century, and flows south eastward across the Celtic Sea, increasing the mean current magnitude, and reducing its (relative) variability.

### 5.1.2 Residual current variability

We now consider the residual current uncertainty ellipse approach of Tinker et al. (2022). For each grid box, we can fit ellipses around the bivariate normally distributed residual currents, to encapsulate 95% of the data – we can then use the properties of these ellipses, and how they vary over time and the ensemble, to describe the residual current similarity. To aid the reader in the interpretation of residual current uncertainty ellipses, we have included examples focusing on the regions with the greatest changes (Figure S15, S16). Furthermore, Tinker et al. (2022) include explanatory figures in their appendices (reproduced in our Figure S2), and released a python toolbox to undertake the analysis.

Over most of the shelf, there are only small percentage changes in variability of the residual currents (e.g. Figure S15, Figure S16), as given by the uncertainty ellipse area (Figure 8b, f, j). While there are no NWS regions with large reduction in variability, there are substantial regions with large variability increases (e.g. Figure S16d, e, g) – these include the NNSI, to the west of Ireland, the slope current and the Norwegian Trench. Where this increase in variability is coupled by a decrease in the mean current magnitude, we see a large decrease in the significance of the residual circulation (such as in the NNSI, the Dooley Current and the Norwegian Trench). Where the increase in variability is matched by an increase in the mean current magnitude, the significance of the current tends to increase (i.e. the currents to the west of Ireland), but this is not always the case (e.g. the slope current to the west of Shetland).

The residual current ellipses allow us to compare the variability of the residual current to its mean, and quantify how many standard deviations the mean is from zero– when this is greater than 2.45, the mean current doesn't cross zero more than 95% of the time. As we have shown the residual circulation magnitude is typically reducing, and the variability is increasing, the number of standard deviations is also generally decreasing, and this is reflected in the amount of NWS residual circulation that is significant (Figure 8d, h), which we use to show the NWS residual circulation configuration.

We can also use the residual current ellipses to quantify similarity between two periods, two ensemble members, or across an ensemble. As each residual current ellipses represents a bivariate normal distribution, which integrates to 1, we can integrate the volume under 2 or more bivariate normal distributions to show how similar they are (illustrated in Figure S2i), where 1 is perfect agreement, which decrease towards zero as the mean, variance or covariance of the distributions differ. This allows us to quantify the residual circulation ensemble spread in the early and late 21st century, and how the different the residual circulation is between the early and late 21st century.

In addition to the general reduction in the ensemble mean residual current magnitude (Figure 8i) and increase in residual current variability, there is also increase in the ensemble spread and diversity. This is reflected in the decrease in ellipse ensemble overlap (Figure 8c, g, k), which is a measure of how similar the pointwise residual current distributions are across the ensemble. When this is high for a given point (near the maximum value of 1), the residual current distribution (in terms of U and V mean and variance, and their covariance) are similar across the ensemble. In the early 21$^{st}$ century (Figure 8c), over half of the NWS has and ens_OVL > 0.65, whereas by the late 21st century there has been a general reduction across most of the NWS with the ens_OVL < 0.53 (Figure 8g), but with a particular reduction in the Northern North Sea, Norwegian Trench, and to the south and west of Ireland (Figure 8k), where ens_OVL<0.3. Conversely, there is a slight increase in ens_OVL in the Irish Sea, suggesting a convergence in the ensemble behaviour.

ens_OVL quantifies the spread of the ensemble in early and late century (Figure 8c, g), and how much it has changed (Figure 8k). It is possible for ensemble spread to remain the same (ensemble OVL doesn't change over the 21$^{st}$ century), but the future being nonetheless very different from the present day. To quantify how much the circulation changes over the 21$^{st}$ century in each ensemble member, we calculate the ellipse overlap between the early and late 21$^{st}$ century, and report the ensemble mean in Figure 8l. This shows that over most of the NWS, there is little change in the circulation, whereas the northern North Sea, NT and to the west of Ireland there are regions of substantial change. While this is clear in the ensemble mean, some ensemble

members show little change in their circulation, and others showing a significant change reflecting changes in their current magnitudes and variances.

We have produced some exemplar uncertainty ellipses (for example locations) to help visualise some of these changes. The present-day ellipses tend the be very similar across ensemble, which is reflected by ens_OVL>0.6 in most locations. The Irish Sea (e.g. Figure S16a,b), and the outer most point to the west of Ireland (e.g. Figure S16c) being the exceptions, which is captured in Figure 8c, where the ensemble tends to converge. In the northern North Sea (Figure S15) and to the south and west of Ireland (Figure S16) the uncertainty ellipses are larger in the future, and there is much less agreement across the ensemble, with less overlap, captured by a much lower future ens_OVL (e.g. Figure S15 g). Other have a similar level of ensemble agreement (e.g. Figure S15 d), and so less change in the ens_OVL over the 21st century. Some locations show a relatively little change in the ensemble spread (similar values of the ens_OVL over the 21st century), but a large difference in the residual currents (e.g. Figure S15 e) reflecting little overlap between the present-day and the future, which is captured by a low OVL.

### 5.2    Projected changes at the end of the 21$^{st}$ century

The NWS SST (Figure 9; Table 5; Figure S17) rises by 3.11 °C (±0.98 °C) by the end of the century (2079-2098 compared to 2000-2019), when averaged across the shelf (ensemble mean ± 2 ensemble standard deviations). There is a seasonal cycle in the warming, with greater warming in the summer (3.57 °C ±1.09 °C) and autumn (3.73 °C ±1.07 °C) than in the winter (2.72 °C ±0.97 °C) and spring (2.43 °C ±1.01 °C). This is consistent with the wider UKCP18 projections (Murphy et al., 2018) which exhibit greater warming in the summer than the winter, which increases the amplitude of the temperature seasonal cycle, and is consistent with their key findings of projected hotter drier summers, and warmer wetter winters (Lowe et al., 2019). There is a region of reduced warming to the NW of the NWS (e.g., 60°N 15°W) which is more visible in winter (it may be obscured by summer stratification) and may influence the NWS. A likely cause of this is the North Atlantic Warming Hole (Menary and Wood, 2018), and a reduction in the Atlantic Meridional Overturning Circulation (AMOC), transporting less heat from the tropical Atlantic. This will be considered later in the discussion.

On the NWS, the southern and central North Sea show considerable warming, particularly in summer and autumn, with typically greater than 3.5°C warming. The greatest NWS warming is in the Dover strait in autumn, where the ensemble mean SST warms by up to 5°C. There is an adjacent region of reduced warming in the centre of the Southern Bight (~52°N 3°E) in all seasons, relating to an apparent weakening of the warm plume of water flowing from the English Channel into the Southern North Sea (Figure S13a).

The Near Bottom Temperatures exhibit a more modest rise than the SST (Figure 10; Table 6; Figure S18), with an annual mean NWS NBT warming of 2.49 °C (±0.94 °C). This largely reflects a reduced warming under stratified regions. In the SST, we saw a greater warming in summer than winter. As the spring initialisation of stratification isolates bottom water from the atmosphere, NBTs remain close to the winter spring temperatures under stratification. Therefore, the weaker winter SST warming will reduce the summer NBT warming in stratified regions. This is illustrated in the difference in the southern and northern North Sea NBT summer warming, 3.55 °C (±1.00 °C) and 2.07 °C (±1.00 °C) respectively. The southern North Sea summer NBT rises to a similar level to that of the SSTs in the same region and season (3.78 °C (±1.04 °C)), while the northern North Sea summer NBT warming is closer to the winter SSTs (2.71 °C (±0.99 °C)) than the summer SSTs (3.61 °C (±1.20 °C)) in the same region.

It is also interesting to note that the northern North Sea winter (and autumn) NBTs exhibit greater warming than the summer. The northern North Sea is the main place where the North Sea connects to North Atlantic, and so most affected by the reduced North Atlantic warming. In the winter, the northern North Sea may also be being warmed by the atmosphere, while when stratified, this would not be possible. This would tend to lead to a greater northern North Sea NBT rise in winter than in summer, as simulated.

The fact that the stratified SSTs tend to rise more than the NBTs, leads to an increase in the difference between the surface and bed temperatures (DFT, Figure 11; Table 7; Figure S19), which reflects an increase in the stratification magnitude.

Stratification can be quantified by the amount of energy needed to mix the stratified water column, which is formalised with the Potential Energy Anomaly (PEA) defined in ( 2 ). There is a general strengthening in the magnitude of summer stratification, although there is little change in its spatial extent (Figure S21, Table S4). PEA can be broken down into a thermal and haline component (PEAT and PEAS, Table S5 and S6 respectively). Most of the increase in PEA is associated with an increase in thermal stratification (PEAT), although in places this is modified by changes in the haline stratification (PEAS). Around the shelf edge between the Atlantic and the NWS, there is a strengthening in the haline stratification throughout the year. This balances a co-located band of reduced thermal stratification (PEAT) in the winter and spring. This haline stratification extends into the western half of the Celtic Sea, leading a substantial increase of local winter PEA. Finally, there is a reduction of haline stratification adjacent to the Norwegian trench, which at its peak in summer balances any increase in PEAT, leading to no increase in PEA at ~60°N, 2.5°E.

The changes we see to the NWS temperatures also reflect a change in the seasonal cycle. The SST seasonal cycle subtly changes between the early and late 21$^{st}$ century (when averaged over the NWS), with a slight preferential warming between late June and early December, which leads to the greatest surface warming being in autumn. This is consistent with the broader behaviour of the PPE, and (a subset of) CMIP6 models (Cotterill et al., 2023), which shows an atmospheric circulation change that increases the frequency of drier summer-type regimes, and a decrease in stormy winter types in autumn. There is little change in the NBT seasonal cycle between the early and late 21$^{st}$ century (when averaged over the NWS). As the NWS is still mixed in the winter throughout the 21$^{st}$ century, the change in the seasonal cycle of DFT (the difference between the surface and bed temperatures) is dramatic. From January to May, DFT (averaged over the NWS) does not change over the 21$^{st}$ century, while at its peak (in mid-August) NWS mean DFT increases by 1.5 °C. These seasonal maximum increase in NWS DFT over the 21$^{st}$ century, is approximately a month later than the present-day NWS DFT seasonal peak. Therefore, there is little difference to the DFT in the early summer, and a much greater change in the autumn, which effectively extends the seasonal stratification.

We assess the timing of onset and break down of the stratification, and its duration, using the method described in section 2.9. We find that average NWS onset is typically 4.16 (± 2.31) days earlier when averaged over the ensemble (± 1 ensemble standard deviation), and breakdown 18.91 (±3.50) days later, leading to an increase in stratification of 23.07 (± 5.06) days. This is a greater increase in stratification duration than reported by Holt et al. (2010), who found that the onset of stratification was 1.29 days earlier, the break down was 3 days later, giving a much smaller increase of duration of 6.36 days when averaged over the NWS (Figure 12).

There is a substantial freshening across the domain, which is greatest in the open ocean to the west of the domain (Figure 13; Table 8; Figure S20). The NWS also shows substantial freshening, with -1.01 psu (±0.93 psu) when averaged over the shelf, and little spatial or seasonal variations. There is also substantial increase in the ensemble variance, and a much smaller increase in inter annual variability. This increase in salinity appears to be oceanic in original, and predominantly entering the NWS via the NNSI. Regions that are less connected to the NNSI show a slightly reduced freshening. For example, there is slightly less freshening in the Celtic Sea, English Channel, and southern North Sea, which in part may reflect the use of climatological river forcings, but also the more southerly origin of their Atlantic water. There is also a reduced freshening in the Norwegian Trench, which is likely related to the use of climatological Baltic LBCs.

When looking at surface salinity of the wider North Atlantic in the HadGEM3-GC3.05 PPE we start to see a meridional freshening dipole, with a general freshening north of ~40°N (20°N in the Canary Current), with an increase in salinity to the south (Figure S23). This pattern strengthens throughout the 21st century and across the ensemble. The greatest freshening is from Gulf of St Lawerance and the Labrador Sea, and along the pathway of the Gulf Stream and North Atlantic drift, towards the NWS. There are regions of increasing salinity north of the dipole, including Hudson Bay and the in the Norway Basin (63°N 5°W). The same pattern and temporal evolution occurs in all ensemble members, although the timing and strength of the pattern varies. The substantial freshening North Atlantic freshening across the ensemble is highly correlated with the NWS

freshening (Figure S24a) and is of a similar strength (Figure S24b), suggesting the NWS response is driven by the Atlantic, and that it also explains the increased in ensemble spread.

### 5.3    Temporal and ensemble evolution over the 21ˢᵗ century

We now assess the regional means timeseries from the NWSPPE, to illuminate the NWS's temporal evolution. We also compare to the Copernicus Marine Reanalysis (RAN) to provide near present-day information for context (Figure 14, Figure 15). We note that our use of climatological Baltic boundary conditions may explain the behaviour and the reduced interannual variability in the Skagerrak/Kattegat and Norwegian Trench regions.

The SST regional mean timeseries are in relatively good agreement with the absolute values of the RAN (Figure 14). There is

a near linear trend in the ensemble mean SST. Each individual ensemble member also has near linear trends (typically $r > 0.9$), but have much larger interannual variability, which is averaged out in the ensemble mean (Figure S25). While the ensemble mean interannual variability appears much lower than that of RAN, there is relatively good agreement when comparing to individual ensemble members (Figure S25), particularly on the shelf. The RAN timeseries appear to have a different trend to the ensemble but is consistent with the scale of the variability of individual ensemble members (not shown). The SST ensemble

spread remains relatively constant in some regions, and slightly widens in other regions. The SST interannual variability is also relatively constant throughout the projection (Figure S25). The annual mean NBT data show similar behaviour to that of the SST, so is not included in this paper.

The SSS regional mean timeseries tend to be saltier than the RAN reanalysis, which lies outside the ensemble spread in several regions. This is particularly true in the Skagerrak and (downstream) in the Norwegian Trench where the NWEPPE is much

fresher than the RAN. This difference partly reflects the differences in the Baltic lateral boundary conditions between the RAN and the NWSPPE, and the complexity of modelling this region, especially when using Baltic and Atlantic lateral boundary conditions from a different ocean model (as the RAN and NWSPPE both do). Moreover, the RAN surface salinity to too salty in the Norwegian Trench (the Skagerrak is outside the RAN evaluation region) (Renshaw et al., 2019).

There is good agreement in SSS interannual variability with the reanalysis in the North Sea and English Channel, while the

PPE has greater interannual variability in other shelf regions. There is also a substantial increase in SSS interannual variability between 1990-2019 and 2069-2098 in almost all regions of the NWS (Figure S26) – absent in SST (Figure S25)

The NWS SSS behaves very differently to that of SST (Figure 14). Rather than near linear evolution seen in SST (Figure S27), SSS has an increasing freshening, and divergence of the ensemble. For most ensemble members, and shelf regions the correlation tends to improve when fitting different straight lines for the beginning and end parts of the timeseries (Figure S28).

While most ensemble member have similar initial SSS gradients, there is a greater range of gradients in the second half of the century, and a range of timings of the change in slope – both of which contribute to the divergence of the ensemble. This is consistent with the arrival and strength of the freshening signal from the NAE.

## 6 Discussion

We have downscaled a PPE of HadGEM3-GC3.05 to give a set of climate projections for the NWS. The PPE was developed for the UKCP18 climate projections to explore the range of an important source of uncertainty in the climate projections and was run under the RCP8.5 scenario. We have downscaled 12 members of this PPE with the shelf seas model NEMO CO9 (NWSPPE) from 1990-2098 (plus a discarded 10-year spin up period). We have driven CO9 with atmospheric fluxes and ocean lateral boundary conditions interpolated directly from HadGEM3-GC3.05 (without any bias correction). We have used

a climatology for the river forcings and for the exchange with the Baltic Sea. We have undertaken extensive model evaluation (for SST, subsurface T and S, SSH and tides) and explored the first order change the NWS. These simulations represent the state-of-the-art for NWS marine projections.

This set of projections is closely related to, and compatible with, the 200-year present day control simulation of Tinker et al. (2020) (PDCtrl). We have combined both datasets into a single data release, which is also described here.

Our NWSPPE is a direct update to the Minerva Projections released in 2016. The same general approach is taken, downscaling the PPE run for UKCP, with a regional shelf seas model, to give a similar set of physical projections. Minerva downscaled the UKCP09 PPE which was based on HadCM3 (Gordon et al., 2000; Pope et al., 2000). HadCM3 was a relatively old model at the time (being a CMIP3 generation model) but was used as its computational efficiency allowed it to be run many times in the ensemble. Here, we use the latest Met Office Hadley Centre CMIP6 generation model HadGEM3-GC3.05. Minerva was

based on POLCOMS (Holt and James, 2001; Holt et al., 2001), a 12km resolution shelf seas model which had already been replaced at the Met Office in favour of NEMO Coastal Ocean model. Here we used NEMO version 4.04, an even more recent version than is used operationally at the Met Office. As the HadCM3 atmosphere has a horizonal resolution of 2.5° latitude by 3.75° longitude, a regional atmosphere climate model (HadRM3, ~25km horizontal resolution, Jones et al., 2004) was used to downscale the HadCM3 atmosphere to provide atmospheric forcings to POLCOMS. As the HadRM3 model domain didn't

cover the full POLCOMS model, the southwest corner had to be cut off. HadGEM3-GC3.05 is of a much higher resolution (~60km) than HadCM3, and so the atmosphere did not need to be downscaled to produce the atmospheric forcing for NEMO. Our projections are run under the RCP8.5 scenario, while the Minerva projections were based on the SRES A1B scenario (Nakicenovic et al., 2000). There is a greater warming in RCP8.5 compared to SRES A1B, with the UK region projected to warm by 3.9°C in 2080-2099 relative to 1981-2000 under RCP8.5 compared to 2.7°C under SRES A1B (Lowe et al., 2019)

(the 50[th] percentile of the UKCP18 terrestrial probabilistic projections). Furthermore, all HadGEM3-GC3.05 PPE members have a relatively high climate sensitivity (Rostron et al., 2020), similar to HadGEM3-GC3.1 and greater than that of HadCM3 (Andrews et al., 2019), which was used in the Minerva projections. This greater warming is reflected in our new projections. We project an SST rise of 3.11 °C ($\pm 2\sigma = 0.98$ °C), and an SSS freshening of $-1.01$ psu ($\pm 2\sigma = 0.93$ psu) for 2079-2098 relative to 2000-2019. Tinker et al. (2016) projected shelf and annual mean SST rise of 2.90 °C ($\pm 2\sigma = 0.82$ °C), and an SSS freshening

of -0.41 psu ($\pm 2\sigma = 0.47$ psu) for 2069–2098 relative to 1960–1989. It is difficult to directly compare these changes due to the changes in baseline however, we note that we project a greater change over a shorter period.

When we plot the regional mean timeseries for both sets of projection, as anomalies relative to a common baseline, the difference is more dramatic (Figure 16). While both Minerva and the NWSPPE show a near-linear SST rise and similar ensemble divergence, there is a notably greater rate of warming in our new projections. When looking at SSS, the difference

is striking. While the Minerva projections showed a substantial reduction in salinity, we show a much greater freshening. In the Minerva projections, the ensemble mean salinity is mainly driven by a steplike change in a few ensemble members with particularly high climate sensitivity. Almost all our ensemble members show a substantial increase in the rate for freshening (Figure S10) driving a much greater ensemble mean change. Figure 13 suggests the salinity changes are driven by changes in the salinity in the open ocean adjacent the NWS.

Over most of the NWS, we show a similar pattern of SST change to Minerva (Tinker et al., 2016) with the southern North Sea showing the greatest warming, and the SST fingerprint of a reducing shelf break current to the north and west of Scotland

(Figure 9). When we look across the whole model domain, we see important differences in the SST change spatial pattern (compared to Minerva). The most prominent difference is the area of near zero winter (and spring) warming in the open ocean to the North West of the domain in NWSPPE, which is absent in Tinker et al. (2016). This is consistent with a slowdown in the AMOC (Table S9) and the associated North Atlantic warming hole (Drijfhout et al., 2012). Figure S29 (maps of the correlation between AMOC slow down and NWS change) show a concurrent region of highly correlated SSTs to the south of Iceland, and a pattern of highly correlated NBTs in the deep waters to the west of the NWS. These correlations reflect that the ensemble members with the weakest reduction in AMOC having the greatest warming locally, and also that the ensemble members with the strongest reduction in AMOC actually have local cooling. The tongue of highly correlated SSTs in the Icelandic Basin is a branch of the North Atlantic Current (Perez et al., 2018), while the pattern of the highly correlated NBTs reflects the complex deep water pathways and bathymetry of this region. There is a competition between the warming associated with a rising global mean temperature, and a cooling associate with a reduced AMOC (Drijfhout, 2015) - over most of the domain the warming dominates, whereas in this small region the cooling dominates, resulting in this off shelf region of negligible projected SST warming.

We see a substantial lengthening of the seasonal stratification, which starts 4 days earlier and finishes 19 days later. This is a much greater lengthening than seen by Holt et al. (2010). Stratification is an important feature of the NWS oceanography and has a particularly important impact on the NWS ecosystem. The formation of stratification in spring confines phytoplankton to the sun-lit surface mixed layer of the NWS, providing it with enough light to initialises the spring Phytoplankton bloom, which drives the NWS ecosystem. As the summer progresses the nutrients in the upper layer are used up, and the autumnal breakdown of stratification allows mixing of subsurface nutrients, leading to a secondary autumn bloom. Most phytoplankton growth happens in the spring bloom, especially diatoms, which dominate in the seasonally stratified zone. There is a much smaller dinoflagellate bloom that happens in late summer/autumn (Van Leeuwen et al., 2015) – our longer projected stratification may suppress mixing and so reduce the nutrient supply to these phytoplankton and restrict their growth; alternatively it might keep them in the sunlit zone for longer and increase growth, as long as there is a nutrient supply, e.g. from short lived mixing events. The summer bloom in the stratified North Sea has been observed to be increasing in duration, by extending further into autumn (Silva et al., 2021), although the exact physical mechanism were not identified. Any change in bloom duration or strength would affect primary production and so knock on to the rest of the ecosystem. The importance of the spring bloom relative to the autumn bloom, may mean that slightly earlier initialisation may be more impactful that then substantially delayed autumnal break down, as there is potential more growth there.

We have shown a general weakening of the residual circulation of the NWS, over the 21st century, although its large-scale configuration remains consistent. As the residual currents are important for the transport of heat, salt and matter, their weakening can lead to changes in the mean state. For example, we see a region of reduced warming in the Southern Bight (e.g. Figure 9). There is eastward transport bringing warm water from the English Channel into the Southern Bight of the North Sea, which leads to a plume of warmer, saltier water (e.g. Tinker et al., 2022). This eastward volume transport decreases into the future (Figure S13a), as does the associated heat transport. This weakens the Southern Bight warm plume, so reduces the localised warming. Tinker et al. (2016) found a substantial change in the circulation of the northern North Sea and three of their most high climate sensitivity ensemble members, with a reversal in the flow. Holt et al. (2018) undertook an extensive analytical study to replicate this, and to understand the mechanisms behind the changes. They reported a change in the North Sea circulation with the shelf slope current bypassing the Norwegian Trench and so reducing the exchange of the North Sea with the North Atlantic, leading to a more estuarine like circulation for the North Sea. One of the mechanisms they found was an increase in the haline stratification, increases the baroclinic Rossby radius, making it harder for the shelf slope current to turn the tight corner into the Norwegian trench. While we do find the NWSPPE slope current tends to bypass the Norwegian Trench, we do not see the large-scale circulation configuration change in this region. We note that both Tinker et al. (2016) and Holt et al. (2018) found this while downscaling relatively coarse ocean models – Tinker et al. (2016) downscaled the 1.25°

HadCM3 ocean, and Holt et al. (2018) downscaled the 1° ORCA1 configuration of the NEMO model. Holt et al. (2018) also downscaled ¼° ORCA25 configuration of NEMO found this circulation change to be weaker and that the configuration did not flip to a new state. Our NWSPPE also downscale ORCA025 (the ocean component of HadGEM3-GC3.05) so perhaps this northern North Sea configuration change, in part, depends on a coarser driving ocean model.

The wider North Atlantic shows a substantial freshening in the HadGEM3-GC3.05 PPE, which is consistent with the changes
we see on the NWSPPE (Figure S23). There are a number of mechanisms that could be contributing to this large-scale change, including changes to the hydrological cycle Rhein et al 2013, change in stratification (Zika et al., 2018), Greenland Ice Sheet loss (Ruan et al., 2019), and AMOC slow down (Zhu and Liu, 2020). The Atlantic Freshening is consistent with the changes we see on the NWS. We have also shown that there is a slight reduction to the North Sea inflow (Figure S12), which will also lead to a reduction of the NWS salinity. However, we do not see the North Sea circulation configuration change seen by Holt
et al. (2018), which effectively isolates the North Sea from the Atlantic, with the North Sea becoming more estuarine, so while our slight reduction in the modelled NNSI will play a role in the NWS freshening, the main driver is likely to be the freshening of the North Atlantic.

As our NWSPPE is consistent to the PDCrtl, the two dataset can be used together. By using the projected climate evolution from NWSPPE, with the estimated unforced variability of the PDCtrl, it is possible, for example, to estimate the Time of
Emergence of the climate signal from that climate variability. Most GCM climate projections are started from a convenient point within a climate spin-up simulation, when the modelled climate is stable (or its drift is acceptable), and time counter started from the year 1850 to reflect the start of the "historical" forcing period. Climate projections do not use data assimilation (or a similar process) to constrain the present-day climate to observed state of reality as used in seasonal or decadal forecasting systems (e.g. MacLachlan et al., 2014; Hermanson et al., 2022). This means that the present day of the climate simulation
doesn't match reality, in terms of the phase of the various climate modes and variability. This does not matter in the distant future (late century) where we can be confident that the climate change signal dominates over the possible range of present-day variability, but potentially limits the utility of climate projections in the near future due to the potentially confounding influence of internal variability (e.g. IPCC, 2021b). By estimating the Time of the climate signal Emergence (ToE), we can provide a lower bound of how soon you can start using uninitialised climate projections.

We have applied the method of Lyu et al., (2014) (described in section 2.10) to give an estimate of the Time of Emergence of the SST and SSS climate signal from the present day variability (Figure S30), and show the ensemble median and 16th-84th percentile range, masking out points where climate emerges in less than 84% of the ensemble (10 of the 12 members). We find the SST climate signal emerges from variability across the shelf with a relatively low (16th - 84th) range across the ensemble, whereas the SSS signal only emerges the climate variability for 42% of the NWS (with emergence in greater than 10 ensemble
members). Furthermore, the SSS climate signal emerges later than SST, and with greater ensemble spread. To compare the ToE of NWS SST and SSS, we take the NWS means of the ToE median (and range) for the grid boxes ToE emerges, and we find that the SST typically emerges in 2034 (with an 8-year range) while SSS emerges in 2046 with a 33-year range. This can guide where and when to use the NWSPPE for near future temperature and salinity projections.

UK policy makers are interested in warming levels, such as the projected conditions under a 2°C and 4°C world. These are a
useful policy tool to describe the climatic conditions when the when global mean temperatures rise to either 2°C or 4°C above pre-industrial levels. Warming levels have been used in the recent IPCC 4th assessment report (IPCC, 2021b) and in the UK's third Climate Change Risk Assessment (CCRA; Betts et al., 2021), and have been calculated for the UKCP18 HadGEM3-GC3.05 PPE. Gohar et al. (2018) use a time shifting methodology (Herger et al., 2015; Schleussner et al., 2016) to derive a pair of UKCP18 based 2°C and 4°C products. They use the HadGEM3-GC3.05 PPE (among other simulations) to find the
years where the global mean temperature crosses the 2°C and 4°C threshold relative to the preindustrial conditions (considered to be 1851-1900). As we use the same PPE, the same years can be used as the basis of 2°C and 4°C projections of the NWS, however, as our simulations do not include the preindustrial period, some care must be taken when trying to apply and interpret

this methodology to the NWSPPE. Their table 2 shows the timing of a centred 20-mean passing 2°C and 4°C of global mean temperature rise since the preindustrial period for the anonymised models in their study (adapted in our Table 9). For example, their model 1 (our r001i1p00000) passes the 2°C threshold in 2030, and the 4°C threshold in 2063. While we can calculate the NWS conditions for r001i1p00000 in a 20-year period centred on 2030 (2021-2040), and can iterate through our ensemble, we cannot show how much the NWS has change since the pre-industrial period, as our simulations start in 1980. Therefore, any difference between the NWS and the global mean that have occurred since the pre-industrial period are not captured. This is a common problem for regional studies. One approach is to say how different the NWS would be from the early century, under 2°C and 4°C warming levels. For this, for each ensemble member, we can use the time-slices in Table 9 and subtract the 2000-2019 period, and then average across the ensemble. Another approach is to add the global warming between 1850-1899 and 2000-2019 as calculated from the HadCRUT5 dataset (Morice et al., 2012), which equal 0.987°C. As the NWSPPE does not begin in the pre-industrial period, the exact timings of the 2°C and 4°C warming levels will still remain uncertain, especially as the HadGEM3-GC3.05 PPE was not designed to span all sources of uncertainty. The exact methodology used to choose the period of the 2°C and 4°C warming levels may impact the averaging periods, but the methodology proposed here should give a first order estimate.

These climate projections are a one-way forced downscaling of a GCM (HadGEM3-GC3.05) using climatological Baltic exchange, and rivers. This approach is based on two main assumptions. Firstly, that the driving GCM can simulate the large-scale climate and how it changes, and secondly, the downscaling approach allows improved resolution and additional physical processes important to the shelf seas, without becoming inconsistent with the climate of GCM.

HadGEM3-GC3.05 has been extensively evaluated at the global and regional scale (Williams et al., 2018), and for the parameters relevant for the NWS (Tinker et al., 2020). Dynamically downscaling GCMs for shelf seas regions is a well-established approach (Gröger et al., 2013; Mathis and Pohlmann, 2014; Tinker et al., 2016; Holt et al., 2010; Olbert et al., 2012). We follow the methodology of Tinker et al., (2020) using NEMO COx on AMM7, a very well-established model and domain. NEMO COx is used operationally at the Met Office for 6-day forecasts (O'Dea et al., 2017; Tonani et al., 2019), and in their Reanalysis (Renshaw et al., 2019). Furthermore, it has been used as a research model on a number of time scales (including seasonal predictions (Tinker and Hermanson, 2021), centennial climate projections (Hermans et al., 2020b) and for the present-day unforced climate variability (Tinker et al., 2020)). The choice of domain is important for downscaling – too big, and the interior of the model will diverge from the climate of the parent model, and too small and the boundary conditions will dominate the interior of the model. Furthermore, the geography of the domain is important, in the case of the NWS shelf edge processes are complex and should be either excluded (e.g. Olbert et al., 2012), or be far enough from the LBCs to behave independently from them. As NEMO COx run on the AMM7 domain has been extensively used and evaluated, we are convinced that our approach is fit for our purposes, although we do note some limitations.

The exchange with the Baltic is complex and helps controls the dynamics of the Norwegian Trench, which is "downstream" of most of the NWS (including the UK). We use a climatology for this, which is a common approach (e.g. Tinker et al., 2015; Holt et al., 2010). The climatological Balic may have an impact in the Skagerrak and along the coast of Norway, where the mean salinity (its variability) may be affected. This, with the use of climatological rivers, is an important limitation for understanding the NWS water cycle and salinity response as the climatological rivers and Baltic LBCs don't "see" any hydrological changes simulated in the parent GCM. This approach could be improved upon. A consistent set of transient Baltic Sea climate projections could give a trend to the climatology, or even a set of transient boundary conditions. It may be possible to develop a Baltic Sea box model to simulate the exchange (e.g. Stigebrandt, 1987). The best approach would be to run both NWS and Baltic projections together. Care must be taken with the mean sea level of the Atlantic and Baltic LBCs, which can be inconsistent when taken from different models. We also use a river climatology, which is also a common approach, and is used operationally for the Met Office NWS forecasts (Tonani et al., 2019). HadGEM3-GC3.05 has online river routing, sousing this river output directly may be a better option but would need assessment. The estimated Baltic outflow (668 km$^3$yr$^{-1}$ ±32

km$^3$ yr$^{-1}$ for the period 2002-2021, Boulahia et al., 2022) has relatively low interannual variability and appears relatively small compared to the variability of the North Sea riverine inflow (NOSCCA, 2016) – this may suggest that improving the representation of the riverine climate response is a greater priority.

We use the HadGEM3-GC3.05 atmosphere directly. This has a spatial resolution of about 70km, which is relatively low. Future projections may benefit from higher resolution atmosphere, however, there may be little benefit in increasing the atmospheric resolution until you get to convection permitting resolutions (~2 km) (e.g. Vautard et al., 2013; Kotlarski et al., 2014). This could either come from the using a higher resolution GCM, output from a regional atmospheric climate model to downscale the atmosphere, or by moving to a regional coupled climate model. Furthermore, other studies bias correct the GCM model output before use (Mathis et al., 2013). This could be explored for uncoupled future studies.

We use a COx on the AMM7 domain, which has a 7 km horizontal resolution, with 50 terrain following vertical levels. AMM7 is an eddy permitting model, but as the internal Rossby radius is ~4km on the NWS (Holt and Proctor, 2008), a model resolution greater than this would allow the simulation of a much richer eddy field, and improved representation of shelf exchange processes. AMM15 is a 1.5 km resolution model covering a similar region to AMM7. AMM15 will replace the use of AMM7 for many purposes. AMM15 is eddy resolving and shows an improved representation of the mean state across the domain, compared to AMM7 (Graham et al., 2018). As well as having a higher resolution, it requires a smaller timestep, so is much more expensive to run computationally than AMM7. Currently both AMM7 and AMM15 are run operationally at the Met Office to provide operational 6-day forecasts: AMM15 is coupled to a wave model; while AMM7 is coupled to the expensive biogeochemistry model as AMM15 to currently too expensive to run with biogeochemistry. AMM15 may be too expensive to use as the basis of ensemble climate projections. However, it may be useful to run companion runs, so show how these higher resolution processes may influence the projections.

Many users are interested in estuaries or the coastal zone. These are currently beyond the scope of either AMM7 or AMM15. CO6 has a minimum depth of 10 m (in PDCtrl), to ensure the model doesn't dry out and crash. This is not necessary in CO9 (NWSPPE) where a wetting and drying module allows for more realistic coastal bathymetry. We suggest locations within three grid boxes of the land are used with care, and that the impact of the model's omission of coastal processes is considered on a case-by-case basis.

Our methodology uses an uncoupled, one-way forcing approach. While the HadGEM3-GC3.05 is coupled, and so implicitly includes coupled processes in the COx model forcings, COx is uncoupled, and so cannot feed back to the atmosphere or wider ocean. Coupled atmosphere ocean processes can be very important in the tropics, and there is increasing attention to their importance in the NWS. For example, forecasting sea fog (Fallmann et al., 2019), ocean-waves coupling Lagrangian trajectories (Bruciaferri et al., 2021), and heat waves (Petch et al., 2020), can be improved by using a regional coupled model. The Met Office is developing a regional ocean-atmosphere waves coupled model (UKC4) using the NEMO COx AMM15 model and the 2.2 km variable resolution MOGREPS UK (Hagelin et al., 2017) with the RAL3.1 scientific options (Flack et al., 2022). This will eventually couple with a biogeochemistry model and river model, to allow forecasting of multi-hazard compound events such as coastal flooding and erosion. This model is very complex and expensive to run, with the atmosphere component being much more expensive to run than the (already expensive) AMM15 model. While exploratory climate simulations of future time slices and case-studies are possible, this is even less likely than AMM15 to be used as the basis of an ensemble of transient climate projections. However, its use with an ensemble of projections can help understand what processes are poorly represented, and how the projections may over or under predict certain phenomena, when compared to (things like) UKCP18.

Our NWSPPE gives a quantification of one source of climate uncertainty. By varying parameters within the atmosphere, land and aerosol components of HadGEM3-GC3.05, the range of the responses of the NWS climate can be assessed. This is an important source, but there are many other sources of uncertainty in climate projections. Other important sources include emission uncertainty, model structure uncertainty (of the global model, the shelf seas model, and even the regional atmospheric

climate model), initial condition uncertainty, and methodological uncertainty (coupling and driving uncertainty). To date, the
Minerva Projections (Tinker et al., 2016) have been the only systematic and comprehensive assessment of climate uncertainty within the NWS (Tinker and Howes, 2020). While our NWSPPE updates the Minerva Projections, they do not give any further insights in these other sources. A climate uncertainty budget for the NWS, which quantified and compared these sources, would be a useful contribution to the research field.

Our NWSPPE updates the Minerva Projections by making use of more advanced models but does not extend their use. The next generation of NWS climate projections may couple to other components, such as a wave and or a biogeochemistry models – both of which are coupled to COx for the Met Office operational synoptic forecasts. This will represent the third (next) generation set of NWS climate projections.

Our approach updates the Minerva NWS climate projections, which were based on those of Holt et al. (2010). We feel that these projections are needed for the UK's upcoming 4th CCRA, but that the next set of UK climate projection should include the third generation NWS climate projections. These could address many of the limitations outlined above, by utilising: improved resolution (AMM15); improved coupling methodology (UKC4); additional model systems (biogeochemistry, waves/surge); different sources of uncertainty. These may require a range of methodologies, perhaps employing a cascade of model resolutions: AMM7 transient ensembles to address different sources of uncertainty, transient simulation of the standard ensemble member with AMM15, and case studies and times slices with UKC4. While coupling to a wave model does not increase the computational expense much (~10%), adding biogeochemistry does (3-6 times), so BGC may be limited to the lower resolution AMM7 (or with mixed resolution) simulations (as it is currently in operational forecasts). Adding biogeochemistry would allow a much wider range of end users to use the projections, while adding waves and surge would increase consistency and improve confidence. The next set of NWS climate projections may require a more community led approach to support their greater complexity. Close proactive engagement and collaboration with end users during the planning and development stage will enable uptake, and the generation of user-relevant impacts to inform evidence-based coastal decision-making (Weeks et al., 2023).

## 7    Conclusions

Our key findings and conclusions are:

- The NWS annual mean SST rises by 3.11 °C (±0.98 °C) for 2079-2098 relative to 2000-2019, and with a greater warming in the summer and autumn.

- The duration of the seasonal stratification increases by almost one month, initialising 4 days earlier, and breaking down ~19 day later.

- There is a region of limited warming in the deep water south of Iceland, associated with the AMOC slow down.

- There is substantial freshening across the NWS and domain, with the NWS freshening of −1.01 psu ($\pm 2\sigma = 0.93$ psu) for 2079-2098 relative to 2000-2019. The rate of freshening increases through the 21$^{st}$ century.

- There is a general reduction in strength of most of the NWS residual circulation over the 21$^{st}$ century, and little change in its configuration, apart from the south and west of Ireland, where there is a slight change in the configuration, and a strengthening of the circulation. 80% of the area to the south and west of the Ireland shows an increase in the residual current magnitude, whereas 60% of the remainder of the NWS shows a decrease.

- Using the NWSPPE with the PDCtrl allows the climate signal to be considered in the context of unforced natural variability. We give an example of how these can be used together to estimate the Time of Emergence of the climate signal emerges from the natural variability. For example, the ToE for the annual mean SST averaged over the NWS is 2034, suggesting that the entire ensemble has moved out of the natural variability compared to 2000-2019.

- The ToE gives confidence in the use of our projections in the mid-century, and we discuss how they could be used to simulate changes consistent with 2°C and 4 °C warming levels.

- These projections are for the physical environment, and so give no information about possible changes to the ecosystem (e.g. productivity, biomass, chlorophyll), biogeochemistry (e.g. nutrient levels, oxygen levels) or the wave conditions (e.g. significant wave height, wave period). In addition to including these other parameters, future marine climate projections may have greater spatial resolution, employ more sophisticated coupling strategies, and assess additional sources of climate projection uncertainty.

## 8    Code/Data Availability

The NWSPPE and PDCtrl datasets are available via CEDA

(https://catalogue.ceda.ac.uk/uuid/832677618370457f9e0a85da021c1312), and can be accessed via their Digital Object

Identifiers:

- https://dx.doi.org/10.5285/edf66239c70c426e9e9f19da1ac8ba87
- https://dx.doi.org/10.5285/66e39885a60e4b6386752b1a295f268a
- https://dx.doi.org/10.5285/bd375134bd8c4990a1e9eb6d199cc723

See Appendix A (section 15, specifically section 15.9) for more information.

The post processing code (for the data release) is available on GitHub, see section 15.8 for more information.

The NEMO shelf climate configurations are also available on GitHub, see section 2.2.2 for more information.

The residual current uncertainty ellipse python code is available on GitHub: https://github.com/hadjt/CurrUncertEllipses.

## 9    Author Contribution

DS, KY and JR developed and ran the HadGEM3-GC3.05 PPE and added the specific diagnostics for the shelf seas downscaling. JT and MP designed the experiment. JT and EO adapted the NEMO configuration for use in the climate simulations. JT pre- and post-processed the files, undertook the analysis and visualisation of the data, and prepared the data release. BH helped with code review. JT, with MP and BH, prepared the initial manuscript, and all authors contributed to subsequent iterations.

## 10    Competing Interests

The authors declare that they have no conflict of interest.

## 11    Acknowledgement

All authors were supported by the Met Office Hadley Centre Climate Programme funded by BEIS and Defra.

We would like to thank several people who helped and advised during this study. Alex Arnold and Ségolène Berthou described the regional coupled configuration (UKC4), and Ségolène Berthou helped interpret the changes in the seasonal cycle. Carol McSweeney and Dan Bernie advised on the 2 °C and 4 °C warming levels. Daley Calvert helped move the NEMO configurations from the Met Office repository to GitHub. Leon Hermanson advised on the analysis and interpretation of the Atlantic Meridional Overturning Circulation (AMOC) slow-down. Laura Jackson advised on the projected changes in the North Atlantic salinity. Richard Renshaw advised on the Copernicus Marine Reanalysis, and the Baltic Exchange, and the comparison of its variability to the North Sea river variability. Susan Kay advised on the possible ecosystem impacts of a change in the stratification. Fai Fung helped with the definition of the dataset, and (with Carol McSweeney), the alignment with the UKCP18 project and datasets. Jeff Polton gave insight into the 360-day tides. Stephen Dye advised on the use of ICES climate timeseries data. Richard Renshaw helped with code review. Matt Frost, Lou Rutterford and Paul Buckley advised on marine biodiversity legislation and the policy relevance of the study. We would particularly like to thank Diane Knappett and Ag Stephens for archiving and publishing the dataset.

This paper is dedicated to Alexa Vega Martin and is in remembrance of Max Tinker.

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

**Table 1 Overview of observation datasets (and their analysis) used in the NWSPPE evaluation.**

| Variable | Dataset | Method |
|---|---|---|
| **Tidal phase and amplitude** | O'Dea et al. (2012, 2017) | Co-tidal charts compared to evaluated AMM7 cotidal charts of O'Dea et al. (2012, 2017) |
| **SST** | OSTIA analysis (Roberts-Jones et al., 2012) | Ensemble mean biases for the 4 seasons, and assessment of where OSTIA is within the NWSPPE distribution |
| **Sub-surface temperature and salinity** | EN4 quality-controlled subsurface profiles dataset (Good et al., 2013). | Methodology adapted from Tinker et al. and outlined in section 3.2. |
| **Mean sea level** | AVISO MDT (Rio et al., 2014) | 1993–2012 and compare to the NWSPPE ensemble mean SSH for the present-day ensemble statistics (2000-2019). |
| **Sea Level interannual variability** | Satellite: C3S SLA product (Legeais et al., 2018)<br>Tide gauges: Permanent Service for Mean Sea Level (PSMSL, Holgate et al. (2013) | Satellite: Interannual variability is calculated as the standard deviation of the detrended the annual means (1993–2018 for C3S product, 2000-2019 for NWSPPE).<br>Tide Gauges: timeseries are compared, by adding an offset to the NWSPPE calculated from a common overlap period. |
| **ICES long term Temperature and Salinity timeseries.** | ICES Report on Ocean Climate, 2023 (González-Pola et al., 2022). ICES, Copenhagen, https://ocean.ices.dk/core/iroc on the December 2023. | Model and observations compared for 2000-2019. Data profiles extracted for the nearest model gridbox, and are depth processed into timeseries to match the observations. |
| **Volume Transport through cross-sections** | Observed volume transport estimates from the literature (Fernand et al., 2006; Brown et al., 1999; Svendsen et al., 1991; Turrell et al., 1992; Danielssen et al., 1997; Prandle et al., 1996; Holt et al., 2001). | Modelled volume transport averaged into a 2000-2019 seasonal cycle. The ensemble members, and mean is plotted compared to the observational estimate (and observational estimate where available). |

**Table 2 ICES long term climate timeseries used in the evaluation of the NWSPPE, giving the location, depth of water, and how the timeseries was calculated from the temperature and salinity depth profiles.**

| | Location | Depth of water | Depth processing |
|---|---|---|---|
| **Faroe Shetland NAW** | 61.00°N 3.00°W | 712 m | Upper 200m mean |
| **Faroe Shetland MNAW** | 61.50°N 6.00°W | 194 m | Upper 200m mean |
| **Utsira A** | 59.00°N 0.50°E | 141 m | Near-bed |
| **Utsira B** | 59.00°N 4.00°E | 273 m | High salinity core of Atlantic Water. Timeseries from the depth of maximum salinity |
| **Fair Isle Current Water** | 59.00°N 2.00°W | 75 m | Depth mean - averaged from 2 locations (59.283°N 2.233°W; 59.283°N 1.933°W) |
| **Helgoland Roads** | 54.18°N 7.90°E | 14 m | Surface |
| **Plymouth WCO E1** | 50.03°N 4.37°W | 69 m | Upper 40m mean |
| **Astan** | 48.78°N 3.94°W | 59 m | Surface |
| **Irish Sea AFBI** | 53.78°N 5.63°W | 85 m | Near-bed |
| **Ireland M3** | 51.22°N 10.55°W | 140 m | Surface |
| **Ireland Malin** | 55.37°N 7.34°W | 19 m | Surface |

**Table 3 Tide gauges used for model comparison.**

| ID | Name | Longitude | Latitude | Period |
|---|---|---|---|---|
| 01 | Aberdeen I | -2.08 | 57.14 | 1931-2022 |
| 02 | Bergen | 5.32 | 60.40 | 1915-2022 |
| 03 | Brest | -4.49 | 48.38 | 1807-2021 |
| 04 | Cherbourg | -1.64 | 49.65 | 1974-2021 |
| 05 | Delfzijl | 6.93 | 53.33 | 1865-2022 |
| 06 | Den Helder | 4.75 | 52.96 | 1865-2022 |
| 07 | Dieppe | 1.08 | 49.93 | 1954-2021 |
| 08 | Holyhead | -4.62 | 53.31 | 1938-2022 |
| 09 | Immingham | -0.19 | 53.63 | 1959-2022 |
| 10 | La Rochelle, La Pallice | -1.22 | 46.16 | 1941-2021 |
| 11 | Lerwick | -1.14 | 60.15 | 1957-2022 |
| 12 | Lowestoft | 1.75 | 52.47 | 1955-2022 |
| 13 | Newhaven | 0.06 | 50.78 | 1991-2022 |
| 14 | Newlyn | -5.54 | 50.10 | 1915-2022 |
| 15 | North Shields | -1.44 | 55.01 | 1895-2022 |
| 16 | Sheerness | 0.74 | 51.45 | 1832-2022 |
| 17 | St Nazaire | -2.20 | 47.27 | 1941-2021 |
| 18 | Stornoway | -6.39 | 58.21 | 1977-2022 |
| 19 | Tobermory | -6.06 | 56.62 | 1989-2022 |
| 20 | Wick | -3.09 | 58.44 | 1965-2022 |

**Table 4** Evaluation Transport cross-sections, giving the cross-section name and number, the longitude and latitude of the two ends, the direction that is considered positive, and whether the transport is the positive component, or the net transport. The observed estimates (with error estimates where available), and NWSPPE Ensemble mean annual mean (and seasonal range).

| Cross-sections | | Lon-Lat | | | | | |
|---|---|---|---|---|---|---|---|
| # | Name | start | stop | Observations (uncertainty range) | NWSPPE Annual and Ens Mean (± 2σ Ens Mean Seasonal range) | Positive Direction (that transport is going to) | Component of transport |
| 1 | Into Skagerrak (excl. outflow) | 57.20°N 9.00°E | 58.47°N 9.00°E | 1.00 Sv (0.50 - 1.50 Sv) | 1.21 Sv (0.94 - 1.41 Sv) | E | +ve |
| 2 | North of Dogger Bank | 56.00°N 2.00°E | 55.00°N 2.00°E | 0.05 Sv | 0.08 Sv (0.03 - 0.11 Sv) | E | Net |
| 3 | Dooley Current | 58.00°N 0.00°W | 57.00°N 0.00°W | 0.25 Sv (0.12 - 0.38 Sv) | 0.25 Sv (0.23 - 0.30 Sv) | E | Net |
| 4 | Shetland-Norwegian Trench (inflow) | 60.73°N 0.78°W | 60.73°N 3.00°E | 0.60 Sv | 0.61 Sv (0.51 - 0.71 Sv) | S | +ve |
| 5 | Norwegian Trench-Norway (outflow) | 60.73°N 3.00°E | 60.73°N 4.89°E | 1.80 Sv | 1.56 Sv (1.31 - 1.81 Sv) | N | +ve |
| 6 | Fair Isle | 60.20°N 1.33°W | 59.00°N 3.00°W | 0.20 Sv | 0.32 Sv (0.20 - 0.46 Sv) | SE | Net |
| 7 | Hebrides Shelf | 59.67°N 5.00°W | 58.47°N 5.00°W | 0.25 Sv | 0.43 Sv (0.33 - 0.58 Sv) | E | Net |
| 8 | Irish Shelf | 53.33°N 12.22°W | 53.33°N 9.78°W | 0.25 Sv | 0.12 Sv (-0.04 - 0.20 Sv) | N | Net |
| 9 | St Georges outflow (net) | 51.27°N 6.44°W | 52.13°N 7.44°W | 0.18 Sv | 0.11 Sv (0.02 - 0.20 Sv) | SW | Net |
| 10 | Dover Strait | 51.20°N 1.33°E | 50.87°N 1.78°E | 0.10 Sv | 0.07 Sv (0.03 - 0.12 Sv) | NE | Net |
| 11 | Shelf Current at 56.7°N | 56.67°N 10.00°W | 56.67°N 8.00°W | 1.90 Sv (1.30 - 2.50 Sv) | 2.52 Sv (1.80 - 3.33 Sv) | N | Net |
| 12 | Shelf Current FS Current (net) | 61.20°N 3.44°W | 60.53°N 2.22°W | 5.50 Sv (4.00 - 7.00 Sv) | 2.68 Sv (1.08 - 3.89 Sv) | NE | Net |

**Table 5 Projected regional mean SST changes between 2000-2019 and 2079-2098. Ensemble mean changes are given with ±2 (ensemble) standard deviations.**

|  | Shelf | Southern North Sea | Central North Sea | Northern North Sea | English Channel | Irish Sea | Celtic Sea |
|---|---|---|---|---|---|---|---|
| **ANN** | 3.11 °C (±0.98 °C) | 3.72 °C (±1.03 °C) | 3.59 °C (±1.07 °C) | 3.14 °C (±1.02 °C) | 3.34 °C (±0.88 °C) | 3.22 °C (±1.03 °C) | 3.01 °C (±0.90 °C) |
| **DJF** | 2.72 °C (±0.97 °C) | 3.55 °C (±1.20 °C) | 3.20 °C (±1.07 °C) | 2.71 °C (±0.99 °C) | 3.06 °C (±0.95 °C) | 3.03 °C (±1.00 °C) | 2.43 °C (±0.83 °C) |
| **MAM** | 2.43 °C (±1.01 °C) | 3.02 °C (±1.08 °C) | 2.97 °C (±1.16 °C) | 2.56 °C (±1.08 °C) | 2.69 °C (±0.88 °C) | 2.56 °C (±1.04 °C) | 2.21 °C (±0.87 °C) |
| **JJA** | 3.57 °C (±1.09 °C) | 3.78 °C (±1.04 °C) | 4.05 °C (±1.18 °C) | 3.61 °C (±1.20 °C) | 3.71 °C (±0.91 °C) | 3.67 °C (±1.09 °C) | 3.62 °C (±0.96 °C) |
| **SON** | 3.73 °C (±1.07 °C) | 4.55 °C (±1.06 °C) | 4.11 °C (±1.07 °C) | 3.68 °C (±1.05 °C) | 3.88 °C (±0.96 °C) | 3.65 °C (±1.11 °C) | 3.78 °C (±1.12 °C) |

**Table 6 Projected regional mean NBT changes between 2000-2019 and 2079-2098. Ensemble mean changes are given with ±2 (ensemble) standard deviations.**

|  | Shelf | Southern North Sea | Central North Sea | Northern North Sea | English Channel | Irish Sea | Celtic Sea |
|---|---|---|---|---|---|---|---|
| **ANN** | 2.49 °C (±0.94 °C) | 3.65 °C (±1.01 °C) | 2.84 °C (±0.96 °C) | 2.28 °C (±0.96 °C) | 3.15 °C (±0.85 °C) | 2.87 °C (±0.97 °C) | 2.19 °C (±0.87 °C) |
| **DJF** | 2.71 °C (±0.99 °C) | 3.54 °C (±1.20 °C) | 3.18 °C (±1.07 °C) | 2.57 °C (±0.98 °C) | 3.06 °C (±0.95 °C) | 3.01 °C (±1.00 °C) | 2.37 °C (±0.89 °C) |
| **MAM** | 2.39 °C (±1.02 °C) | 2.98 °C (±1.07 °C) | 2.83 °C (±1.10 °C) | 2.41 °C (±1.04 °C) | 2.63 °C (±0.86 °C) | 2.43 °C (±1.01 °C) | 2.24 °C (±0.92 °C) |
| **JJA** | 2.29 °C (±0.95 °C) | 3.55 °C (±1.00 °C) | 2.43 °C (±0.99 °C) | 2.07 °C (±1.00 °C) | 3.25 °C (±0.83 °C) | 2.63 °C (±0.97 °C) | 2.09 °C (±0.88 °C) |
| **SON** | 2.58 °C (±0.94 °C) | 4.53 °C (±1.06 °C) | 2.90 °C (±0.89 °C) | 2.06 °C (±0.97 °C) | 3.66 °C (±0.93 °C) | 3.42 °C (±1.04 °C) | 2.08 °C (±0.87 °C) |

**Table 7 Projected regional mean DFT (SST – NBT) changes between 2000-2019 and 2079-2098. Ensemble mean changes are given with ±2 (ensemble) standard deviations.**

|  | Shelf | Southern North Sea | Central North Sea | Northern North Sea | English Channel | Irish Sea | Celtic Sea |
|---|---|---|---|---|---|---|---|
| **ANN** | 0.62 °C (±0.34 °C) | 0.07 °C (±0.04 °C) | 0.75 °C (±0.36 °C) | 0.86 °C (±0.44 °C) | 0.19 °C (±0.08 °C) | 0.35 °C (±0.14 °C) | 0.82 °C (±0.41 °C) |
| **DJF** | 0.01 °C (±0.16 °C) | 0.01 °C (±0.02 °C) | 0.02 °C (±0.03 °C) | 0.14 °C (±0.16 °C) | 0.00 °C (±0.01 °C) | 0.02 °C (±0.03 °C) | 0.06 °C (±0.24 °C) |
| **MAM** | 0.03 °C (±0.27 °C) | 0.04 °C (±0.04 °C) | 0.14 °C (±0.22 °C) | 0.15 °C (±0.25 °C) | 0.06 °C (±0.05 °C) | 0.13 °C (±0.10 °C) | -0.03 °C (±0.38 °C) |
| **JJA** | 1.28 °C (±0.63 °C) | 0.23 °C (±0.14 °C) | 1.63 °C (±0.75 °C) | 1.53 °C (±0.78 °C) | 0.46 °C (±0.19 °C) | 1.04 °C (±0.39 °C) | 1.53 °C (±0.67 °C) |
| **SON** | 1.15 °C (±0.60 °C) | 0.02 °C (±0.02 °C) | 1.20 °C (±0.63 °C) | 1.62 °C (±0.89 °C) | 0.22 °C (±0.12 °C) | 0.23 °C (±0.15 °C) | 1.70 °C (±0.70 °C) |

**Table 8** Projected regional mean SSS changes between 2000-2019 and 2079-2098. Ensemble mean changes are given with ±2 (ensemble) standard deviations.

| | Shelf | Southern North Sea | Central North Sea | Northern North Sea | English Channel | Irish Sea | Celtic Sea |
|---|---|---|---|---|---|---|---|
| **ANN** | -1.01 psu (±0.93 psu) | -0.94 psu (±1.00 psu) | -0.97 psu (±0.94 psu) | -1.06 psu (±1.03 psu) | -0.81 psu (±0.80 psu) | -0.98 psu (±0.77 psu) | -0.97 psu (±0.79 psu) |
| **DJF** | -1.01 psu (±0.92 psu) | -0.90 psu (±1.01 psu) | -0.96 psu (±0.92 psu) | -1.06 psu (±1.02 psu) | -0.80 psu (±0.79 psu) | -0.96 psu (±0.78 psu) | -0.98 psu (±0.79 psu) |
| **MAM** | -1.02 psu (±0.92 psu) | -0.94 psu (±1.01 psu) | -0.99 psu (±0.93 psu) | -1.08 psu (±1.02 psu) | -0.83 psu (±0.78 psu) | -1.02 psu (±0.78 psu) | -0.97 psu (±0.77 psu) |
| **JJA** | -1.00 psu (±0.94 psu) | -0.96 psu (±1.02 psu) | -0.97 psu (±0.96 psu) | -1.02 psu (±1.05 psu) | -0.81 psu (±0.82 psu) | -1.01 psu (±0.78 psu) | -0.94 psu (±0.79 psu) |
| **SON** | -1.02 psu (±0.95 psu) | -0.94 psu (±1.00 psu) | -0.95 psu (±0.96 psu) | -1.05 psu (±1.09 psu) | -0.80 psu (±0.82 psu) | -0.92 psu (±0.76 psu) | -0.97 psu (±0.83 psu) |

**Table 9 Timings of exceeding global mean warming levels of 2°C and 4 °C from the UKCP18 HadGEM3-GC3.05 PPE. The global warming levels are derived from the model simulated global annual mean anomaly relative to 1981-2000 baseline plus the observed warming from 1850-1900 mean to 1981-2000 mean based on HadCRUT4 (Morice et al., 2012) Timings are based on a centred 25 year running mean. While all simulations pass 2 °C of global mean warming some simulations do not reach global mean warming levels of 4 °C by the end of the century. Adapted from Gohar et al. (2018).**

| Ensemble Member | Global 2°C earliest passing, and range | Global 4°C earliest passing, and range |
|---|---|---|
| r001i1p00000 | 2030 (2021-2040) | 2063 (2054-2073) |
| r001i1p00605 | 2027 (2018-2037) | 2061 (2052-2071) |
| r001i1p00834 | 2030 (2021-2040) | 2060 (2051-2070) |
| r001i1p01113 | 2027 (2018-2037) | 2060 (2051-2070) |
| r001i1p01554 | 2032 (2023-2042) | 2066 (2057-2076) |
| r001i1p01649 | 2029 (2020-2039) | 2064 (2055-2074) |
| r001i1p01843 | 2031 (2022-2041) | 2064 (2055-2074) |
| r001i1p01935 | 2032 (2023-2042) | 2070 (2061-2080) |
| r001i1p02123 | 2028 (2019-2038) | 2057 (2048-2067) |
| r001i1p02242 | 2032 (2023-2042) | 2067 (2058-2077) |
| r001i1p02491 | 2029 (2020-2039) | 2064 (2055-2074) |
| r001i1p02868 | 2035 (2026-2045) | 2067 (2058-2077) |

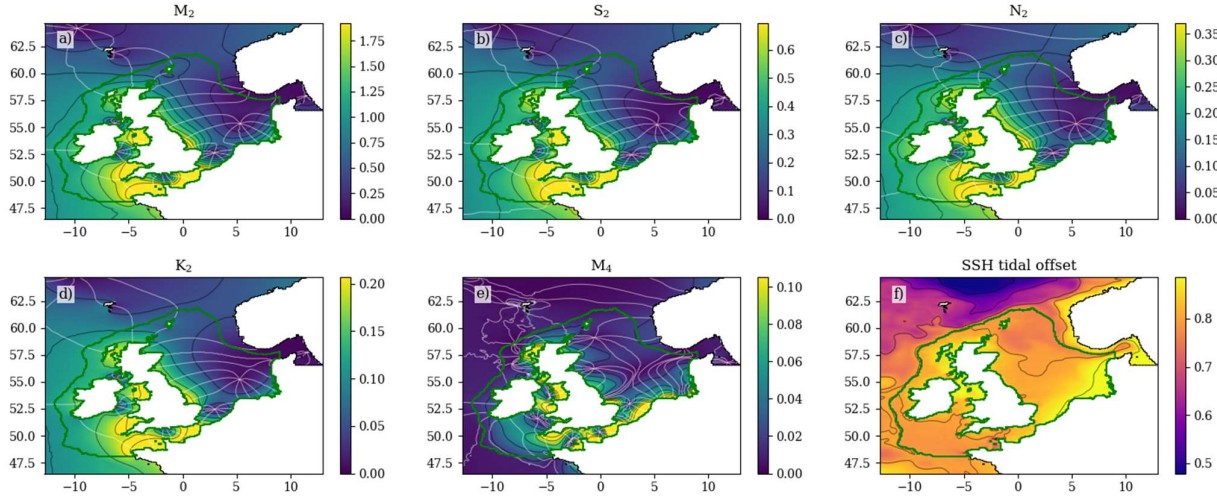

**Figure 1 The (2000-2019) ensemble mean Co-tidal chart for the 4 largest constituents (a) M$_2$; b) S$_2$; c) N$_2$; and d) K$_2$), e) the shallow water component (M$_4$); and f) the offset (mean sea level). Amplitude (m) is represent by the colour map, with black contours matching the colorbar tick labels. The grey contours give the phase in 45° intervals.**

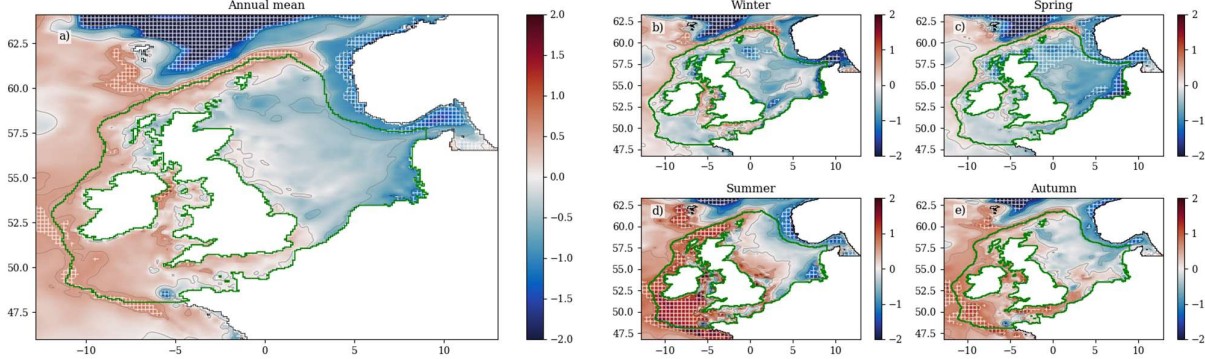

**Figure 2 SST bias of the PPE Ensemble mean for 2000-2019 (for the 4 seasons) compared to the OSTIA SST. The hatching shows where the OSTIA SSTs is more than two (ensemble) standard deviations from the ensemble mean.**

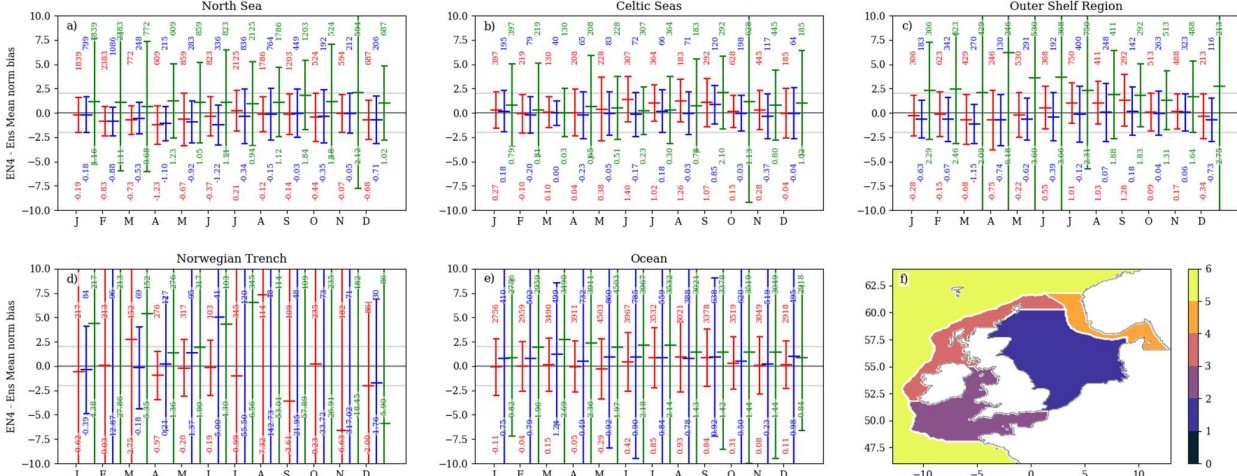

**Figure 3 Regional mean distribution of normalised Model minus EN4 profile observations, for SST (red), NBT (blue) and SSS (green). Each EN4 profile – model pair is used to calculate a normalised bias (model minus ensemble mean divided by ensemble standard deviation), as shown in Figure S4-Figure S6. These data points are separated into distributions by validation regions and month – the different regions are plotted as separate subpanels, and the months are separated along the x axis (showing month number). Each distribution is plotted as the mean and ± 2 standard deviation, with 2 numbers– the number of points in each distrbution is the upper number, and the distirbution mean is the lower number. The validation region mask is giving in subpanel f, which denotes a) the North Sea; b) the Celtic Seas; c) Outer Shelf Region; d) the Norwegian Trench (and Skagerak and Kategat) and e) the oceanic regions. This figure is repeated with the Wakelin region mask in Figure S3, giving greater granualarity, but smaller sample sizes.**

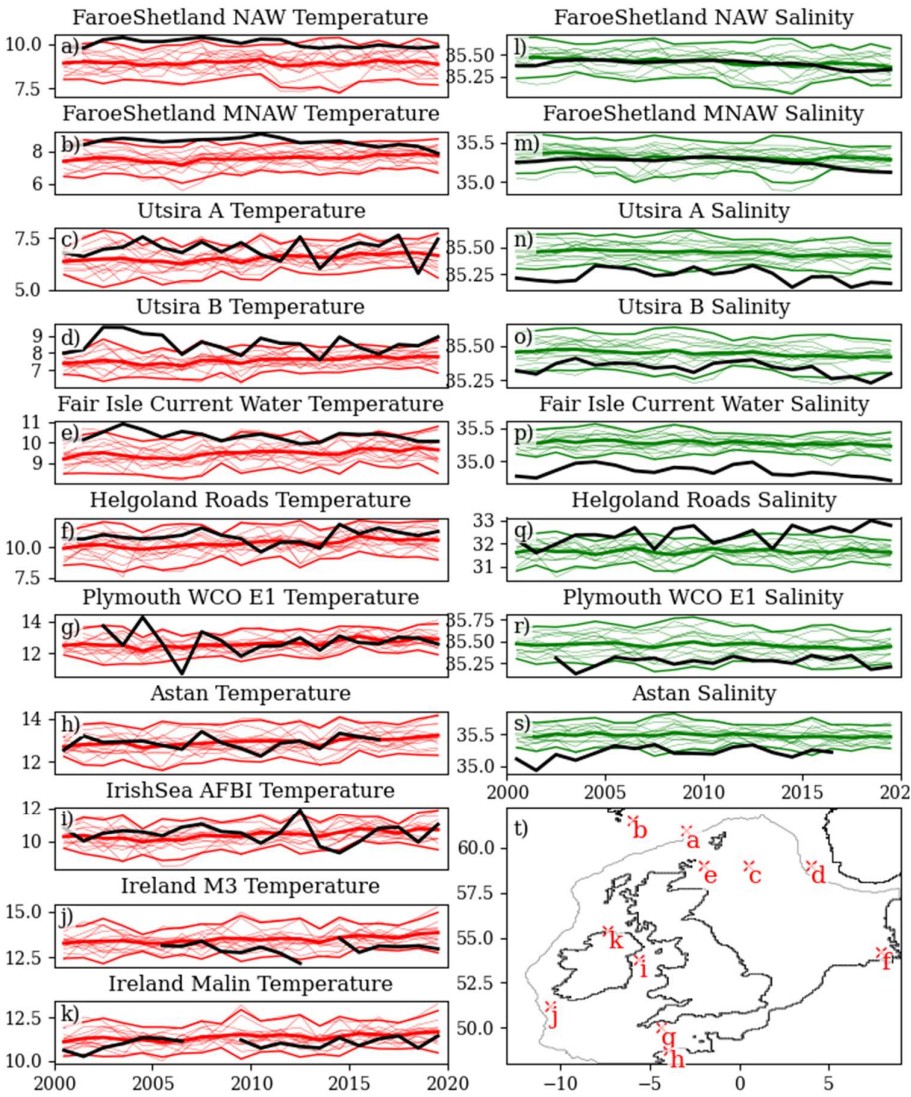

# T/S timeseries(2000-2019)
## ICES moorings vs NWSPPE


**Figure 4 comparison of the NWSPEE with long annual mean timeseries of in situ T and S, from around the NWS (see text). The left column shows temperature, and the right shows salinity. i-k only measured temperature. These records are a mix of surface, near bed, water column mean and the mean of the upper water column.**


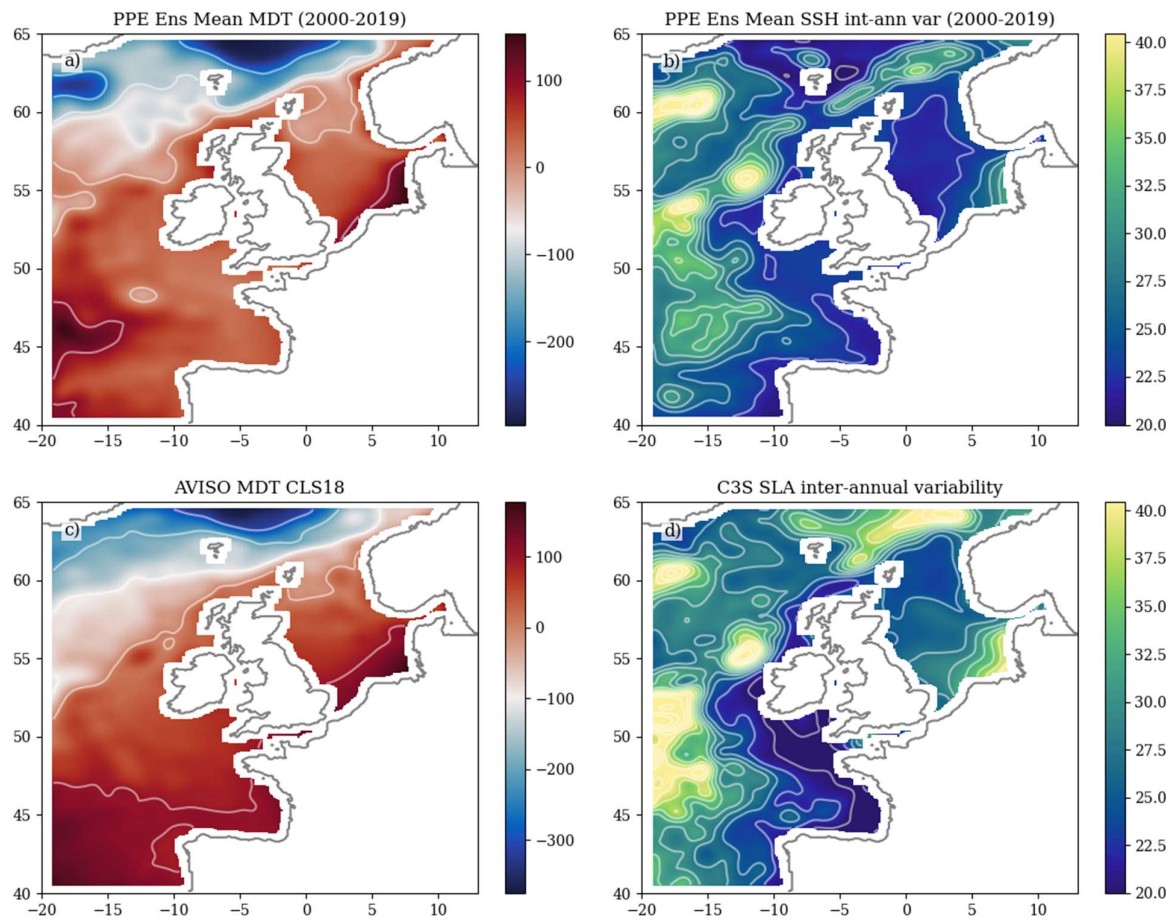

**Figure 5 Comparison of Modelled SSH with satellite altimetry (mm). a) Ensemble mean SSH anomaly (2000-2019) b) Ensemble interannual variability (standard deviations of annual mean SSH, averaged over the ensemble, 2000-2019), c) AVISO MDT (mean SSH), d) CS3 interannual variability of Sea Level Anomaly.**

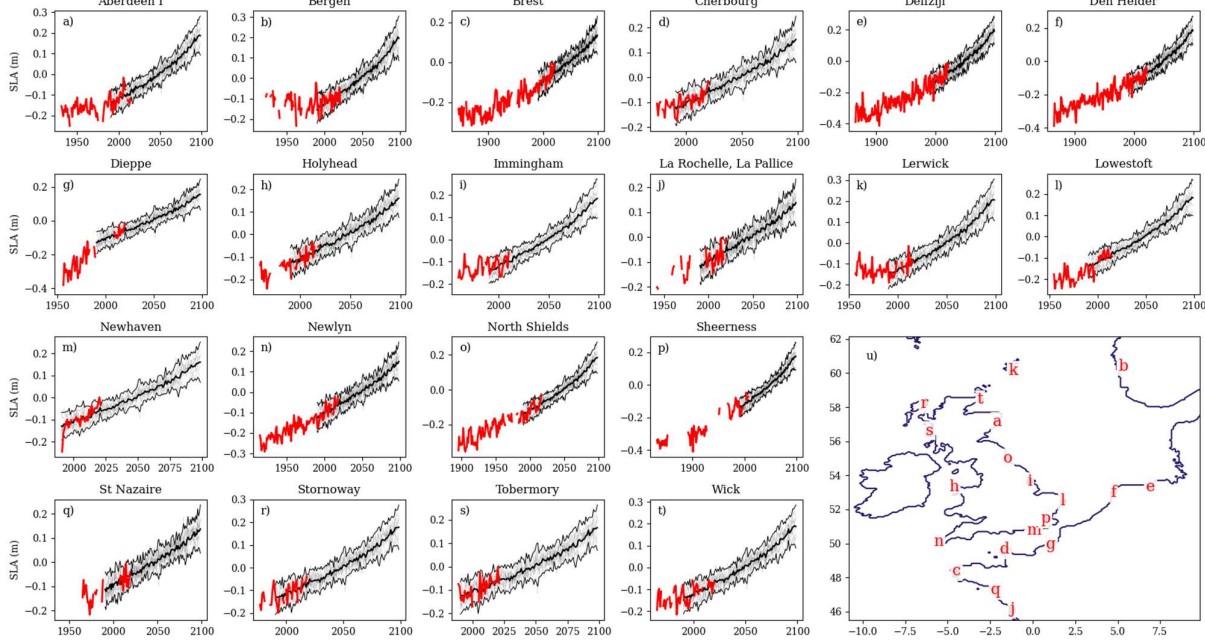


**Figure 6 Tide gauge - model sea level anomaly comparison (metres). a-t) timeseries of annual mean SSH from tide gauges (red), and the 12 ensemble members (grey), with the ensemble mean ± 2 ensemble standard deviations shown in black. No account is taken of the differing bench marks between the model and the tide gauges, so an arbitrary offset is added to align the two datasets to allow**
**visual comparison of the trends and variability. Note the differing time tide gauge record length leads to different ranges on the x and y axes. u) Tide gauge locations are given in Table 3.**

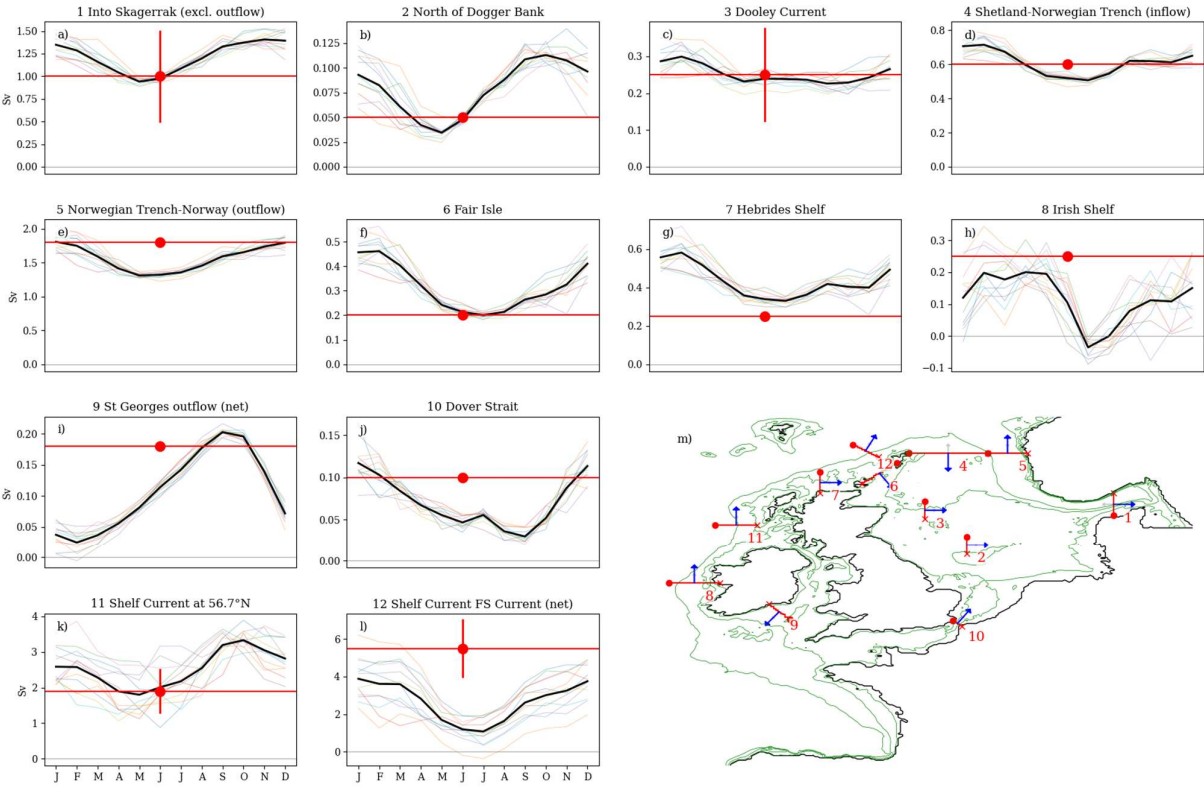


**Figure 7 Comparison of modelled and observed volume transport through cross-sections. a-l: the NWSPPE Ensemble mean (bold black) seasonal cycle (and ensemble members, coloured thin lines) compared to observed transport estimate (red circle, and horizontal line), and observation error estimate (vertical red line) where available. M) map of the NWS giving the location of the cross-sections. Table 4 gives the longitudes and latitudes of the ends of the cross-sections, and summarise the data.**


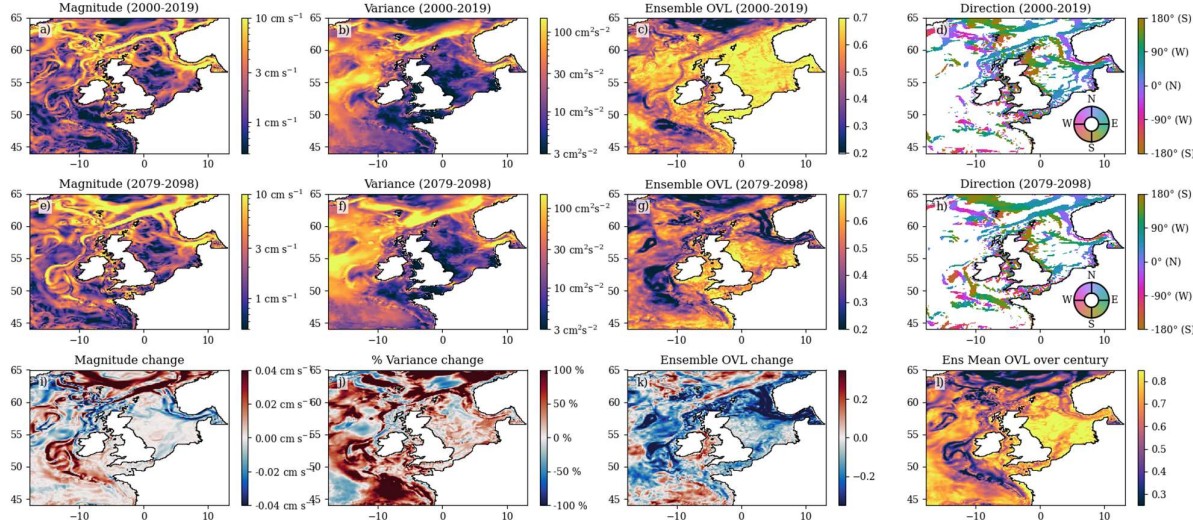

Figure 8 Barotropic residual current statistics. The early 21st century (2000-2019), late 21st century (2079-2098) and difference of the ensemble mean current magnitude (a, e, i respectively), variance, as measured by the area of the 2.45 standard deviation current ellipse (b, f, j), the Ensemble OVL (the overlap of the ensemble, as a measure of how similar the ensemble members are (c, g, k) and the early and late 21st century current configuration (colours show the direction of the currents where the mean is more than 1 standard deviation. The ensemble mean of the OVL for each ensemble member when comparing early and late 21st century current distribution.

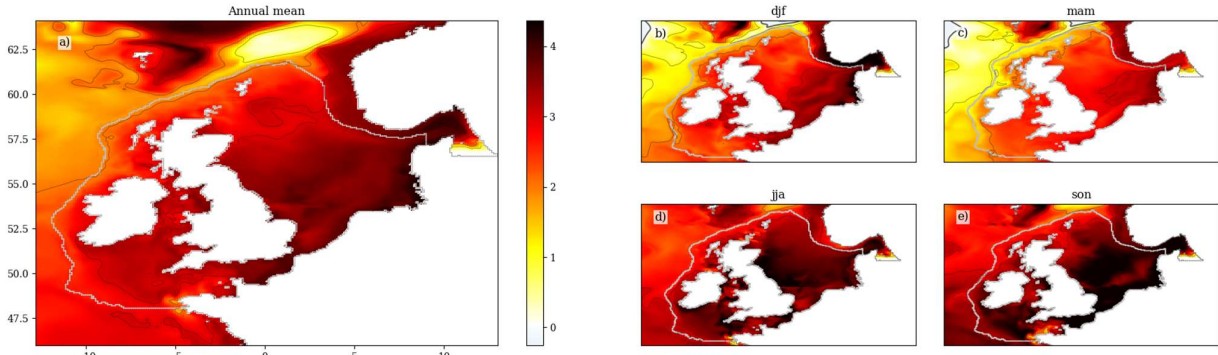


**Figure 9 The SST ensemble mean change over the 21$^{st}$ Century (2079-2098 compared to 2000-2019), for the: a) annual mean; b) winter (December to February); c) spring (March to May); d) summer (June – August) and e) autumn (September – November). Black contours relate to the colour bar tickes, and the zero contour is a thicker black line. The coast line and NWS region are delinated with a grey line, and Full comparsion between the early and late century, and the components of variance (ensemble variance and interannual variability), see Figure S17.**


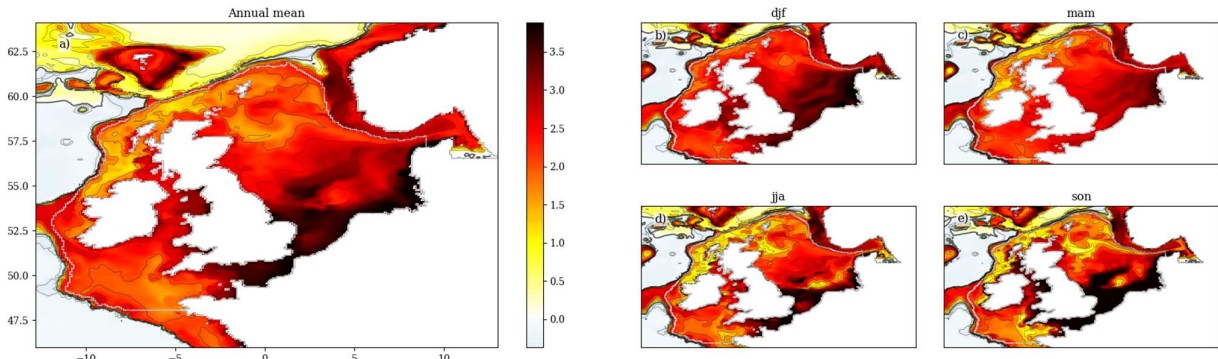

**Figure 10 The NBT ensemble mean change over the 21ˢᵗ Century (2079-2098 compared to 2000-2019), for the: a) annual mean; b) winter (December to February); c) spring (March to May); d) summer (June – August) and e) autumn (September – November). Black contours relate to the colour bar tickes, and the zero contour is a thicker black line. The coast line and NWS region are delinated with a grey line, and Full comparsion between the early and late century, and the components of variance (ensemble variance and interannual variability), see Figure S18.**


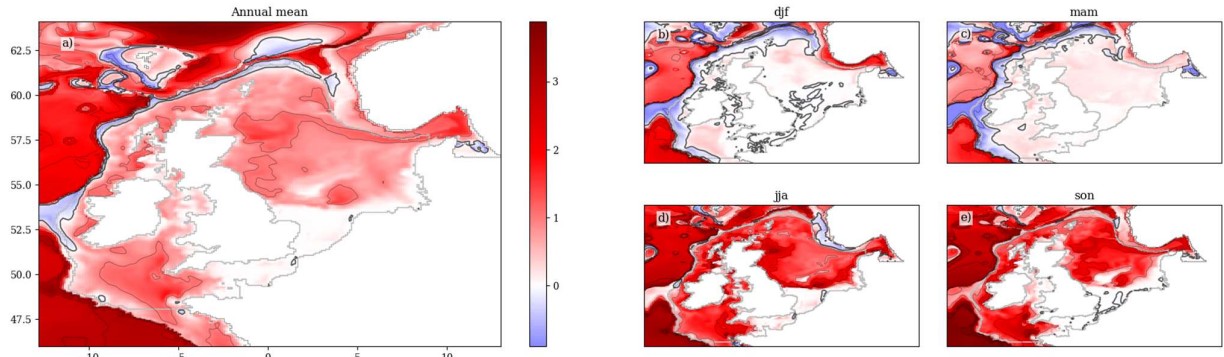

**Figure 11 The surface-bed temperature difference (DFT) ensemble mean change over the 21ˢᵗ Century (2079-2098 compared to 2000-2019), for the: a) annual mean; b) winter (December to February); c) spring (March to May); d) summer (June – August) and e) autumn (September – November). Black contours relate to the colour bar tickes, and the zero contour is a thicker black line. The coast line and NWS region are delinated with a grey line, and Full comparsion between the early and late century, and the components of variance (ensemble variance and interannual variability), see Figure S19.**


## Stratification timing statistic (pea_gt_10_mld_lt_50)

**Figure 12 The ensemble mean duration, initialisation and break down of stratification, in day of year, (first, second and third column respectively) for the early 21$^{st}$ century (2000-2019), late 21$^{st}$ century (2079-2098) and how it changes between the two periods (first, second and third rows respectively).**

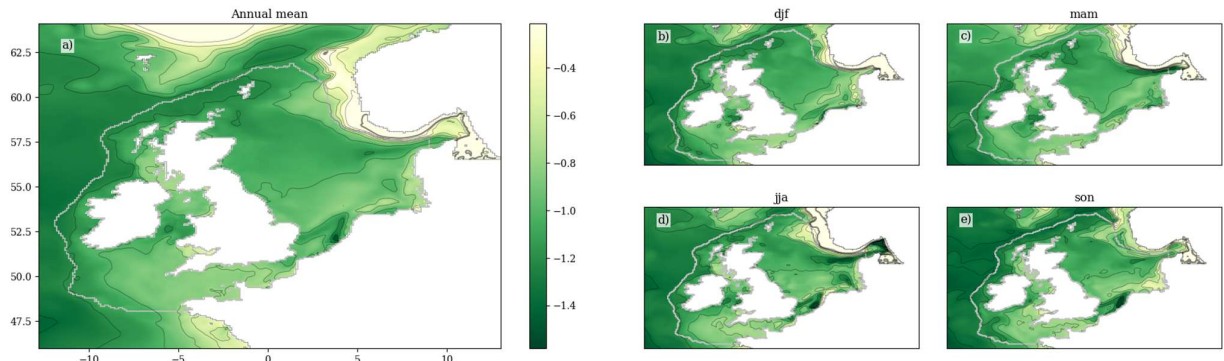

**Figure 13 The SSS ensemble mean change over the 21ˢᵗ Century (2079-2098 compared to 2000-2019), for the: a) annual mean; b) winter (December to February); c) spring (March to May); d) summer (June – August) and e) autumn (September – November). Black contours relate to the colour bar tickes, and the zero contour is a thicker black line. The coast line and NWS region are delinated with a grey line, and Full comparsion between the early and late century, and the components of variance (ensemble variance and interannual variability), see Figure S20.**


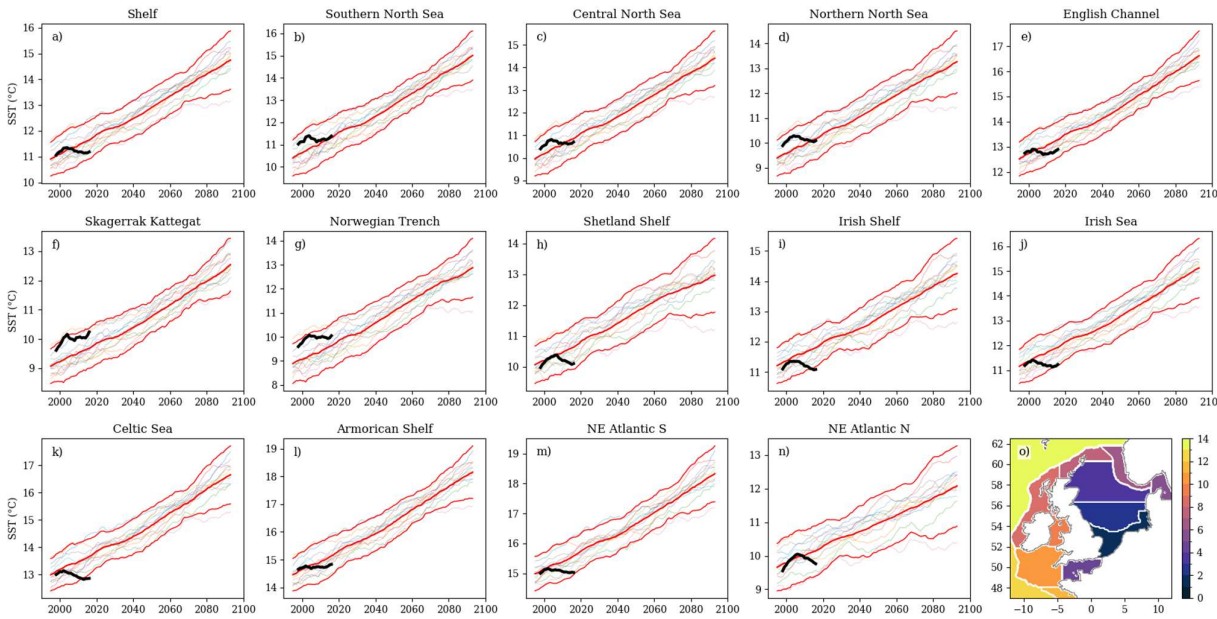

**Figure 14 Temporal evolution of the regional mean annual SST. Each region is given in a separate sub-panel (a-n), with o) showing the NWS region mask. Each ensmble member is filtered with a 5-year low-pass filter (shown faintly), with the ensemble mean, and ± 2 standard deviations shown in red (both based on the filtered ensemble members). The Copernicus Marine RAN reanalysis regional mean SST (also filtered) is shown in black for comparision. The y-axis limits change with each region.**


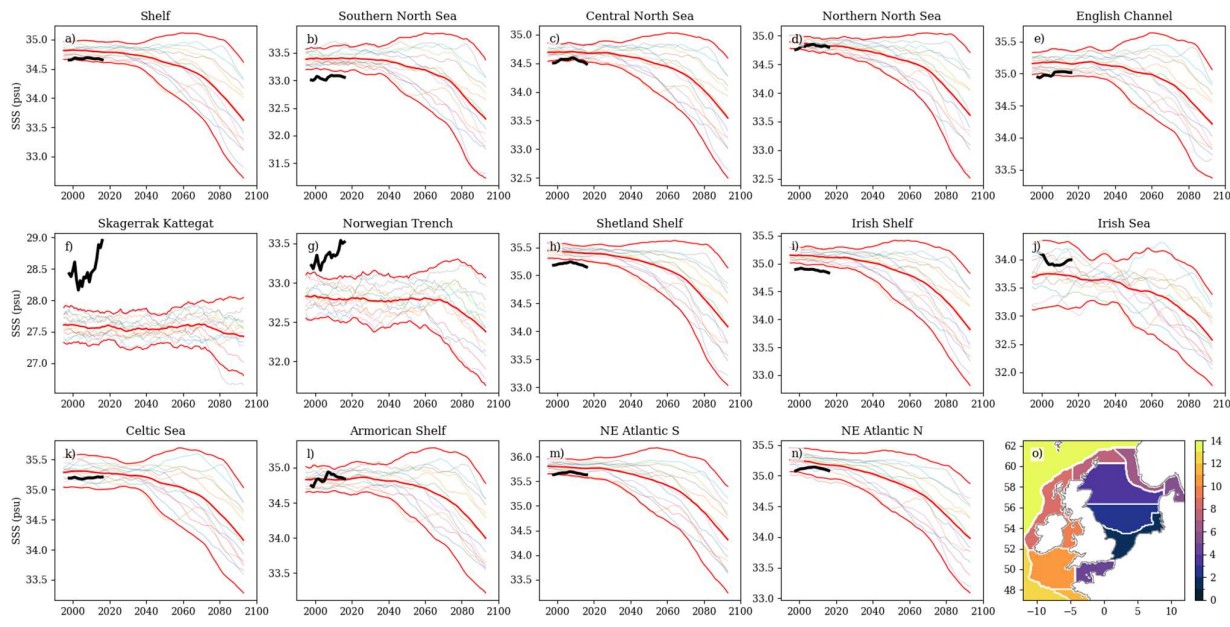


**Figure 15 Temporal evolution of the regional mean annual SSS. Each region is given in a separate sub-panel (a-n), with o) showing the NWS region mask. Each ensmble member is filtered with a 5-year low-pass filter (shown faintly), with the ensemble mean, and ± 2 standard deviations shown in red (both based on the filtered ensemble members). The Copernicus Marine RAN reanalysis**

**regional mean SST (also filtered) is shown in black for comparision. The y-axis limits change with each region.**

.

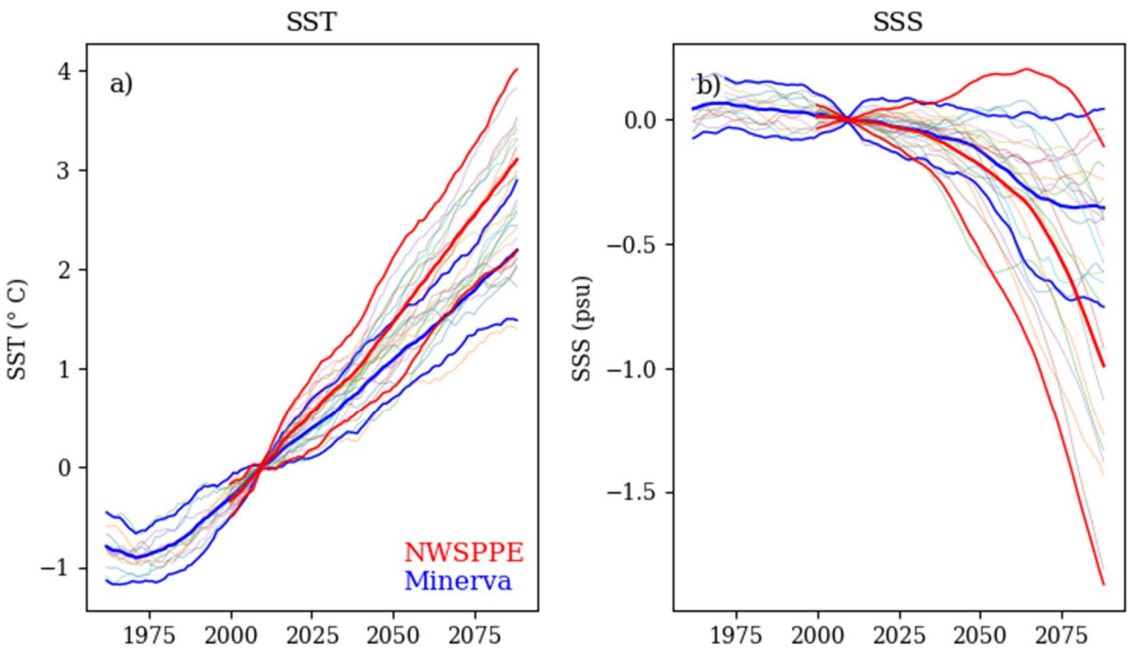


**Figure 16 Comparison of a) the SST anomaly and b) SSS anomaly of the current NWSPPE ensemble (red) with the Minerva Projections (blue), for the NWS. Each ensemble member is filtered, and the 2000-2019 mean is removed, to give the anomaly. Note the current ensemble is run under rcp85, while Minerva was run under SRES A1B.**

## 15    Appendix A: Dataset Description

The data underlying these NWSPPE projections, with the PDCtrl data, have been released on the CEDA website (https://catalogue.ceda.ac.uk/uuid/832677618370457f9e0a85da021c1312). Here we will describe these files, the data structure, and the available variables. For the NWSPPE we provide monthly means of two-dimensional variables for every ensemble member, month, and year between 1990 and 2098. We also provide climatological means and standard deviations for each ensemble member, for an early and late century period (2000-2019 and 2079-2098). We use these climatologies to provide ensemble statistics, which are the basis of many of our figures. We also provide regional mean timeseries.

NEMO is discretised onto an Arakawa "C" grid, and so separates variables onto a T, U and V grid. Most scalar variables on the T grid (the centre of the grid boxes). The U and V grids are offset between the T points (at the edges of the grid boxes). Therefore, variables on the T grid have a different location (in terms of longitude and latitude) compared to their equivalent U and V grid. We use this distinction and have three sets of files, with most variables in the T grid files, and only the U and V components of the barotropic current in the U and V grid files. We expect the T grid files to be sufficient for most purposes and applications. We also calculate the barotropic current magnitude on the T grid (reported in the T grid files) – this gives users a first look at the circulation.

### 15.1    Directory structure

Here we describe the directory structure. At the highest level, the NWSPPE data (Tinker, 2023b) is separated from its resulting ensemble statistics (Tinker, 2023a), and the PDCtrl data (Tinker, 2023c). Under the NWSPPE data, there are sub-directories for each ensemble member, each of which have sub-directories for the annual files (`annual`), the regional mean files (`regmean`) and the climatologies (`clim`). PDCtrl has sub-directories for the annual files (`annual`) the regional mean files (`regmean`):

```
NWSClim

1)  NWSPPE
    a)  r001i1p00000
        i)   annual
        ii)    regmean
        iii)   clim
        (repeated for all 12 NWSPPE ensemble members)

    b)  …
2)  EnsStats

3)  PDCtrl
    a)  annual
    b)  regmean
```

### 15.2    File names

Here we give example file names to help explain our naming convention.

**NWSPPE/r001i1p00000/annual**

- `NWSClim_NWSPPE_r001i1p00000_1990_gridT.nc`
- `NWSClim_NWSPPE_r001i1p00000_1990_gridU.nc`
- `NWSClim_NWSPPE_r001i1p00000_1990_gridV.nc`

For the T, U and V grid files respectively, where the ensemble number and year change.

**NWSClim/NWSPPE/r001i1p00000/regmean**

- `NWSClim_NWSPPE_r001i1p00000_1990-2098_regmean.nc`

For the regional mean files. The regional mean files give monthly means for the whole period.

**`NWSClim/NWSPPE/r001i1p00000/clim`**

- `NWSClim_NWSPPE_r001i1p00000_clim_xxx_YYYY-YYYY_gridT_mean.nc`
- `NWSClim_NWSPPE_r001i1p00000_clim_ xxx_YYYY-YYYY_gridT_stddev.nc`

For the climatological mean and standard deviation respectively on the T grid – again the U and V grid variables are in separate files. The xxx denotes the monthly (m01-m12 for January to December), seasonal (djf (December to February), mam (March to May), jja (July to August), son (September to November) for winter, spring, summer
or autumn respectively), or annual means (ann), and the YYYY-YYYY denotes the climatological period, and may be 2000-2019 or 2079-2098.

We note that the variable names are unchanged (SST in one file is the 20-year mean, and in the other the 20-year standard deviation). The long_name variable attribute is also unchanged, but the method is captured in the cell_method.


**`NWSClim/PDCtrl/annual`**

- `NWSClim_PDCtrl_1990_gridT.nc`
- `NWSClim_PDCtrl_1990_gridU.nc`
- `NWSClim_PDCtrl_1990_gridV.nc`

The PDCtrl monthly mean files are analogous to the NWSPPE files described above.

**`NWSClim/PDCtrl/regmean`**

- `NWSClim_PDCtrl_2050-2250_regmean.nc`

The NWSPPE regional mean files are analogous to the NWSPPE files described above.

**`NWSClim/EnsStats`**

- `NWSClim_NWSPPE_EnsStats_clim_xxx_YYYYYYYY_gridT_stats.nc`

For the ensemble statistics (on the T grid, U and V grid variables are in separate files), where xxx again denotes monthly, seasonal or annual means. While the ensemble statistics are produced for the same climatological period (i.e. YYYYYYYY can be 2000-2019 or 2079-2098), they are also produced for the difference between the periods, so YYYYYYYY can also be 2079-2098minus2000-2019.

Here the file contains 4 statistics per variable, and so the variable names are modified to give the statistic name – SST_ensmean, SST_ensvar, SST_intvar, SST_ensstd. The long name and cell methods also reflect the statistic.

### 15.3   Variables

The variables that we have released include sea surface (SS-), near bottom (NB-) Temperature (T) and Salinity (S), and their difference (DF-) (SST, NBT, DFT, SSS, NBS, DFS), Potential Energy Anomaly (PEA), Mixed Layer Depth (MLD) and
Sea Surface Height (SSH), and the U and V components of the barotropic (depth mean) velocity (DMU and DMV), and barotropic (depth mean) speed (DMUV). These are all on the T grid, apart from DMU and DMV on the U and V grid respectively.

The regional mean timeseries include RegAveSST, RegAveNBT, RegAveDFT, RegAveSSS, RegAveNBS, RegAveDFS, RegAveSSH and RegAvePEA (the regional means of SST, NBT, DFT, SSS, NBS, DFS, SSH and PEA respectively) - all the
T grid variables, apart from DMUV and MLD.

Additional variables and frequencies may be available to collaborators, contact the author for details.

### 15.4 File structure

Here we describe the monthly mean fields, climatologies and ensemble statistics available on the CEDA data centre.

The released dataset covers the period between 1990 and December 2098 and is in CF-compliant netCDF files (CF-convention 1.8).

#### 15.4.1 Annual (two-dimensional fields)

Monthly mean data is available for all months from January 1990 to December 2098, for all 12 ensemble members, and the present-day control simulation from 2050-2250 (each year still representing the year 2000). For each year, there are three files for the model T, U and V grid, with 12 monthly means within each. All variables are on the T grid, apart from the U and V components of the barotropic velocity (on the U and V grid respectively). Only 2D data fields are being released, with 3D variables such as Temperature and Salinity being reduced to surface (SS-) and near-bed fields (NB-), and the difference between them (DF-).

Details of the files are given in Table A1, in terms of the netCDF variable names and their dimensions. The dimensions of the files are "time", "lon", "lat" and "bnds", (simplified to t, y, x and 2 respectively in Table A1). Time is given in seconds since 1950-01-01 00:00:00, using a 360_day calendar. The time variable (time) is supplemented by the time bounds (time _bounds), giving the start and end of the averaging period (hence having two dimensions). The data 2D data fields are references with the longitude (lon) and latitude (lat) variables in each of the files – these are slightly offset between the T, U and V grids, although this is unlikely to cause any difficulty in most applications. Time is given in as double-precision ("f8") floating point numbers, whereas other variables are given as single precision ("f4") floating point numbers.

#### 15.4.2 Climatologies

The monthly mean files are used to produce 20-year early century (2000-2019) and late century (2079-2098) climatologies, giving the mean and standard deviation in separate files. These are produced for annual means (ann), seasonal means (djf, mam, jja, and son) as well as monthly means (m01-m12 for January to December). The netCDF files have the same format as outlined in Table A1. The time_bounds variable sets the start and end of the climatology. For climatologies with an even number of years, the average time can be misleading, especially for monthly or seasonal climatologies where, for example, a winter climatology time_counter will be for summer. However, if you think of 2 sequential winters, the average time (in seconds since 00:00 1/1/1950) will be the summer between them – this is an unfortunate, if correct, feature of climatologies with an even number of years.

#### 15.4.3 Ensemble Statistics

These climatologies are used to produce the ensemble mean, ensemble variance and standard deviation and interannual variance. As the NWSPPE is designed to capture the likely uncertainty (spread) associated with uncertain parameters, rather than to give the most likely outcome, the ensemble mean is not the most likely value. However, the ensemble mean and the ensemble standard deviation, are a useful way to summarise the (typically Gaussian) distributions. While the ensemble standard deviation is simply the square root of the ensemble variance, we think it will be a more useful statistic, so we include it to save end users having to calculate it. We also give the interannual variability, following Tinker et al. (2016). These are given for the near-present day (2000-2019) and future period (2079-2098). For the difference between the periods (2079-2098minus2000-2019) the statistics are simply the difference between the future and present-day statistics. This is useful to show how the ensemble has changed (i.e. how the ensemble mean and ensemble standard deviation have changed), however, this does not give information on the uncertainty associated with a projected change. We therefore provide two additional statistics in the ensemble statistics difference files, the projected ensemble mean (projensmean) and the projected ensemble standard deviations. Here, we remove the present-day climatological mean from the future climatological mean to give a resulting anomaly ensemble – we then calculate the ensemble mean and standard deviation. This is useful when we want to say how the SST has increased, with an estimate of the uncertainty on the projection, for example when we say the southern

North Sea winter SST increased by 3.55 °C (±1.20 °C), we are taking winter southern North Sea regional mean of SST_projensmean (3.55 °C) and (2x) SST_projensmean (1.20 °C).

We consider the late-century climate change by comparing the early-century (2000-2019) to the late-century (2079-2098), for monthly, seasonal, and annual means. These time slices have been updated since UKCP09 and the Minerva projections, partly to make them closer to the present, but also as the simulations started in 1980 rather than 1952.

Following Tinker et al. (2016) we use two times the standard deviation ($2\sigma$) to describe the ensemble spread (average $\pm$ x, where x = $2\sigma$). We calculate $2\sigma$ for each grid box, for a given month (and season and year) across each of the 20-year time periods and plot these in Figure 9-Figure 13. The typical interannual variability, however, is often greater than the ensemble time mean spread. Tinker et al. (2016) developed a more sophisticated analysis of the ensemble variability by decomposing of the total variance ($\sigma^2_{tot}$) into an interannual variability component ($\sigma^2_{int}$), which is the mean of the 20-year interannual variances for each ensemble member, and an ensemble spread component ($\sigma^2_{ens}$), which is the variance of the 20-year means for the ensemble:

$$\sigma^2_{int} = \frac{\sum_{e,y}\left((x_{e,y})^2\right)}{n_e n_y} - \frac{\sum_e\left(\left(\frac{\sum_y x_{e,y}}{n_y}\right)^2\right)}{n_e} \qquad (\text{A1})$$

$$\sigma^2_{ens} = \frac{\sum_e\left(\left(\frac{\sum_y x_{e,y}}{n_y}\right)^2\right)}{n_e} - \left(\frac{\sum_{e,y} x_{e,y}}{n_e n_y}\right)^2 \qquad (\text{A2})$$

$$\sigma^2_{tot} = \sigma^2_{int} + \sigma^2_{ens} \qquad (\text{A3})$$

$$= \frac{\sum_{e,y}\left((x_{e,y})^2\right)}{n_e n_y} - \frac{\sum_e\left(\left(\frac{\sum_y x_{e,y}}{n_y}\right)^2\right)}{n_e} + \frac{\sum_e\left(\left(\frac{\sum_y x_{e,y}}{n_y}\right)^2\right)}{n_e} - \left(\frac{\sum_{e,y} x_{e,y}}{n_e n_y}\right)^2$$

$$= \frac{\sum_{e,y}\left((x_{e,y})^2\right)}{n_e n_y} - \left(\frac{\sum_{e,y} x_{e,y}}{n_e n_y}\right)^2$$

where this decomposition uses the sum of squares formula and where $x_{e,y}$ denotes a variable (such as SST) for a given year $y$, and ensemble member $e$, $n_y$ is the number of years in the sample (20) and $n_e$ the number of ensemble members (12). For each variable in the monthly means, there are now three ensemble statistics, so rather than produce three files, we have added a suffix to the variable name.

### 15.4.4 Regional Means

Maps of climatological means are good at showing spatial patterns and changes, but do not show how the system evolves temporally. Regional mean timeseries show how the system evolves, how the ensemble spread changes and also give an estimate of the NWS response to model parameter uncertainty. They also allow a first look at the data, to help inform a more detailed analysis. We follow the methodology of Tinker et al. (2019) by averaging the model fields over a relevant region mask every time step. We use the region mask of Wakelin et al. (2012) as it divides the NWS into regions that make geographic an oceanographic sense. We also include a "shelf" region, by combining several NWS regions (excluding the Atlantic, Norwegian Trench, Skagerrak/Kattegat and Armorican shelf regions) – when we refer to a NWS mean, or averaged over the NWS, we refer to this "shelf" region. NEMO calculates these regional means every time-step and then averages them into the monthly mean timeseries (hourly means are also available) which are released for SST, NBT, DFT, SSS, NBS, DFS, PEA, and SSH. We also include the Wakelin et al. (2012) region mask in the file (mask). The time, time_bounds, lat and lon (for the mask) variables are also in other files. The variables are prefixed by RegAve (e.g. RegAveSST, RegAveSSS, RegAvePEA), and have two dimensions – time and region. Two one-dimensional variables reg_id and cnt specific to region mean files are included reg_id and cnt. reg_id is the region id, and is a value from 0 to 13 – this aligns with the region names (given in the global attribute region_names). cnt is a count of the number of grid boxes within each region (ranging from 792-25523)

One difference in the regional means from NEMO 3.6 and NEMO 4.0.4 (used in PDCtrl and NWSPPE respectively) is that NEMO3.6 simply averaged the values of all the grid boxes in a region, while NEMO 4.0.4 takes an area weighted average. For a direct comparison between the two datasets, it is fairly easy to reprocess the regional means from the annual files, with or without area weighting. Example code is provided in the GitHub python package.

### 15.5 Sea Surface Height (SSH)

Care must be taken when using the projected SSH – this is not to be used directly as a sea level projection. Sea level is affected by a wide range of processes, and so sea level projections are built up from several different models (Fox-Kemper et al., 2021). Furthermore, given the different resolutions between HadGEM3-GC3.05 and COx, the different sources of lateral boundaries for the Atlantic and Baltic, the absences of tides in the GCM, the SSH in COx is likely to diverge from HadGEM3-GC3.05. However, COx improves the representation of coastal ocean processes, and so does give additional information on sea level

change (e.g. Hermans et al., 2020a, b; Tinker et al., 2020), but this is only part of the Sea Level puzzle (Palmer et al., 2018). The SSH data provided here is obtained from downscaling the HadGEM3-GC3.05 simulation of ocean Dynamic Sea-Level (DSL). DSL is defined as the local height of the sea-surface above the geoid and is corrected to remove time-dependent variations of the sea-surface due to atmospheric loading referred to as the inverse barometer effect. In the driving model (HadGEM3-GC3.05) DSL is defined to have zero global ocean area mean, therefore in the downscaled simulations the SSH

data is providing the local anomaly in geocentric sea-level that results from local ocean density (steric) and circulation (dynamic) effects. While COx does simulate tides and the inverse barometer effect, its representation of SSH change include only part of the drivers of sea-level change active in the real world, which we briefly outline below.

Sea level change arises from process that alter the total volume of water in the ocean column or processes that alter the mass of water by redistributing water mass between the oceans and the continental land-surface. Simulated changes to ocean density

from climate models can be used to estimate the effect of changes in ocean volume. However, many important processes that alter ocean mass, notably the contributions from loss of land-ice mass are not included in climate models and estimates for these process must be supplied from offline models. Additionally, changes to distribution of water mass in the oceans and over the land surface will produce spatially varying patterns of sea-level change associated with the Earth's gravitational field, axial-rotation, and deformation of solid-earth surface. The effect of the Gravitation, Rotation and Deformation process referred to

collectively as GRD must also be estimated using offline models. Finally local sea-level projections for a coastal location may refer to changes relative to a local solid surface rather (relative sea-level change) than a fixed geoid (geocentric sea-level change) and use additional models to account for local vertical movement of the land surface (e.g. subsidence). These disparate sources are pulled together into the Sea Level puzzle (Palmer et al., 2018).

### 15.6 Missing Data

Over the 110 years, 12 months and 12 ensemble members, there were 23 files that were not archived correctly. Most of these were recreated by averaging the daily mean files giving near identical files (with the difference at the noise level). 3 files (Table A2) could not be created this way, and so were replaced by averaging the monthly mean for the same month from the previous and following year. The approach should maintain the seasonal cycle, and any trends in the data, however, may impact some statistics, such as estimates of interannual variability. Full details of the replacement

files are given in Table A3 and Table A4.

### 15.7 Calculation of the barotropic current speed on the T grid (DMUV)

The use of model currents can often be complicated, especially as the U and V grids are offset. Often maps of current speed are useful, as they give an illustration of circulation. We have therefore provided the barotropic current magnitude on the T grid. First, we interpolated the U and V components of the barotropic current on the T grid:


$$DMU\_T_{j,i} = \frac{DMU_{j,(i-i)} + DMU_{j,i}}{2}$$

( A4 )

$$DMV\_T_{j,i} = \frac{DMV_{j,i} + DMV_{(j-1),i}}{2}$$

Where DMU and DMV are the U and V components of the barotropic current on their native U and V grids, and once they have been converted onto the T-grid they become DMU_T and DMU_T respectively. We then calculated their magnitude (DMUV)


$$DMUV_{j,i} = \sqrt{DMU\_T_{j,i}^2 + DMU\_T_{j,i}^2}$$

( A5 )

### 15.8   Code

This post-processing was undertaken in python, and is available from GitHub as:

https://github.com/hadjt/NWS_simulations_postproc

### 15.9   Digital Object Identifier (DOI)

Each of the top-level datasets has been issued with a DOI.

Physical Marine Climate Projections for the North West European Shelf Seas: NWSPPE. 20 July 2023.

https://dx.doi.org/10.5285/edf66239c70c426e9e9f19da1ac8ba87

Physical Marine Climate Projections for the North West European Shelf Seas: PDCtrl. 20 July 2023.

https://dx.doi.org/10.5285/66e39885a60e4b6386752b1a295f268a

Physical Marine Climate Projections for the North West European Shelf Seas: EnsStats. 20 July 2023.

https://dx.doi.org/10.5285/bd375134bd8c4990a1e9eb6d199cc723

## 15.10  Appendix A Tables


16    **Table A1 Details of NetCDF files.**

| Grid and File | NC variable name | NC variable Long-name | Data precision | Dimensions |
|---|---|---|---|---|
| T | time | Time | double | t |
| T | time_bounds | Time bounds | double | t,2 |
| T | lat | Latitude | float | y |
| T | lon | Longitude | float | x |
| T | SSH | Sea Surface Height above Geoid | float | t,y,x |
| T | SST | Sea Surface Temperature | float | t,y,x |
| T | SSS | Sea Surface Salinity | float | t,y,x |
| T | NBT | Near Bed Temperature | float | t,y,x |
| T | NBS | Near Bed Salinity | float | t,y,x |
| T | DFT | Difference between Sea Surface and Near Bed Temperature | float | t,y,x |
| T | DFS | Difference between Sea Surface and Near Bed Salinity | float | t,y,x |
| T | PEA | Potential Energy Anomaly | float | t,y,x |
| T | MLD | Mixed Layer Depth using Kara approach | float | t,y,x |
| T | DMUV | Barotropic current speed on T grid | float | t,y,x |
| U | time | Time | double | t |
| U | time_bounds | Time bounds | double | t,2 |
| U | lat | Latitude | float | y |
| U | lon | Longitude | float | x |
| U | DMU | Eastward Ocean Barotropic current | float | t,y,x |
| V | time | Time | double | t |
| V | time_bounds | Time bounds | double | t,2 |
| V | lat | Latitude | float | y |
| V | lon | Longitude | float | x |
| V | DMV | Northward Ocean Barotropic current | float | t,y,x |

1765

**18    Table A2 Files missing from the NWSPPE, that we recreated by averaging adjacent files.**

| Ensemble Member | Date | File type |
|---|---|---|
| **r001i1p02242** | 205105 | U grid Month Mean |
| **r001i1p02868** | 205111 | T grid Monthly Mean |
| **r001i1p02832** | 205006 | Regional mean timeseries |

1770

23    **Table A3 Monthly Mean 2 dimensional files that were not archived, and how they were replaced: "Daily Means" the daily mean files were archived, and have been averaged into monthly means.**

| Ensemble Member | Date | Grid | Replacement averaged from |
|---|---|---|---|
| **r001i1p00605** | 205107 | T | Daily Means |
| **r001i1p00605** | 205008 | U | Daily Means |
| **r001i1p00605** | 208403 | V | Daily Means |
| **r001i1p00834** | 205102 | T | Daily Means |
| **r001i1p00834** | 205103 | T | Daily Means |
| **r001i1p01113** | 208409 | V | Daily Means |
| **r001i1p01935** | 205110 | T | Daily Means |
| **r001i1p02123** | 205010 | U | Daily Means |
| **r001i1p02123** | 208311 | V | Daily Means |
| **r001i1p02242** | 205105 | U | **Adjacent Years** |
| **r001i1p02242** | 208311 | U | Daily Means |
| **r001i1p02335** | 205009 | T | Daily Means |
| **r001i1p02868** | 205111 | T | **Adjacent Years** |
| **r001i1p02868** | 205112 | T | Daily Means |
| **r001i1p02868** | 205108 | U | Daily Means |
| **r001i1p02868** | 205006 | V | Daily Means |

1775

27    **Table A4 Regional Mean monthly mean files not archived.**

| Ensemble Member | Date | Replacement averaged from |
|---|---|---|
| **r001i1p00605** | 205008 | Daily Means |
| **r001i1p00834** | 205109 | Daily Means |
| **r001i1p01935** | 202905 | Daily Means |
| **r001i1p01935** | 202906 | Daily Means |
| **r001i1p02123** | 205010 | Daily Means |
| **r001i1p02491** | 205008 | Daily Means |
| **r001i1p02832** | 205006 | **Adjacent Years** |

## 28  Appendix A: Tides in 360-day Calendar

The dynamics of the NWS and other shelf seas are dominated by tides. They provide the mixing energy to seasonally mix the water column in winter and define which whether a region is mixed or stratified in the summer(Simpson and Bowers, 1981). They drive important aspects of the residual and instantaneous circulation (Tinker and Hermanson, 2021). They interact with the sediment and coastline leading to geomorphological processes (Pingree and Griffiths, 1979).

Tides are controlled by aspects of the orbits of the earth moon sun systems (Pugh, 1987). The tidal frequencies are based on 5 astronomical frequencies, including the tropical year, the sidereal day and the sidereal month (Pugh, 1987). Different combinations of these frequencies give the frequencies of the tidal constituents, such as $M_2$ and $S_2$ that dominate in most regions (Pugh, 1987). The addition of these terms, and the beating between them, lead to important behaviours, including the spring-neap cycle, and the equinoctial tides (Pugh, 1987).

Climate models use a variety of calendars. In addition to the Gregorian (365+leap years) calendar, other common calendars include the 365 day with no leap year calendar, and the 360-day calendar. The 360-day calendar is made up of twelve 30-day months which make post processing, and calculation of monthly and seasonal means much simpler. However, the 360-day calendar is not directly compatible with the astronomical constants of the real world. This is not very important from the climate modellers point of view, as most current climate models do not have tides - from a global climate perspective, realistic, dynamic tides are not particularly important, and their role can generally be performed by a mixing parameterisation (Tinker et al., 2022). It becomes a problem when climate models are dynamically downscaled for shelf seas regions (e.g. Holt et al., 2010; Tinker et al., 2016). As global climate models advance, tides will become increasingly integrated, and then this may also become a problem with GCMs.

When running a shelf seas model with a 360_day calendar, the default approach (used within NEMO) is to reset the tides at the beginning of every month, so the equinoctial tides always occur at the correct time of the year (hereinafter "reset", see Table B3). However, this introduces a jump in the tides as days are repeated of skipped (for non-30-day months), which could introduce noise into the system, and renders tidal analysis (based on least squares multiple linear regression) useless. Another approach is to convert the 360-day model forcings into 365 and to make use of the standard Gregorian calendar. This is typically achieved by repeating days, or interpolating (through time) to stretch the forcings. Both have drawbacks, including affecting the speed of synoptic systems. Another approach is to run the tides continuously from the start of the run (on their Gregorian calendar) and accept that they will become out of sync with the model running on the 360_day calendar, gaining ~5 days every year (hereinafter "drift", see Table B3). This means that the equinoctial tides drift through the year. A final approach is to adjust the astronomical frequencies, and so the tidal constituent frequencies, to match a 360-day year, which correctly fixes the equinoctial tides to a point in the year (hereinafter "compress", see Table B3). Each of these approaches have strengths and weaknesses, which may influence the simulations. Here we describe the three tidal implementations (reset, drift, and compress) and their effect on the tidal characteristics of the NWS. We do not assess the modification of the atmospheric and oceanic forcings.

We first give brief overview of relevant aspects of tidal theory, how the astronomical constants may be adjusted for the 360-day year, and the impact of this on the tidal constituent periods, and the important beating frequencies. We then assess the impact of this tide in a model simulation, and compare to the other methodologies.

For the NWSPPE and PDCtrl, we take a conservative approach and use the "drift" method, following Holt et al., (2010) and Tinker et al., (2016).

## 28.1 Method

 ### 28.1.1 Tidal theory

Here we give a brief overview of relevant aspects of tidal theory, mainly taken from Pugh (1987).

Harmonic expansion of the tide breaks it into many sinusoidal constituents with astronomically defined frequencies, and amplitudes and phases that vary with space. For a given location, the effect of the tide on the sea level can be defined as:

$$\eta = \sum \; a_n \sin(\sigma_n t + \; \phi_n) + \eta_0 \qquad \text{( B1 )}$$

Where a $\eta$ is the sea level (and $\eta_0$ is the mean sea level), and for constituent $n$, $a_n$ is the amplitude, $\sigma_n$ is the frequency ($2\pi/T$ where $T$ is the tidal period), and $\phi_n$ is the phase. The $a_n$ and $\phi_n$ can be found with least squares multiple linear regression (Pugh, 1987).

There are a number of important astronomical frequencies that define the frequencies of the tidal constituents: the mean solar day ($\omega_0 = 2\pi/1$ day $= 2\pi/24$hr); the mean lunar day ($\omega_1 = 2\pi/1.0351$ day $= 2\pi/24.8424$hr); the sidereal month ($\omega_2 = 2\pi/27.3217$ day $= 2\pi/655.7208$); the tropical year ($\omega_3 = 2\pi/365.2422$ day $= 2\pi/8765.8128$ hr); moon's perigee ($\omega_4 = 2\pi/8.85$ years); the regression of the moon's nodes ($\omega_5 = 2\pi/18.61$ years); and the perihelion ($\omega_6 = 2\pi/20942$ years).

The tidal constituent frequencies ($\omega_n$) can be calculated as:

$$\omega_n = i_1\omega_1 + i_2\omega_2 + i_3\omega_3 + i_4\omega_4 + i_5\omega_5 + i_6\omega_6 \qquad \text{( B2 )}$$

where $i_1$, $i_2$, $i_3$... are small integers ($|i|<5$). The use of both the solar and lunar days is not necessary as $\omega_0 = \omega_1 + \omega_2 - \omega_3$. The species of the constituent is defined by $i_1$ ($i_1 = 2$ semi-diurnal, $= 1$ diurnal; 0, long period) (Pugh, 1987). Doodson notation concisely describes the tidal constituents: $i_2$-$i_6$ are incremented by 5 (so there are no negative integers); $i_1$-$i_6$ are combined into a single number. For example, $M_2$ has a Doodson number of 255.555 ($i_1 = 2$, $\omega_n = 2\omega_1$), $S_2$ is 273.555 ($i_1 = 2$, $i_2 = 2$, $i_3 = -2$, $\omega_n = 2\omega_1 + 2\omega_2 - 2\omega_3$), and $K_2$ is 275.555 ($i_1 = 2$, $i_2 = 2$, $\omega_n = 2\omega_1 + 2\omega_2$),

When two tidal constituents of slightly different frequencies are added together, they beat as they come in and out of phase. The $M_2$ and $S_2$ constituents beat to give the spring neap cycle, and the $S_2$ and $K_2$ constituent beat to give the equinoctial tides. The beat envelope function is:

$$\pm\left[a_1 + \; a_2\left[\cos\left((\sigma_1 - \sigma_2)t + \; (\phi_1 - \phi_2)\right)\right]\right] \qquad \text{( B3 )}$$

where the constituents are ordered by amplitude size (i.e. $a_1 > a_2$). This gives the beat frequency ($\sigma_1 - \sigma_2$) and phase ($\phi_1 - \phi_2$), so the $M_2$-$S_2$ spring neap cycle has a frequency of $2\pi/12.42 - 2\pi/12 = 354.85$ hr $= 14.7$ days, and the $S_2$-$K_2$ has a beat frequency of 182.6221 days.

You can also calculate the beat frequencies from the Doodson numbers. For example, the difference between $M_2$ and $S_2$ is 255.555 and 273.555 implies a difference in frequency of ($\omega_n = 2\omega_2 - 2\omega_3$), while $S_2$ and $K_2$ is 273.555 and 275.555 is ($\omega_n = -2\omega_3$).

### 28.1.2 "Compress" tidal scheme.

We introduce a tidal scheme where one of the astronomical frequencies modified to the 360-day year. The frequency of the tropical year ($\omega_3$) is modified to $2\pi/360$days $= 2\pi/8640$hr. This has no impact on the $S_1$ or $S_2$ tidal frequencies but reduces the $M_2$ period by 1.84 seconds. The impact on the beat frequencies is larger, with the spring neap cycle increasing from 14 days 18hr 22 mins to 14 days and 18 hr 47 mins, and the equinoctial tidal frequencies decreasing from 182.621 days to 180 days, exactly half a 360-day year.

We have implemented this scheme in the shelf seas model NEMO Coastal Ocean model 9 (CO9, NEMO version 4.0.4), and will assess the impact of this of the simulation of the tides. Note that we have not used this scheme in the NWSPPE.

### 28.1.3    Models and Experimental Design

We use the Nemo version 4.04, Coastal Ocean model version 9 (O'Dea), run on the 7km NWS domain (Atlantic Margin Model 7km - AMM7), in a set of 40-year simulations with the different approaches to modelling the tide (see Table B5). We use these simulations to investigate the impact of the different tidal schemes on the resulting emergent tidal behaviour. We assess these runs for their amphidromic systems, and distribution of amplitude and phase of the different tidal constituents. We also pay particular attention to the difference in amplitude and phase between key constituent pairs, as this controls the behaviour of

their tidal beats, which is a particular weakness of some of the tidal schemes We do not assess the impact on other oceanic properties (such as temperature),which is likely to be more subtle, and require longer simulations to investigate.

We assess the second 20-year period is used in this assessment, as this ties in with the inline tidal harmonic analysis within NEMO (within DIA/diaharm_fast.F90, within https://github.com/hadjt/NEMO_4.0.4_CO9_shelf_climate/tree/master/src/OCE/).

### 28.1.4    NEMO model code

Most of the code added to NEMO is within SBC/tide_mod.F90 (https://github.com/hadjt/NEMO_4.0.4_CO9_shelf_climate/blob/master/src/OCE/SBC/tide_mod.F90). This allows the user to choose between the three tidal schemes using the `nam_tides360` namelist, and the `ln_tide_drift` and `ln_tide_compress` logical keywords.

### 28.1.5    Analysis techniques: Least Squares tidal harmonic analysis

We use a tidal harmonic analysis based on a least-squares method to calculate the phase and amplitudes of the tidal constituents as described by Pugh (1987). This method considers the tidal signal as:

$$T(t) = Z_0 + \sum_N H_n f_n cos\left[\sigma_n t - g_n + (V_n - u_n)\right]$$

( B4 )

where the $Z_0$ is the mean sea level, $H_n$ and $f_n$ are the amplitude and the nodal factor for the $n^{th}$ tidal constituent, $t$ is time $\sigma_n$ is the tidal frequency $g_n$ is the tidal phase, and $V_n$ and $u_n$ are the phase angle at time zero and the nodal angle respectively. $t$ and

the terms $f_n$, $\sigma_n$, $V_n$ and $u_n$ are known $a\ priori$, as is the surface elevation $T(t)$. ( B4 ) is solved for $Z_0$, $H_n$, $g_n$ by using a least-squares estimation.

This is done online by NEMO (see https://github.com/hadjt/NEMO_4.0.4_CO9_shelf_climate/blob/master/src/OCE/DIA/diaharm_fast.F90) using consecutive 20 year periods of the model simulation. We use the second 20-year period to avoid any spin-up issues.

One issue of the standard NEMO tide configuration ("reset") is that the time counter jumps at the end of 360-day calendar months for months that should have 31 days (or February). NEMO's online harmonic analysis code takes account of this, as both the tide, and the time counter jump. We have written python code that replicate the NEMO online harmonic analysis code. We can then to see how these jumps affect the tidal analysis, if not taken into account – i.e. if someone simply analysed the hourly sea surface height model output. This is used in Figure B2c (c.f. Figure B2b) which shows very large Fis, and (not

shown) a near zero $M_2$ amplitude, and larger $S_2$ amplitude.

## 28.2    Results

### 28.2.1    Harmonic analysis

We fit a least square tidal fit to the data (Figure B1). We use the RMSE of the residuals to quantify how well the data is fit (Figure B2). As the tides see discontinuous time for Reset, with days repeated and missing - we carefully recreate the time

array for its tidal analysis, otherwise the most diurnal constituents (including $M_2$) appear to be $S_2$. We include the RMSE when

this is not done, see Figure B2c, which shows an ~ 10-fold increase in RMSE. Generally, all models have similar RMSE, which are small compared to the tidal range, and so we can use the computed phase and amplitudes for the rest of the study. The locations of the amphidromes are insensitive to the simulation (Figure B3). The difference of the amplitudes between the simulation is typically less than 5%. The phases of the Compress experiment have the largest difference.

The phase of individual diurnal and semidiurnal constituents is not thought to be of any particular importance as the peak will move through the day (apart from $S_1$, $S_2$ – hence spring HW is always at the same time). However, the phase difference between constituents give the phase of the tidal beats, which can be important, and can occur at the same time of the year.

### 28.2.2 Important beat frequency and amplitude

To identify the important beat frequency pairs, we use ( B3 ) to find the beat frequency of each pair of constituents, and their

amplitude ($a_2$ in ( B3 ), i.e. the smaller of the two amplitudes) for the shelf mean (Figure B4). We focus on beat pairs with periods greater than 100 days. Figure B4 shows that the $S_2$-$K_2$ pair are the dominant biannual beat pair, with $K_1$-$P_1$ playing a secondary role. The annual beat frequencies are much weaker, with the $S_2$-$T_2$ pair dominating, and $K_1$-$S_1$ and $P_1$-$S_1$ having a lessor *(and similar)* role. $K_2$-$T_2$ is triannual, while $N_2$-$NU_2$ and $2N_2$-$MU_2$ occur every 202 days and 10hr20min, and 202 days 13hr35min respectively.


Figure B4 and Table B6 show that the proposed "compress" tidal system correctly adjusts the beat frequencies to the 360-day year, however the phase of the beat is also important – it is what makes the equinoctial tides occur during the equinoxes. We can use ( B3 ) to find the beat phase (the difference between the individual tidal constituent phases) and calculate the timing of the peak of the beats - we use this to show how well the different tidal systems reproduce the phase of the beating waves.

Figure B5 shows that Gregorian and the reset have the same timing of the maxima. The drift can be seen to drift throughout the period, progressing ~5 days per year. Compress is comparable to Greg and Reset, showing that it is capable of reproducing the period and timing of the tidal beats. We can look at the maps of the spatial patterns of the timing of the beat pair maximum (Figure B6). For the most important pair ($S_2$-$K_2$), the patterns of the timing agree visually, and the timing is within ~10 days, which is considered reasonable.

### 28.3 Conclusions

We have compared the ability of different 360-day tidal schemes to simulate the emergent behaviour of the NWS tides from simulation with the standard Gregorian calendar. We base our analysis on the results of a least squares tidal harmonic analysis. This has to be modified for two tidal schemes (using the modified tidal periods for Compress, and using a synthetic time for Reset).

Reset is the standard tidal scheme within NEMO, but leads to jumps within the tide at the end of the month. This changes the average $M_2$ period, and so the errors associated with the (uncorrected) harmonic analysis are much higher (Figure B2c).

With the corrected the harmonic analysis, all 360-day tidal schemes simulate the co-tidal charts of the control (Greg) well (Figure B1) with a similar (pattern and magnitude of) RMS errors (Figure B2). This is the most important quality, and each scheme performs equally well.

The beat phases give the timings of the equinoctial tide. Both Reset and Compress simulate the correct timings, whereas the Beat Period doesn't match the 360-day calendar, so even if the phase is correct, the timings drift through the calendar each year, typically gaining 5 days every year.

For both these criteria, the Compress tidal scheme performs well, and no other scheme out performs it. However, it is unpublished, and some users may be impacted by some of its features ($M_2$ tide shortened by 2 second, Spring-Neap cycle

increases by 25 minutes). We therefore use the Drift tidal scheme in our projections, as it has been used in earlier projections (e.g. Holt et al., 2010; Tinker et al., 2016), and is more conservative choice.

## 28.4   Appendix A Tables

**Table B1 360-day tide methodologies, and their advantages and disadvantages.**

| Methodology | Advantages | Disadvantages |
|---|---|---|
| **Reset** | Default methodology in NEMO<br>Equinoctial Tides at correct time of the year. | Tidal boundaries are discontinuous at end of months that do not only have 30 days.<br>This, on average, significantly changes the period of the $M_2$ (and other) tidal constituents.<br>This renders standard tidal harmonics analysis ineffective. |
| **Drift** | Allows a continuous stimulation of the tides.<br>Doesn't modify the periods of the tidal constituents.<br>Standard Tidal analysis is applicable. | Equinoctial Tides occur ~5 days earlier every year. |
| **Compress** | Allows a continuous stimulation of the tides.<br>Standard Tidal analysis is applicable when using modified tidal constituent periods<br>Equinoctial Tides occur at correct time of the year. | Slightly modifies the periods of the tidal constituents, following the calculations described in this paper.<br>New methodology, not used in published studies. Impact of modifications on all end users not known. |
| **Modification of the atmospheric and lateral oceanic forcings to fit the Gregorian calendar (included here as a control)** | Tides are not modified at all. | Different methodologies well have different implications.<br>Temporal interpolation of the atmospheric conditions or skipping/repeating day will affect the passage of synoptic systems. This may affect surface momentum, heat and water fluxes, which could lead to systematic biases in the simulated climate.<br>In our Gregorian, we skip/repeat days. |

**Table B2 Period of updated tidal constituent**

| Constituent (Darwin symbol) | Species | Doodson number | Period | New Period | Difference |
|---|---|---|---|---|---|
| M$_2$ | Principal lunar semidiurnal. | 255.555 | 12.421 hr | 12.420 hr | -1.845 sec |
| S$_2$ | Principal solar semidiurnal. | 273.555 | 12.000 hr | 12.000 hr | 0.000 sec |
| N$_2$ | Larger lunar elliptic semidiurnal. | 245.655 | 12.658 hr | 12.658 hr | -1.916 sec |
| K$_2$ | Lunisolar semidiurnal. | 275.555 | 11.967 hr | 11.967 hr | -1.713 sec |
| K$_1$ | Lunar diurnal. | 165.555 | 23.934 hr | 23.934 hr | -3.426 sec |
| NU$_2$ | Larger lunar evectional. | 247.455 | 12.626 hr | 12.625 hr | -3.813 sec |
| O$_1$ | Lunar diurnal. | 145.555 | 25.819 hr | 25.818 hr | -3.987 sec |
| L$_2$ | Smaller lunar elliptic semidiurnal. | 265.455 | 12.192 hr | 12.191 hr | -1.778 sec |
| 2N$_2$ | Lunar elliptical semidiurnal second-order. | 235.755 | 12.905 hr | 12.905 hr | -1.992 sec |
| MU$_2$ | Variational. | 237.555 | 12.872 hr | 12.871 hr | -3.963 sec |
| T$_2$ | Larger solar elliptic. | 272.555 | 12.016 hr | 12.017 hr | 0.864 sec |
| M$_4$ | Shallow water overtide of principal lunar. | 455.555 | 6.210 hr | 6.210 hr | -0.923 sec |
| Q$_1$ | Larger lunar elliptic diurnal. | 135.655 | 26.868 hr | 26.867 hr | -4.317 sec |
| P$_1$ | Solar diurnal. | 163.555 | 24.066 hr | 24.067 hr | 3.464 sec |
| S$_1$ | Solar diurnal. | 164.555 | 24.000 hr | 24.000 hr | 0.000 sec |


**Table B3 Name and description of simulations**

| Name | Description |
|---|---|
| **Greg** | 360-day forcings are used to run a 360-day simulation. The last days of February forcings have been removed, and the forcings of the 30<sup>th</sup> have been repeated where necessary |
| **Reset** | The tides are reset to the Gregorian calendar at the beginning for every day (using the day, month, year, and allowing non-existent days to continue into the next month if necessary. This is the standard implementation within NEMO. A Least-Square's tidal analysis does not work directly, but can be used by converting the 360-day calendar dates to the Gregorian calendar (e.g. 25th Feb - 30th Feb and 1st - 2nd Jan – becomes 25th-28th Feb and 1st - 4th Jan.). |
| **Reset (uncorrected)** | The same as Reset, but without the correction to the dates into Gregorian calendar. |
| **Drift** | Run the Gregorian tides on the days since 1st January 1900, and then convert these to the 360-day calendar |
| **Compress** | Run with the updated tidal constituent periods. |


**Table B4 Periods of beat pairs**

| Constituent Pair | Gregorian Period (days) | 360-day Period (days) |
|---|---|---|
| $S_2$-$T_2$ | 365.26 | 360.01 |
| $K_1$-$S_1$ | 365.24 | 360.00 |
| $P_1$-$S_1$ | 365.25 | 360.00 |
| $S_2$-$K_2$ | 182.62 | 180.00 |
| $K_1$-$P_1$ | 182.62 | 180.00 |
| $N_2$-$NU_2$ | 205.89 | 202.57 |
| $2N_2$-$MU_2$ | 205.89 | 202.57 |
| $K_2$-$T_2$ | 121.75 | 120.00 |

## 28.5   Appendix B Figures

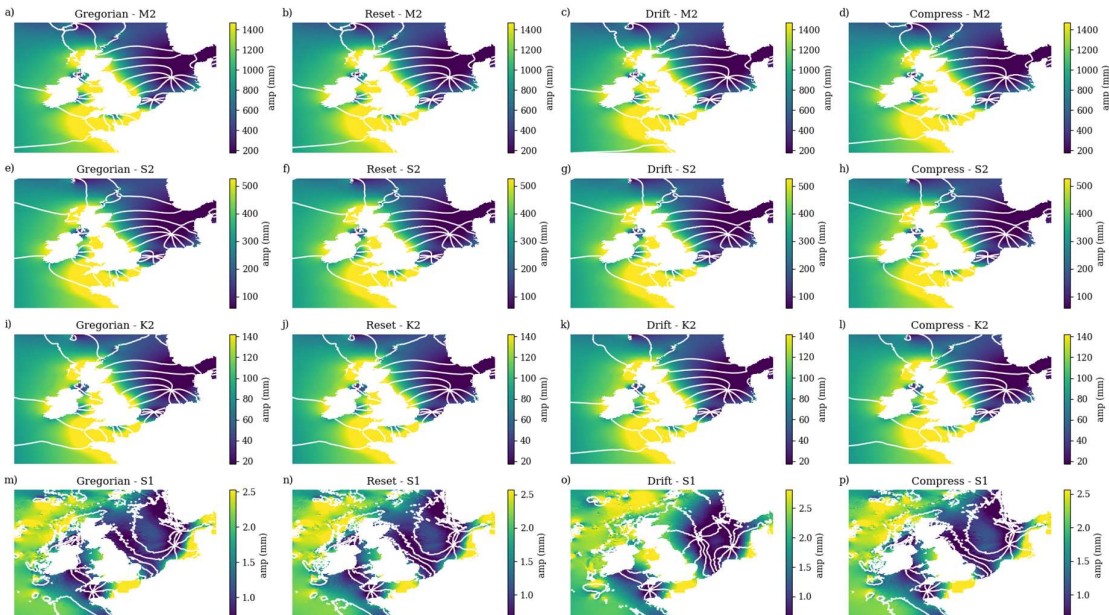

**Figure B1 Cotidal charts for the M₂ (a-d), S₂ (e-h), K₂ (i-l) and S₁ (m-p) tidal constituents under the Gregorian (control, a, e, i, m), Reset (b, f, j, n), Drift (c, g, k, o) and Compress (d, h, l, p) tidal schemes. The colour maps represent the amplitude, and is constant for a given constituent, while the lines are lines of equal phase, (every 45°).**

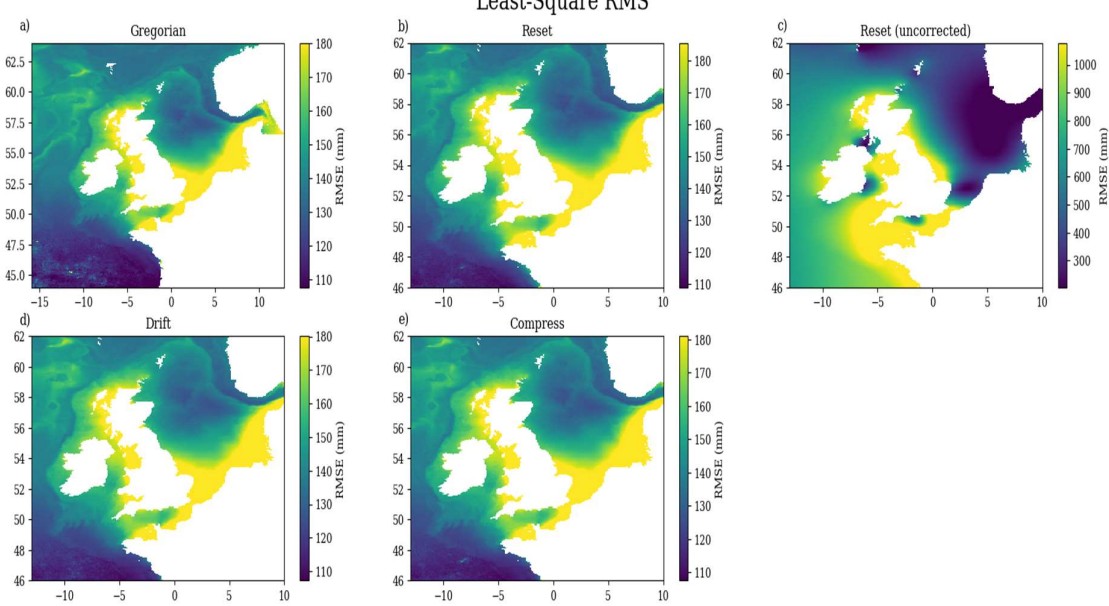

**Figure B2 RMS of least square tidal fit, for the simulations: a Gregorian; b Reset; c Reset (uncorrected); d Drift; and e Compress. For Reset, we construct an equivalent time (resetting to the Gregorian calendar at the beginning of each month) and use this for the least squares. We do not correct this in Reset (uncorrected), and the RMS errors are much greater (c).**

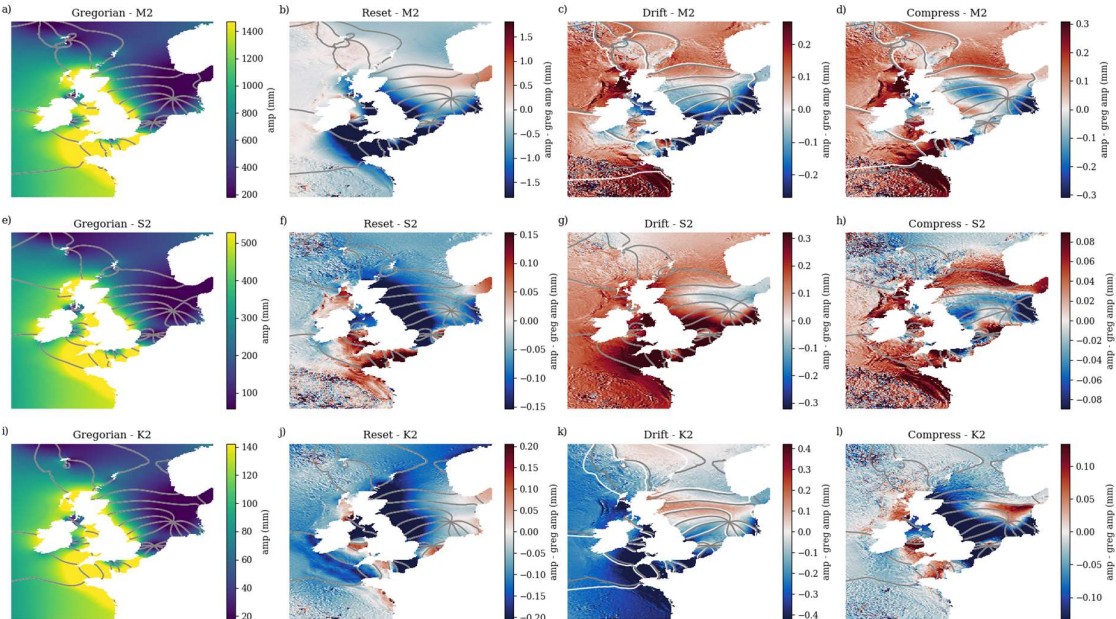

1970 **Figure B3 Cotidal charts for M₂, S₂ and K₂ (upper, middle, and lower row respectively), for Gregorian (left column) and anomalies (relative to Gregorian) for Reset, Drift and Compress (second, third and fourth column respectively). The colours give the amplitude (or amplitude minus Gregorian amplitude anomaly) in mm, the contours give the co-phase lines (every 45 degrees), grey for Gregorian, and white for Reset, Drift and Compress respectively.**

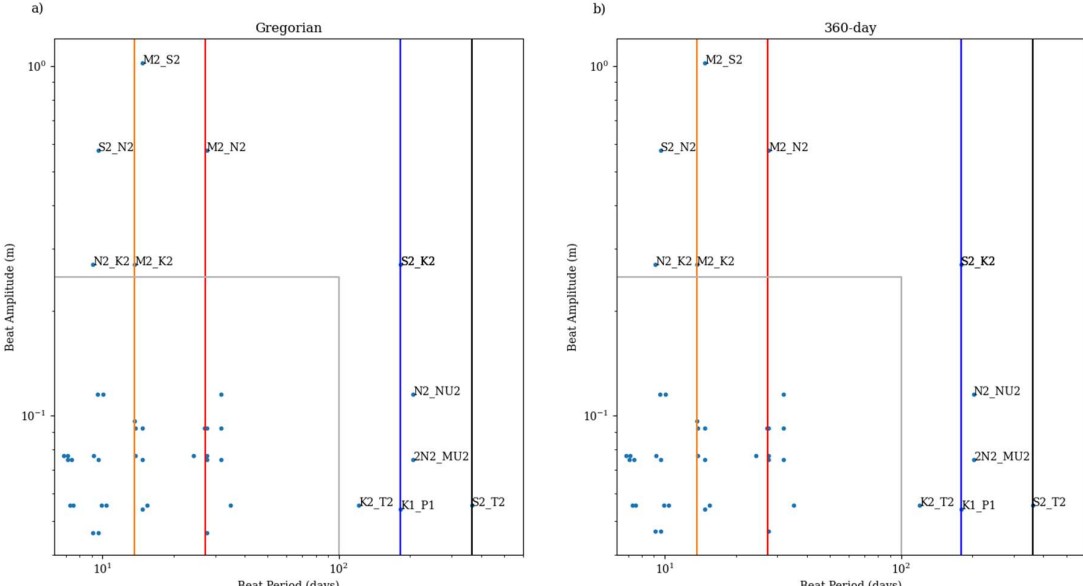

**Figure B4 Beat frequency and amplitude for the standard Gregorian tidal constituent frequencies (left), and the modified 360_day frequencies under the Compress tidal scheme (right). the yellow and red lines show the monthly, and bimontly frequecies, and the blue and black lines show the biennial and annual frequencies (2π/(365.24x24) and 2π/(360x24) respectively). The grey box shows the beat pairs with periods less than 100 days, and ampitudes less than 0.2m.**

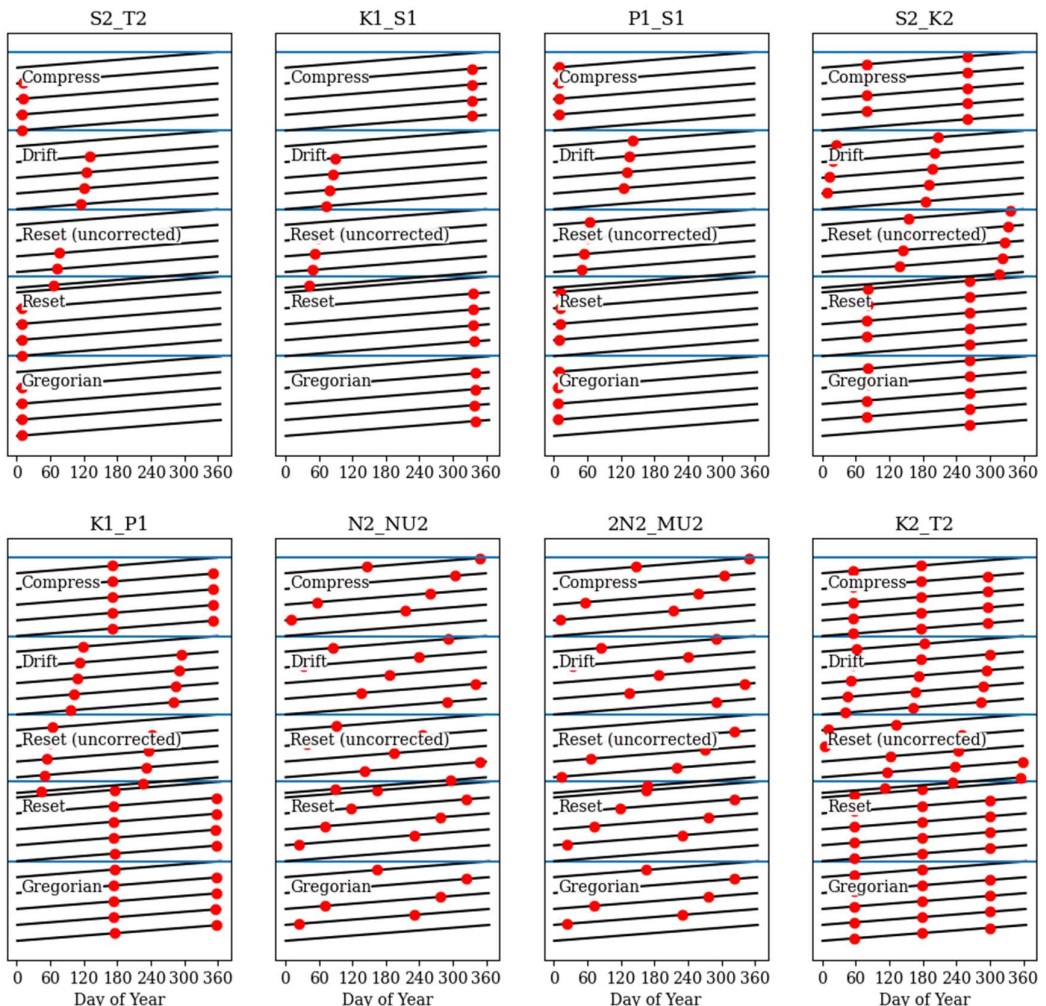

1985     **Figure B5 The timing (day of year) of the maximum of the beating tidal constituent pairs (with periods greater than 100 days) for the four different tidal systems for an example location. Each subplot is a beating tidal pair. The black line represents the day of the year (in the x direction) and the decimal year (in the y direction). The red dot is the local maximum of the beating pair. Each tidal system (Greg, Reset, Reset (uncorrected), Drift, and Compress) is stacked on top of one another.**

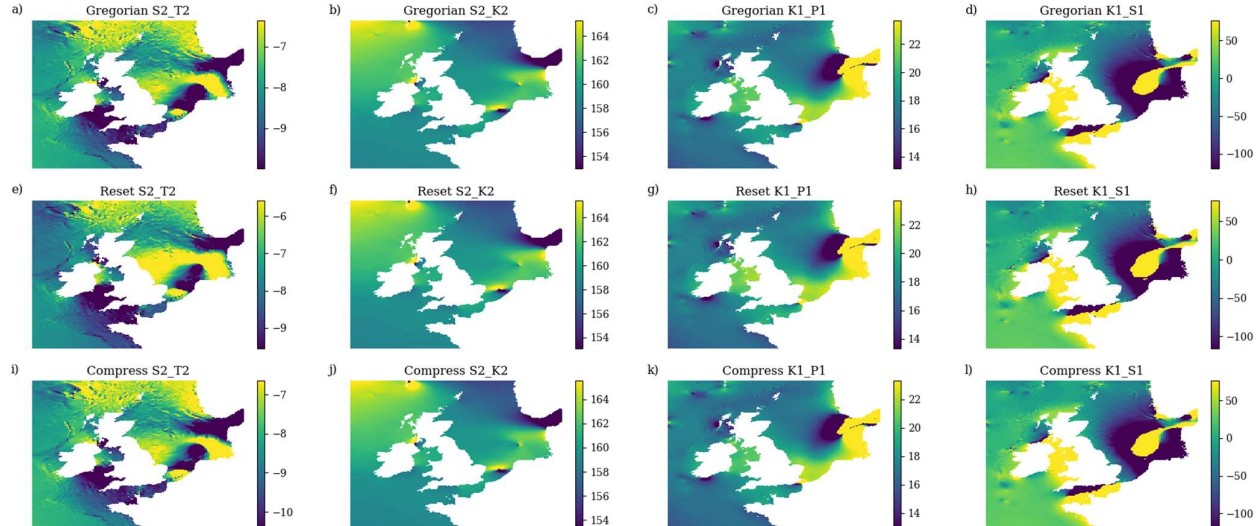

**Figure B6 Timing of largest beat pair maximum (in day of year), for pairs that occurs at approximately the same time of each year. Note $S_2$-$K_2$ are the beating pair with the greatest amplitude.**