# Peer review of "21st century marine climate projections for the NW European Shelf Seas based on a Perturbed Parameter Ensemble."

_EGUsphere, 2023_

## Author Response (AR1)

**Response**

Dear editor and reviewers,

Thank you for your reviews, and suggestions. We have carefully worked through your points and have modified our manuscript to reflect them.

Both reviewers mentioned our use of RCP8.5. Reviewer 1 wanted a scientific elaboration on our use of RCP8, while second reviewer 2 thought that RCP8.5 was still relevant to climate risk planning. We have added a section into our introduction to explain and defend our use of RCP8.5.

Reviewer 2 asked for a more in-depth evaluation, and assessment of the currents, stratification, and more explanation of the observed salinity changes. We have expanded our evaluation, and added additional evaluation against the ICES long term mooring time series (with an additional figure). We have assessed the NWS circulation with volume transport cross sections, and the mean and variability for the barotropic residual currents (with an additional figure in the paper, and several additional ones in the supplement. We have assessed the timing and duration of stratification, which shows a substantial lengthening (with an additional figure). We also show that the salinity change on the shelf is consistent with the changes in the winder North Atlantic, in the driving simulations.

Both reviewers also gave a list of minor edits which we have corrected, although may have introduces new ones in our news sections.

We have now copied the reviewer comments below, and reply to each point in red to ensure we have covered all their points, and point the reviewers to the appropriate line number and section in the manuscript to aid their reassessment.

The revised manuscript is much more in-depth, commensurate with the topic. We have tried to mitigate some of the increase in size from the additional material by moving the data description (~3000 words) to the appendices.

I hope you are happy with this revised version. I will be away for February, and early March, but should still be in email contact.

Kind regards

Jonathan, Matthew, and the co-authors.

**Reviewer 1**

**RC1: 'Comment on egusphere-2023-1816', Anonymous Referee #1, 02 Oct 2023**

The paper is concerned with marine climate projections for 21st century for the north-west European shelf seas. The study is based on numerical modelling and consists of dynamical downscaling of 12 members of HadGEM3-GC3.05 Perturbed Parameter Ensemble using a model NEMO at 7 km resolution. These ensembles are run under the high greenhouse gas emissions RCP8.5 scenario for the time period 1990-2098. The results are then processed to obtain end of 21st century projections of main essential climate variables. The paper is written clearly, is well structures and goes into a great level detail as regards models set-up, including the availability of the output, the directory structure, etc. The results are also

presented in clear and concise way. I recommend this paper for publication given that major comment is addressed in the revised paper and some minor technical/grammar issues are addressed.

**Major comment:**

In the introductory section the authors should elaborate as to why RCP8.5 scenario was chosen. First of all, it is part of the older CMIP5 pool of scenarios rather than one of the CMIP6 scenarios. Outputs from global CMIP6 models have been already available to the scientific communities for downscaling for a few years, thus authors should justify the use of CMIP5 rather than the newer CMIP6. Furthermore, RCP8.5 (and CMIP6' SSP5-8.5), fossil-fuelled development with very high GHG emissions, are widely regarded as highly unlikely scenarios with a consensus that we are already not following the RCP8.5 trajectory. Therefore, I marking a paper down on its scientific significance and I am recommending that the authors elaborate on its scientific value in the revised version. Perhaps the paper still presents a value to the scientific community and it would be good to hear the opinion of the authors. The conclusions section needs to clearly state that reported projections concern a very pessimistic, highly unlikely scenario.

We have added a substantial section into our introduction to respond to this point (lines 85-104 of the introduction). To address your concerns directly we respond directly to your main points:

1) Why RCP8.5 was chosen?
   The choice of forcing scenario is motivated primarily by the desire for a set of NWS marine climate projections consistent with the latest set of UK Climate Projections (UKCP18), which were run under RCP8.5, as their development was begun before the SSP scenarios became available. This consistency allows researchers to look across both the land and marine domains in multi-variate space in a way that has previously not been possible to facilitate, e.g., consideration of compound hazards and the combined effects of multiple climate-impacts drivers.
   See lines 88-93.

2) Why we use a CMIP5 models rather than a CMIP6 models?

   HadGEM3-GC3.1 is the Met Office CMIP6 model, and is very similar to HadGEM3-GC3.05 which we use here (line 115). So while the scenario we use is from CMIP5, the model we use is from CMIP6.

3) Explicit acknowledgement that projections are very pessimistic, highly unlikely scenario.
   RCP8.5 is a relatively high impact "business as usual" scenario from the CMIP5 suite of models, rather than the more recent CMIP6, which are based on the "Shared Socioeconomic Pathway" (SSPs). RCP8.5 has very similar total radiative forcings to the SSP5-8.5 scenario (Tebaldi et al., 2021), although RCP8.5 has a slightly weaker global temperature response, attributed to a lower $CO_2$ concentration (Fyfe et al., 2021).

Recent studies have criticised the use of RCP8.5 in climate projections, particularly in terms of its apparent low likelihood, given emissions reductions pledges associated with the Paris Agreement (Hausfather and Peters, 2020). However, there are several scientific reasons why this remains a useful scenario in the context of policy-relevant climate information. Firstly, it has a high signal-to-noise ratio and can therefore better separate the forced climate response from internal variability. Secondly, RCP8.5 can be readily translated into warming levels, which was the primary basis of the last UK Climate Change Risk Assessment and appear prominently in the 6[th] IPCC Assessment Report (AR6; IPCC, 2021a). Thirdly, risk-based decision making requires a comprehensive picture of the future risk landscape, including higher warming levels associated with any combination of emissions back-tracking, positive carbon-cycle feedbacks, and high climate sensitivity(IPCC, 2021b). Finally, high emissions scenarios such as RCP8.5 provide a useful baseline scenario from which the benefits of mitigation action and avoided costs can be assessed.

This is included in the updated manuscript in lines 85-88, and 94-104.

**Technical comments:**

Ln73: add is after This

Done

Ln79: change evaluation to evaluate

Done

Ln113: change interpolates to interpolate

We changed interpolated to interpolate

Ln122: remove can

Done

Ln124: remove can

Done

Ln128: change is it to it is

Done

Ln130: Figure 3 is introduced before Figures 1 and 2. Figure numbering needs to be revised in the manuscript

Due to the reorganisation of the paper, all the figure and table numbers were revised.

Ln140: remove is

Done

Ln311: change time so to times of

Done

Ln428: change is to are

Done

Ln496: change process to processes

Done

Ln499: change oecan to ocean

We couldn't find this in the manuscript.

Ln608: change if to of

Done

Ln623: change it is to its

Done

Ln623: change computationally to computational

Done

Ln759: add in after interested

Done

Ln776: remove be

Done

Ln778: change give to gives

Done

Ln779: change response to responses

Done

Ln1118: change to no to do not

Done

Ln1211: change which to while

> Done

**Reviewer 2**

**RC2: 'Comment on egusphere-2023-1816', Anonymous Referee #2, 10 Oct 2023**

**General review comments**

In "21st century marine climate projections for the NW European Shelf Seas based on a Perturbed Parameter Ensemble", Tinker and colleagues present a dynamic regional downscaling of a global climate model projection (HadGEM3-GC3.05) for the high emissions scenario RCP8.5 for the north-west European continental shelf region on a 7 km spatial resolution grid using the NEMO coastal ocean model. The paper describes the regional downscaling experimental design, as well as an extensive evaluation of the model against observations and other model data.

The paper is generally well written, and the dataset will be of great value to researchers working on understanding the impacts of global warming on the coastal seas of north-west Europe, especially with a focus on informing decision makers of the adjacent countries on climate action to adapt to threats and opportunities of climate change. The overall structure and detail are good, but there are some significant improvements to be made before the manuscript can be published. I have the following more general comments on the manuscript, in addition to those relating to specific lines.

A list of minor spelling and grammar issues encountered during the review are included at the end for consideration by the authors, but the sections below outline the main points for attention by the authors to improve the manuscript.

> Thank you for your detailed review, we have tried to respond to it in full, but this has led to several new sections and figures, so we have tried to find a balance between a comprehensive response and the overall length of the paper.

**Novelty to the international research community beyond the UK**

At times, the writing is relatively "parochial" to the UK and its climate reporting structures, while the model region covers many other countries EEZs. Through international legislation (Marine Strategy Framework Directive and Oslo-Paris Convention), much of the marine region represented in the model domain is of relevance on a wider international stage. Below some specific examples where the authors can address this in the manuscript and ensure the work receives also some international recognition.

> Thank you for your comment and suggestions. We have consulted with external expert colleagues, and have added a section into the introduction, that hopefully addresses your concerns.

> The full section is in lines 53-77, but the we have copied the first few sentences (first 7 out of 24 lines) here:

> "Beyond the UK, there are numerous legislative mechanisms that support the protection and management of the NWS Marine Environment. At the European level examples include: the EU Common Fisheries Policy (CFP, 2013); The EU Habitats Directive (1992); The Oslo-Paris (OSPAR) Convention for the Protection of the Marine Environment of the North-East Atlantic (1993); and The EU Marine Strategic Framework Directive (2008). There are also important international treaties and conventions such as the United Nations Framework Convention on Climate Change (UNFCCC, 1992) and the Convention on Biological Diversity (CBD, 1992). These are all implemented at the UK level (Frost et al., 2016) and all have varying and overlapping goals and targets linked to the protection and monitoring of the marine environment… "

- L17-18: The authors consider this dataset the "state-of-the-art for marine UK projections". Do they mean projections from a UK climate research group, as in the state-of-the-art that the UK science community produces; or do they mean state-of-the-art for the UK marine region.  If the latter, could this be extrapolated for the wider north-west European continental shelf?  If not, which other groups are providing state-of-the-art climate projections for this region, and how do these compare with the dataset presented here?

  We consider these to be state-of-the-art for NWS marine climate projections. We agree that calling them UK projections is a little "parochial", and that they may be of interested to reader from many other countries. We have changed the text in the abstract, on line 17, to read:

  "These simulations represent the state-of-the-art for NWS marine projections."

- L41-51: The focus here is strongly on the UK's evidence requirements for reporting under its national structures. The Oslo-Paris Convention published an ambitious strategy for the North-East Atlantic region, which has a strong emphasis on climate change and ocean acidification.  The authors may wish to use this and other international legislative frameworks covering climate change (UNFCCC) or marine (EU Marine Strategy Framework Directive) to highlight the relevance of these simulations outside of the UK research community.

  Thank you for pointing this out. We have hopefully captured this in the new paragraph (see above) in the updated manuscript, see lines 53-77.

**Advocacy for RCP8.5 still being relevant to climate change risk planning**

The chosen RCP has not been part of the most recent suite of climate models under the IPCC reporting structures.  There is increasingly wide recognition that our progress on climate targets means that the narrative of RCP8.5 is increasingly unlikely.  However, my understanding is that the total resultant radiative forcing of 8.5 W/m$^2$ is not beyond reality, especially when considering uncertainties in climate model processes and their parameter space.  The authors should guide the reader in understanding why the RCP8.5 suite of downscaled ensembles is still relevant to help policy makers and businesses plan for the worst eventualities of climate change impacts.

Thank you for your comment, we have added a new section in the introduction (lines 85-104) in response to the first reviewer, and your comments were very useful. You might find our response the first reviewer useful.

**Discussion of the salinity signals**

The description of the salinity signals in the present day and future climate scenarios raises more questions than the authors have referred to.

L584-586: The freshening is glossed over quite quickly here. What is the source for this? Is this entirely driven by a reduction in inflow of Atlantic water from the adjacent open ocean? From Figure 9 this doesn't appear to be the case and instead may be in the global climate model simulations? Was this something that has been reported for HadGEM3 PPE?

We have now discussed the freshening in more detail, assessed the larger scale North Atlantic salinity change in the driving simulations, and shown that this is the main source of the freshening. There is a substantial freshening at the lateral boundaries, that propagates into the model, and is of a similar strength and is correlated across the ensemble – see figures S28, S29. It is beyond the scope of the current study to investigate the drivers of the freshening of the driving models, but we have speculated in the discussion as to the possible cause.

We do not think that the freshening is caused by a switch of the North Sea circulation configuration toward a more estuarine type as posited by Holt et al. (2018). We now have a new circulation section (sections 2.5 and 5.1), and discuss the role of circulation in the NWS salinity change – this is discussed in from line 849:

"However, we do not see the North Sea circulation configuration change seen by Holt et al. (2018), which effectively isolates the North Sea from the Atlantic, with the North Sea becoming more estuarine, so while our slight reduction in the modelled NNSI will play a role in the NWS freshening, the main driver is likely to be the freshening of the North Atlantic."

L602-608: The present day appears to be simulated as saltier than the reanalysis model, and the agreement between ensemble members of the trend in freshening appears to break down into the future (my interpretation of the statement "divergence of the ensemble"). Is this due to parametrisations in the water cycle and how different ensembles explore parameter space for this?

Over most of the NWS, there is a relatively good agreement (or near agreement) between the NWSPPE and the RAN, with the RAN being within the NWSPPE range. There are larger differences in the Skagerrak and Norwegian Trench. This is likely due to differences in the treatment of the exchange with the Baltic. This is a notoriously difficult region to model, and which we discuss from line 738 We note that the RAN is too salty in this region, so our simulations may be more realistic here.

Your interpretation is correct, that the divergence in the ensemble SSS through the 21$^{st}$ Century reflects the different ensemble members having different freshening trends. We link this to the different North Atlantic 21$^{st}$ Century freshening across the ensemble. In supplementary Figure S24a, we correlate the NWS freshening (how each

the NWS freshens across the 21st century in each of the 12 ensemble members, to the 21st century HadGEM3-GC3.05 freshening across the 12 ensemble members, for each point. This shows that the ensemble response (which ensemble members freshen most) is highly correlated between the HadGEM3-GC3.05 North Atlantic, and our NWSPPE. Figure S24b then shows that the absolute values of freshening are also comparable.

There are many parameters that are perturbed within the PPE, with a very complex response, and so understanding the water-cycle response to parameter changes would require dedicated research that is beyond scope of the present study. However, the water cycle diversity within the HadGEM3 PPE solely propagates into our model simulations through the Atlantic Lateral Boundary Conditions and the surface fluxes, as we use climatological rivers and Baltic LBCs. This is an important limitation for understanding the NWS water cycle response as the climatological rivers and Baltic LBCs don't "see" any hydrological changes simulated in the parent GCM. We say this on line 917.

**Attention to the circulation (tidal currents and the residual currents)**

The focus of this manuscript is on evaluation against SSH, temperature and salinity. However, the regional model should also be capable of representing changes in the residual circulation of the North Sea. The inflow of Atlantic waters along the northern boundary is particularly important for the productivity of this region. A notable manuscript by Holt and colleagues in 2018 (https://doi.org/10.1029/2018GL078878) presented changes in the Atlantic water exchange of this region. The manuscript makes no reference to this paper, or whether these changes are observed in the dynamic downscaling of the HadGEM3-GC3.05 PPE ensembles. In addition, tidal current strength is an important forcing mechanism for the position of frontal regions in the north-west European shelf region.

Thank you for your comment. We now evaluate the modelled residual circulation in terms of volume transport against observation-based estimates (Sections 3.6 and 4.6). We have also added an in-depth analysis of the residual circulation of the NWS (see sections 2.5 and 5.1, starting on lines 232, 561 and discussed from line 825). We now cite Holt et al. 2018 in the revised manuscript - thank you for highlighting it – and investigate whether we see the same change in the NWS circulation configuration. We see a general reduction in residual circulation strength on the NWS, but we do not see that large scale North Sea circulation change as seen by Holt et al. 2018. However, we do note a change to the south and west of Ireland. We do not assess tidal current strength which we consider outside the scope of the paper, although do assess the tidal characteristics with a co-tidal chart in Figure 1.

**Discussion of key shelf sea processes**

The authors present some of the results on shelf sea stratification and mixed layer depth, but these could be brought more to the fore. There is no mention of the calculation method for PEA, although there appears to be a distinction in the supplementary materials of the PEA due to the salinity structure. There is brief mention of the spatial pattern of PEA, but no mention of any significant changes in the onset or duration of stratification changing under the projected warming scenario.

In the revised manuscript we have defined the Potential Energy Anomaly, and note the temperature and salinity components of it (see section 2.9). We have analysed and described changes in the seasonal cycle of SST (etc), noting that the greatest warming is in the autumn (see paragraph starting on line 677). This leads us into an assessment of the timings and duration of the stratification (see section 2.10, and paragraphs starting on line 701 and 810). We have shown that the spring onset is a few days earlier, but the autumnal break down is substantially later, which is consistent with the change in the seasonal cycle. We later discuss these results, and consider their implication on the ecosystem (see paragraph starting on line 810).

**Quality of graphics and captions**

The authors should review the graphics and how these are included. For many figures, the continuous colour bar makes it difficult to really discern the spatial patterns and the magnitude of signals. A more discrete colour scheme and associated colour bar should be considered. The continuous colour bar in some panels is also meaningless (e.g. Figure 3f). Many of the figures also include too many small panels, which make them difficult to view. Even on a PDF, you need to zoom significantly to view the images. On a printed page, this is even worse. While screen reading is probably more and more common, the authors should consider those with small screens and/or those using a printed copy on A4 paper. Finally many of the captions are insufficient – each figure caption should be capable of being read independently of any other image. The cross-referencing to other figure captions for the details of what is shown is frustrating for the reader.

Thank you for your suggestions. We have made the following changes to address your concerns. Where we use continuous colormap, we have included contours at the colorbar tick levels, to aid interpretation. Where continuous colormaps don't make sense (like in the old Figure 3f), we have used appropriate discrete colormaps. The main SST/NBT/SSS etc change maps (Figures 9, 10, 11, 13) with both mean and variance, have been moved to the supplementary material (Figures S17-S22), and replaced with large mean change maps (for the annual mean, and the 4 seasons). We have also expanded the figure captions, so there is no cross-referencing required.

We will include the high-resolution figures zipped as a supplement).

**Choice of regions**

There appear to be two sets of regions that results are presented for: one as applied in the evaluation against EN4, and one to calculate regional statistics. Why are these regions not the same? Is it due to the underlying data in EN4 (although the highly sampled North Sea should really be within the ability to be evaluated at higher granularity)?

Yes, the regions are different due to EN4 data sparsity in some location, so we decided to aggregate regions up for the EN4 evaluation. We otherwise use the Wakelin regions as they are widely used in the literature, are based on oceanographic conditions and geographic regions, and give a good granularity. We now include the Wakelin regions in the supplement (Figure S2), so the user can see the EN4 stats in the three Wakelin North Sea regions.

**Comments relating to specific lines**

L64-77: The authors chose to add the description of the HadGEM3-GC3.05 PPE in the
"Data" Section.  I would suggest this is moved to the "Model and Methods" Section.  There
is already a section here on the HadGEM3-GC3.05 PPE, and by placing in this section, it
may sit better alongside the wider description of the model and its forcing datasets.  There is
information in Section 2.1 that is not included in 3.2, so the authors should merge these
sections (rather than removing 2.1).  I would suggest the "Data" Section focuses on the
datasets used to validate/evaluate the projections.

Good idea, We have done this.

L136-139: Technical comment on choice of MDT product: The authors use the AVISO mean
dynamic topography to compare to the NWSPPE mean SSH for the present day.  It is my
understanding that the AVISO MDT has had significant improvements since that published
by Rio and colleagues in 2014.  These include revisions in 2018 and 2022.

I had actually used the CLS2018 dataset, and have now clarified this in the
manuscript.

L153-154: The tide gauge at Smögen was excluded due to the significant influence of
GIA.  On line 333, the tide gauge at Bergen is also highlighted as potentially affected by
GIA.  Is this not a known issue, and therefore wasn't excluded before the analysis?  I am
trying to understand why one gauge was excluded from the analysis, but not the other, even
though both are highlighted as having the same issue.

The Smögen tide gauge record appears to be dominated by the GIA, with a substantial
sea level rise that is not represented in our model simulations, and so was excluded. I
noted the possibility of the GIA component of the Bergen tide gauge given its location
on GIA maps, however, We have removed this comment, now, as it was speculation.

L251-252: 3 ensembles were excluded from the original set of 15 in the global PPE (as
described in L225-227).  I would add a brief internal reference to this description here, or add
a "… due to unrealistic representation of the AMOC (see description above)" to aid the
reader in understanding the reduction from 15 members to 12.

Thanks, good point. Given the reorganisation of the text, We have simply removed
this first mention of the 3 excluded ensemble members, as the additional info isn't
necessary at this point.

L268-333: The entirety of Section 4 is brief in its assessment of the model's performance,
while this should be a significant part of the manuscript.  Based on Figure 2, the OSTIA data
are outside of 1.96 ensemble standard deviations for much of the domains, with the exception
of autumn.  The authors do not state whether this therefore requires bias correction of the
projections.  The authors may have incorrectly worded the description, but I would disagree
with the statements in L301-304 given the OSTIA is more than 1.98 standard deviations from
the ensemble mean.

To address you concern on the amount of model assessment, we have expanded this
section, with additional analysis, explanation, and datasets. We have added evaluation

against the ICES climate time series, and volume transport estimates through cross-sections.

The NWEPPE SST is highly correlated with the OSTIA data, and have mean absolute biases less than 0.5 °C. Given all the possible ways of showing the data, We wanted to be as critical of the model performance as possible, but perhaps it is better to start from the annual mean, and then move to the seasonal (or even monthly fields). The basic model climate in terms of the annual mean SST is very good, with most of the NWS within 0.7°C of the OSTIA data, and almost all the NWS having he OSTIA SSTs within the NWS ensemble. It is when you start to look at details of the seasonal cycle that you see regions (to the west of the UK) that are too warm in summer, and in the northern North Sea (and around Scotland) where the SSTs are too cool in the autumn. We have now included an annual mean panel in the figure, and expanded the text.

L305-316: While EN4 offers a great assemblage of the data collected over the entire spatial scale of the north-west European shelf, it lack the extraction of high resolution time series and fixed stations/hydrographic sections. There are some data products out there that do provide such time series (e.g. through the sites monitored for UK Marine Strategy at coastal locations such as Western Channel Observatory and Scottish Coastal Observatory, and through the ICES Working Group on Oceanic Hydrography's Report on Ocean Climate and associated data time series). While the authors point out the inability for EN4 to provide such time series, they have not highlighted that there are some products out there that could remedy this.

Thank you for this suggestion. Long time series allow the comparison of the model and observed climate, without being contaminated by the "weather", which the EN4 analysis suffers. We have now undertaken additional evaluation against 11 ICES T and S long term climate timeseries, and have compared he mean values, and the interannual variability (see section 3.3 and 4.4 and Figure 4).

L439-443: The wording describing the value of the time stamp associated with the climatological seasons is somewhat confusing. The paragraph includes two different examples, first for summer then for winter. The paragraph would be more clear if the example of the second paragraph would be consistent with the previous sentence.

Good point, We have changed this to consider winter in both cases – see section 15.4.2, line 1645

L689: For SST the statement re. confidence in near future projections is likely warranted, but based on my understanding of the Time of Emergence as presented here, I am not sure I would agree with that assessment for salinity, especially given the divergence in the ensembles of future SSS response.

While the SSS doesn't emerge across the NWS, emerges later, and has greater spread, there are some locations where it does emerge.We have therefore changed this statement to *"This can guide where and when to use the NWSPPE for near future temperature and salinity projections."* – see line 872.

L1085: Table 1 – missing that this is for the datasets used, not the quality. The table lacks the description for the assessment of sea level variability.

*This has been added, and the table has been expanded to include the new data sets.*

Figure 5u – the markers for the locations are very difficult to read and may benefit from being in a more obvious colour (such as red).

*We have expanded the map, and change the letters to red (see table 6)*

**List of minor edits (grammar/spelling/clarity).**

Throughout: inconsistency on the spacing of "datasets" (sometimes also "data sets").

*Done, all now are datasets*

L24: Atlantic Meridional Overturning Circulation [remove "s"]

*Done*

L46-48: This is a long sentence. I would suggest splitting in two shorter sentences. One ending after "evidence" on line 47 and the second starting "However, as the NWS climate projections had not been updated, assessment of those aspects …".

*Done*

L62: … observational data sets… [error in "observational data"]

*Done, no in section 3*

L69: "GSI18.0 science" is not the correct phrasing. I would suggest "GSI18.0 scheme" or "GSI18.0 sea ice configuration".

*Done*

L72-73: This is discussed later. [missing "is"]

*Done, also noted by reviewer 1*

L76-77: Tinker et al. (2020) lists all the HadGEM3-GC3.05 variables use to drive our simulations, and their frequency, in their Table S9. [typo in "to drive" and "simulations"]

*Done*

L94: There are few EN4 locations where observations are available … [remove "there are"]

*Done*

L127-128: "… we can then give the number of NWSPPE standard deviations it is from the ensemble mean." [word order+ plural missing]

Done

L237: … North Atlantic Oscillation, and Atlantic Multidecadal Variability. [missing "and"]

Done

Line 332-333: Do you mean "Le Havre"?  I think the letters have been muddled here.

Done

L352: … for each ensemble member [remove "s"]

Done

L446: … associated with uncertain parameters

Done

L447: … then the ensemble mean

Done

L453: … how the ensemble mean and ensemble standard deviation have changed

Done

L454: … this does not give information on the uncertainty …

Done

L533: The text refers twice to DMU_T.  Does the second one need to be DMV_T?

Done

L660: … while the pattern of … reflects the complex [missing "s" on "reflect"].

Done

---

## Author Response (AR2)

**Response**

Dear editor and reviewers,

Thank you for your final review, and your conditional acceptance.

We have made both of reviewer 2's requested changes, and go through these point-by-point.

**Review 2**

I have two minor requests for revisions. There are some minor text edits (plurals/verb agreement etc), which I believe the copy editing process will pick up.

Firstly, for the lines 66 (from sentence starting "Similarly") to line 74 (sentence ending "NWS.") to be removed. These contain some factually incorrect statements (e.g., around the evidence groups – these are UK Marine Strategy, not OSPAR subsidiaries) and also unnecessary detail. It would shorten the introduction a little.

We have removed this sentence.

Secondly, the ICES time series that have been presented are not all from moored instruments (some are from ship-based hydrography). There are a few mentions to moorings in the text - find/replace should solve this quite quickly.

We have changed the text from **"NWS mooring"** to **"NWS *in situ* observation"**.

We have also changed the heading of Figure 4 from Mooring to *in situ*.

We have changed **"ICES long term climate moored timeseries"** to **"ICES long term climate *in situ* observation timeseries"** in the supplementary information.

Following the notes from the previous review file validation, we have also removed the table cell colouring, and have renamed 16.4 Appendix Tables.

Hopefully we have made all your suggested changes, if there are any other points, please contact me again,

Thank you again for all you work on this, and you help to improve our paper,

Kind Regards

Jonathan, Matthew, and co-authors.

---

## Author Response (AR3)

**Response**

Dear editor

Thank you for your final decision, and your conditional on some the small technical corrections.

We have made all of your requested changes, and go through these point-by-point.

> **Public justification (visible to the public if the article is accepted and published)**:
> Thank you for responding to the referee's suggestions. I am happy to accept your paper subject to some small technical corrections listed below. Congratulations!
>
> Throughout text and also Table 8: Salinity has no units (since the 1980 equation of state) so please remove all mention of psu. Instead, on first mention of salinity (and in the caption of Table 8), state that all salinities will be given on the practical salinity scale and therefore have no units. (Of course, it would be better to use absolute salinity as defined in TEOS-10, introduced in 2010, which would have units of g/kg, but I imagine your model produces salinity on the practical salinity scale).

> Our models use EOS80, although we are currently implementing TEOS-10.
>
> We have added a note in the abstract saying the units are unitless:
>
> "We project a Sea-Surface Temperature (SST) rise of 3.11 °C ($\pm 2\sigma$ = 0.98 °C), and a Sea-Surface Salinity (SSS) freshening of -1.01 ($\pm 2\sigma$ = 0.93; on the (unitless) practical salinity scale)."
>
> We have made this clear in a small new section of the methods section before any salinity values are reported (apart from in the abstract):
>
> ### 2.11.1 Salinity units
>
> Both NEMO AMM7 and HadGEM3-GC3.05 use the 1980 equation of state (EOS-80; UNESCO, 1983), which works with the practical salinity scale (which has no units). In this study we report salinities on the practical salinity scale.
>
> With the reference
>
> UNESCO, 1983: Algorithms for computation of fundamental property of sea water. Techn. Paper in Mar. Sci, 44, UNESCO.
>
> We have removed the psu from the paper, tables, figures, and supplementary material.
>
> We have removed psu from table 8 (supplementary tables and 7 and 8), and added "Salinities are given on the (unitless) practical salinity scale."
>
> We have also updated all the figures with the psu unit – where a x label was "SSS (psu)" we have changed it to "SSS".

> There are 29 occurrences of "compared to" which should all be changed to "compared with" (or where appropriate, the more succinct "than").

> We have also changed *compared to,* and changed the hyperlink text colour to black.

Hopefully we have made all your suggested changes, if there are any other points, please contact us again,

Thank you again for all you work on this, and you help to improve our paper,

Kind Regards

Jonathan, Matthew, and co-authors.